# Multi-omics analysis of human mesenchymal stem cells shows cell aging that alters immunomodulatory activity through the downregulation of PD-L1

Yuchen Gao [1,4], Ying Chi[1,4], Yunfei Chen [1,4], Wentian Wang[1,4], Huiyuan Li[1], Wenting Zheng[1], Ping Zhu [1], Jinying An[2], Yanan Duan[2], Ting Sun[1], Xiaofan Liu[1], Feng Xue[1], Wei Liu[1], Rongfeng Fu[1], Zhibo Han[1], Yingchi Zhang[1], Renchi Yang[1], Tao Cheng [1], Jun Wei [1] ✉, Lei Zhang [1,3] ✉ & Xiaomin Zhang [2] ✉

Mesenchymal stem cells (MSCs) possess potent immunomodulatory activity and have been extensively investigated for their therapeutic potential in treating inflammatory disorders. However, the mechanisms underlying the immunosuppressive function of MSCs are not fully understood, hindering the development of standardized MSC-based therapies for clinical use. In this study, we profile the single-cell transcriptomes of MSCs isolated from adipose tissue (AD), bone marrow (BM), placental chorionic membrane (PM), and umbilical cord (UC). Our results demonstrate that MSCs undergo a progressive aging process and that the cellular senescence state influences their immunosuppressive activity by downregulating PD-L1 expression. Through integrated analysis of single-cell transcriptomic and proteomic data, we identify GATA2 as a regulator of MSC senescence and PD-L1 expression. Overall, our findings highlight the roles of cell aging and PD-L1 expression in modulating the immunosuppressive efficacy of MSCs and implicating perinatal MSC therapy for clinical applications in inflammatory disorders.

Mesenchymal stem cells (MSCs) are a heterogeneous population with the capacities of tissue regeneration and immunoregulation[1,2]. In particular, MSCs have remarkable immunosuppressive properties and suppress effector T cell responses[3]; therefore, MSCs have attracted intensive attention for their potential application in the treatment of various immune disorders, such as graft-versus-host disease (GvHD)[4], Crohn's disease[5] and exaggerated inflammation during COVID-19 infection[6]. In vitro-expanded MSCs have been widely used in clinical

studies; however, the immunosuppressive efficacy of these therapies based on MSC products has not been thoroughly characterized, which may lead to inconsistent outcomes of MSC therapy in clinical trials.

Cellular senescence is a state that imposes permanent proliferative arrest on cells in response to various types of stress and damage[7,8]. Senescent stem cells exhibit diminished proliferative, impaired self-renewal potential and altered differentiation capacity[9]. Evidence suggests that aging MSCs tend to have a compromised

[1]State Key Laboratory of Experimental Hematology, National Clinical Research Center for Blood Diseases, Haihe Laboratory of Cell Ecosystem, Institute of Hematology & Blood Diseases Hospital, Chinese Academy of Medical Sciences & Peking Union Medical College, Tianjin Key Laboratory of Gene Therapy for Blood Diseases, CAMS Key Laboratory of Gene Therapy for Blood Diseases, Tianjin 300020, China. [2]Tianjin Key Laboratory of Retinal Functions and Diseases, Tianjin Branch of National Clinical Research Center for Ocular Disease, Eye Institute and School of Optometry, Tianjin Medical University Eye Hospital, Tianjin 300384, China. [3]Tianjin Institutes of Health Science, Tianjin 301600, China. [4]These authors contributed equally: Yuchen Gao, Ying Chi, Yunfei Chen, Wentian Wang. ✉e-mail: weijun@ihcams.ac.cn; zhanglei1@ihcams.ac.cn; xzhang08@tmu.edu.cn

immunosuppressive capability[10], and senescent MSCs exhibit impaired potency for inhibiting T cell proliferation[11,12], inducing Tregs[13] and treating GVHD models[14]. However, the underlying cellular and molecular mechanisms are poorly understood. It is also unknown whether the proportions and states of senescent cells in MSC products derived from various donors and different culture passages is associated with dissonant outcomes across clinical trials of MSC therapies[15].

Here, we perform a single-cell transcriptomic analysis of clinically relevant MSCs derived from adipose tissue (AD), the bone marrow (BM), the placental chorionic membrane (PM) and the umbilical cord (UC). The single-cell transcriptomic profiles reveal heterogeneity in cellular senescence. We find that cultured MSCs undergo progressive cell aging, accompanied by the gradual loss of immunosuppressive function. Moreover, compared to adult MSCs (derived from AD and the BM), perinatal MSCs (derived from the PM and UC) have fewer features of aged cells and exhibit more potent suppressive activity. Mechanistically, we validate that the impaired immunomodulatory function of senescent MSCs is associated with reduced expression of the immunosuppressive molecule PD-L1. Furthermore, through integrated analysis of single-cell transcriptome and mass spectrometry-derived proteomic data, we identify GATA2 as a regulator of MSC senescence, and its expression delays MSC senescence and enhances the PD-L1 expression and immunomodulatory function of MSCs. Collectively, these results provide further insights into the cellular and molecular basis for the immunosuppressive potency of therapies based on MSC products and highlight the potential of PD-L1 abundant perinatal MSC products for clinical application in inflammatory disorders.

## Results

### Single-cell transcriptomic analysis of human MSCs reconstructs progressive cellular senescence in MSC products with various origins

The production of MSC products for clinical research and therapy follows the criteria published by the International Society for Cellular Therapy[16] (ISCT), which define limited characteristics of MSCs but underestimate cellular complexity and variations in production processes. To explore the cellular subsets of MSCs, we generated a single-cell transcriptomic overview of human MSCs derived from multiple tissue origins (Fig. 1a). Cells isolated from human adipose tissue (AD, $n = 3$), bone marrow (BM, $n = 3$), placental chorionic membrane (PM, $n = 3$) and umbilical cord (UC, $n = 3$) were cultured and purified in vitro (Supplementary Data 1). According to the ISCT criteria, we evaluated the expression of MSC surface markers (Supplementary Fig. 1a) and tri-lineage differentiation potential (Supplementary Fig. 1b), only samples that passed rigorous quality control were subjected to droplet-based single-cell RNA sequencing (scRNA-seq). Clustering analysis divided 45,955 current MSC cells from 4 tissue origins into seven subsets (Fig. 1b) with distinct expression signatures (Supplementary Data 2). To further confirm the accuracy of our MSC quality control, we verified in single-cell transcriptomes that cells were positive for expression of a panel of positive MSC markers (CD105, CD73 and CD90), and no expression of negative MSC markers (CD45, CD34, CD14 and CD19) was detected in nearly all the cells (Supplementary Fig. 1c), suggesting the accuracy of our sample processing procedure and the high quality of the datasets. In summary, the data presented here provide a comprehensive single-cell transcriptomic database of in vitro-expanded human MSC products.

To further characterize the functional characteristics of each cell population, we performed GSVA analysis to define enrichment scores for each pseudobulk cluster (Fig. 1c; Supplementary Data 3). The top expressed gene sets in each cluster are displayed on the heatmap, which suggested that MSCs from C1, C2 and C3 highly expressed genes involved in cell cycle progression and proliferation, while the transcription of these genes was restricted in C5, C6 and C7, accompanied

by apparently altered cellular functions (Fig. 1c; Supplementary Data 4). To verify the apparent differences in proliferative capacity between cell subsets, we further conducted cell cycle analysis. Consistent with the results of GSVA, MSCs from C1, C2 and C3 were assigned high G2/M or S phase score based on its high expression of G2/M and S phase markers. Cells from C5, C6 and C7 were likely not in active cycling and were assigned G1 phase, indicating a relatively low proliferative potential as compared to cells from C1, C2 and C3. C4 appeared to be a subset of cells in a transition state (Supplementary Fig. 1d). Furthermore, we conducted pseudotemporal analysis to verify the temporal order of single-cell clusters. Monocle analysis predicted that the evolution of mesenchymal clusters occurred on a onefold trajectory, starting from cells in the active proliferating phase (C1, C2 and C3) and ending with C7 (Supplementary Fig. 1e). Next, application of partition-based graph abstraction (PAGA) analysis revealed transcriptional similarity between active proliferating clusters (C1, C2 and C3), and strong connective structures were observed in phenotype space between C5, C6 and C7 (Fig. 1d). Consistent with the Monocle analysis, the PAGA path also suggested that active proliferating clusters gave rise to cells in C5, C6 and C7, while C4 appeared as a node that linking this transition (Fig. 1d; Supplementary Fig. 1e).

Gene set scoring analysis (see Methods) was performed to explore the cause of decreased proliferative capacity. MSCs from C5-7 showed upregulation of genes related to cellular senescence (Fig. 1e) and decreased expression of pluripotency stem cell signature genes (Supplementary Fig. 1f), accompanied by impaired DNA repair and proliferative function (Fig. 1f, g). Based on the available evidence, it seems possible that the downregulation of MSC proliferative and pluripotency stem cell signature genes may be linked to cellular senescence. According to previous studies, MSCs inevitably undergo senescence when expanded in vitro[10], which was proven to decelerate proliferation and gave rise to functional changes that affect its therapeutic effect. To further demonstrate the senescence phenotype of C5, C6 and C7, we performed differentially expressed gene (DEG) analysis. We detected the upregulation of key molecular markers for senescence in C5, C6 and C7 (Fig. 1h). The tumor suppressors TP53 ($p53$) and CDKN1A ($p21^{CIP1}$)[17] were significantly upregulated in C6 and C7. Furthermore, the senescence-associated secretory phenotypes[18] (SASP) factor IL6 and β-galactosidase (GLB1) were expressed at higher levels in C5, as were the genes encoding heat shock proteins (HSPB1, HSPA9) related to the unfolded protein response[19] (Fig. 1h). Other senescence-related genes, such as important constituents of SASP factors (COL1A1, COL5A2, IGFBP5), significant regulators of epigenetic shift in senescence (EZH2[20], DNMT1[21], HDACs[22]) and genes encoding actin stress fibers (MYLK, CALD1, ACTA2, TAGLN) were found to be differentially expressed in senescent and non-senescent clusters (Fig. 1h; Supplementary Fig. 1g). Collectively, our comprehensive bioinformatic analysis suggests that clinically used MSC products from various tissue origins have an underestimated level of heterogeneity, and our findings have also helped reconstruct a progressive senescence process in MSCs, which is characterized by a decline in proliferation capacity and stem cell features.

### Molecular basis of human MSC senescence

We next investigated the biochemical processes underpinning cellular senescence in MSC subsets. Senescence can be induced by various cellular stresses, such as DNA damage accumulation, the unfolded protein response and epigenetic drift. The abundant expression of genes such as TP53, CDKN1A and CDKN2A[23] suggested that the senescence of C6 and C7 is linked to genomic stress (Fig. 2a, b). In addition, the upregulation of CDKN2A ($p16^{INK4A}$) and CDKN1A ($p21^{CIP1}$) down-regulated the expression of cyclin-dependent kinases (CDK2), E2F family transcription factors (E2F1) and LMNB1[24] were also observed (Fig. 2b). Furthermore, we found that MTOR was downregulated in aging clusters, while PTEN and TSC2, which are dominant repressors of

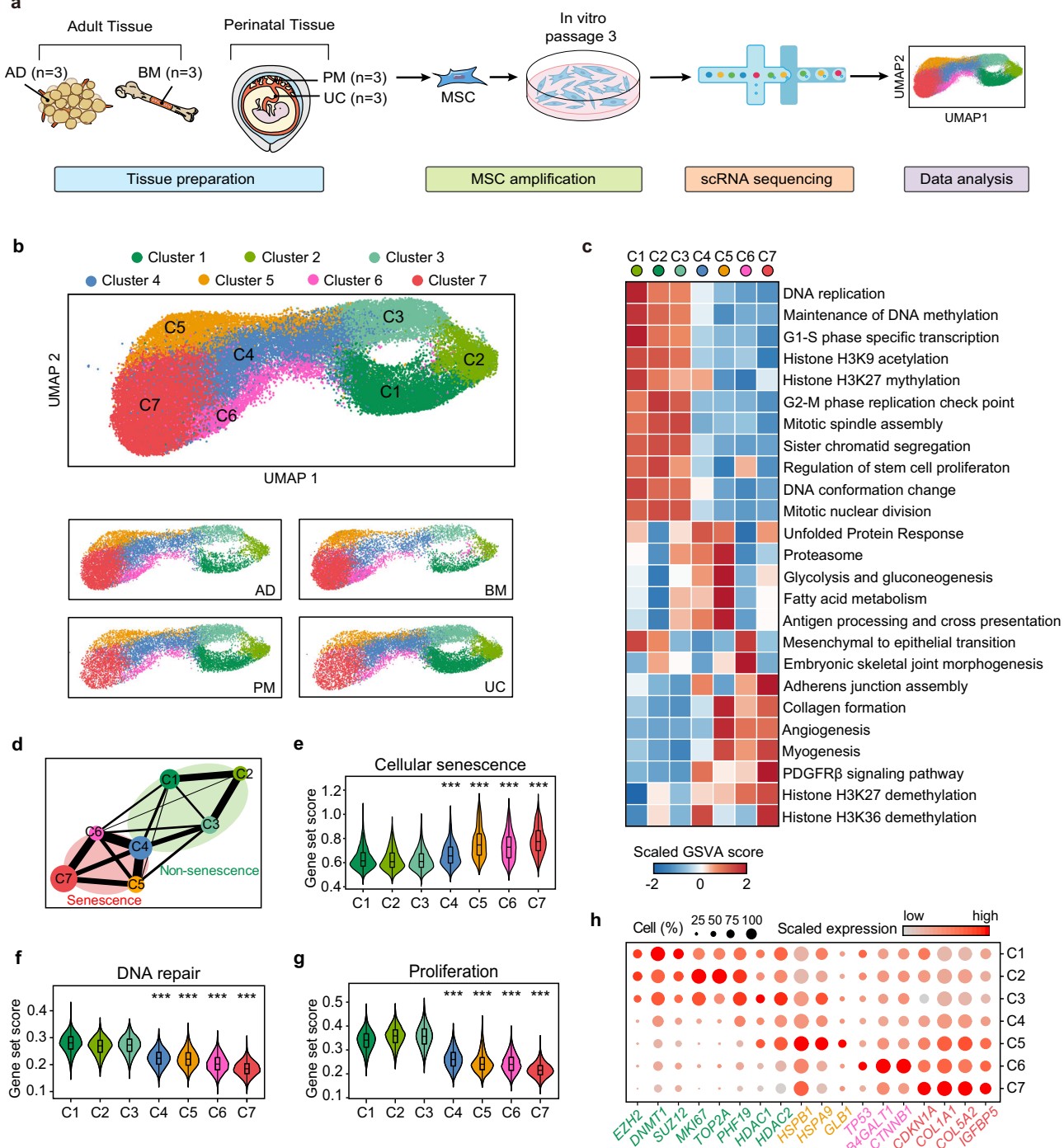

**Fig. 1 | Single-cell transcriptomic atlas of human MSCs reconstructs a progressive cellular senescence process. a** Schematic representation of the study design and experimental procedure. **b** UMAP showing seven subpopulations of MSCs from adipose tissue, bone marrow, placenta membrane and umbilical cord. MSCs were projected together by UMAP (top) and displayed separately by tissue origin (bottom). **c** Heatmap displaying differences in pathway activity calculated by GSVA in distinct cell subclusters. **d** PAGA analysis of each MSC cluster. Nodes represent subsets, and thicker edges indicate stronger connectedness between subsets. **e**–**g** Violin plots showing average expression of cellular senescence genes, DNA repair genes and proliferation genes for each cluster. Box plot within each violin plot indicate median values, and the 25th to 75th percentiles. Asterisks on specific group represent there were statistical differences compared with cluster C1, C2 and C3. The $p$ values were generated by two-sided one-way ANOVA with Tukey's multiple comparisons test. (***$p < 2.2 \times 10^{-16}$, $n = 45,955$ biologically independent cells) **h** Dot plots showing the expression values of representative senescence-related genes for each cluster. Color represents the scaled expression values from Seurat RNA assay. The displayed values were non-batch corrected.

PI3K-AKT signaling, were upregulated in C6 and C7 (Fig. 2b). This is consistent with the report that anti-senescence PI3K-AKT signaling[25,26] can be downregulated by the DNA damage response (DDR)-activated p53 pathway[27,28]. Gene set scoring analysis also suggested that, compared to MSCs from C5 and non-senescent clusters, the unresolved DNA damage might be a contributing factor to the cellular senescence observed in MSCs from C6 and C7, accompanied by downregulated PI3K-AKT pathway activity in these cells (Fig. 2c). Based

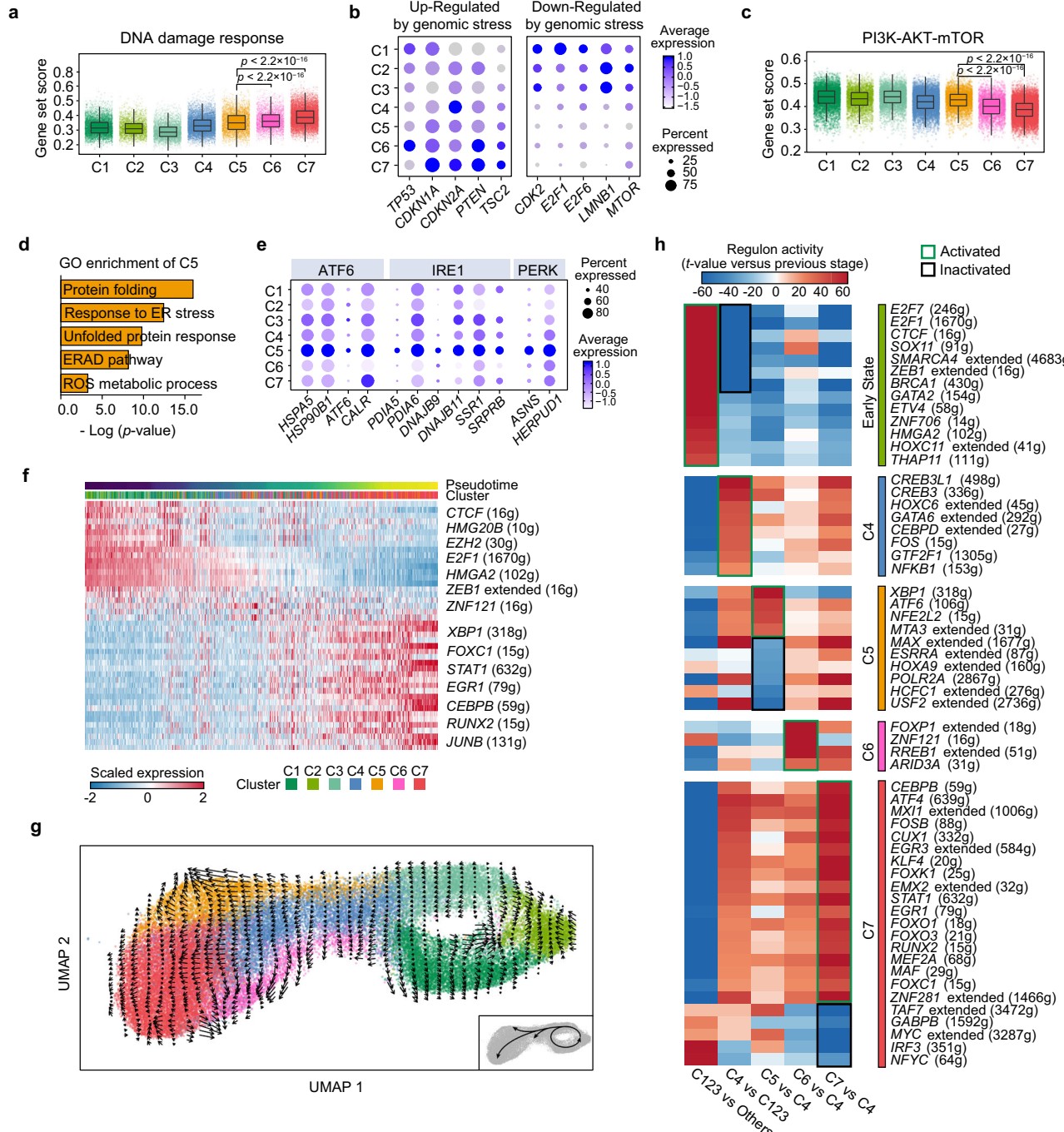

**Fig. 2 | Biochemical processes underpinning senescence progress in MSC.**
**a** Box plot showing average normalized expression value of genes related with DNA damage response pathway. **b** Dot plot showing selected feature genes from each cluster. Including genes down-regulated and up-regulated in DNA damage response. **c** Box plot showing average normalized expression value of genes related with mTOR signaling. **d** Representative Gene Ontology pathways showing enriched expression in Cluster 5. **e** Dot plot showing the proportion of MSCs expressing and the expression level of representative genes associated with ATF6 signaling, IRE1 signaling and PERK signaling. **f** Heatmap displaying the activities of regulons in each cell ordered across pseudotime trajectory generated by monocle, with regulons labels at right. **g** RNA velocity analysis revealing the direction of transformation and the inter-relationship of MSC subpopulations. **h** Heatmap showing the

*t*-values of regulon activity derived by the generalized linear model (GLM, see methods), *t*-values representing activity change between the current developmental stage and the previous one. Only regulons with at least one absolute *t*-value > 20 are showed. Regulons are clustered based on their activation (green box) or inactivation (black box) pattern. For the box plots in **a** and **c**, each single cell was used as an individual sample and annotated in the figure with the color represented by each MSC cluster, the plot center, box and whiskers corresponding to median, IQR and 1.5 × IQR, respectively. In the dot plots, color represents the scaled expression values from Seurat RNA assay, the displayed values were non-batch corrected. For **a** and **c**, *p* values were determined by two-tailed Wilcoxon rank-sum test (*n* = 4621 biologically independent cells in C5; *n* = 3409 biologically independent cells in C6; *n* = 13,105 biologically independent cells in C7).

on these results, we inferred that genomic stress generated during in vitro expansion might be an inducer of cell senescence in C6 and C7.

To further analyze aging-related perturbations in the transcriptome of C5 MSCs, we conducted GO enrichment analysis of marker genes of clusters C5 (Fig. 2d), C6 and C7 (Supplementary Fig. 2a, b). Unexpectedly, the upregulated genes in C5 were enriched in endoplasmic reticulum (ER) stress and unfolded protein response (UPR)-related pathways, including 'response to ER stress', 'protein folding' and 'ERAD pathway', but none of these terms were found in C6 or C7. These results indicate that cellular stress in C5 is likely to be induced by the abnormal accumulation of unfolded or misfolded proteins in the ER. To further confirm the ER stress in C5, we performed gene set scoring analysis, which suggested that the enhanced UPR related pathways (ATF6 signaling, PERK signaling, IRE1 signaling and ERAD pathway) are linked to the senescence of MSCs in C5 (Supplementary Fig. 2c, d). Among the highly expressed genes in C5, we found that HSPA5[29], a well-known central regulator in response to ER stress, was activated in C5, as were UPR genes activated by the ATF6 pathway (ATF6, HSP90B1, CALR), the IRE1-XBP1 pathway (PDIA6, DNAJB11), and the PERK pathway (ASNS, HERPUD1) (Fig. 2e). Together, these results suggest that the loss of proteostasis might contribute to the cellular senescence of C5.

We next explored the transcriptional regulation of MSC senescence. Transcription factors (TFs) related to cell proliferation and anti-senescence factors, such as HMGB2 and E2F1, were highly expressed in the C1, C2 and C3 populations and gradually downregulated in aging clusters (Supplementary Fig. 2e), indicating that transcription factors might contribute to the cellular senescence of MSCs. We also applied single-cell regulatory network inference and clustering (SCENIC) analysis to assess specific global gene regulatory networks associated with MSC senescence (Supplementary Data 5). Assessment of regulons along the pseudotemporal trajectory generated by Monocle indicated upregulation of genes regulated by XBP1, STAT1, EGR1, CEBPB, and RUNX2 and downregulation of genes regulated by EZH2, HMGA2[30] and E2F1, which are known to be downregulated during senescence (Fig. 2f). To further investigate the possible transcriptional networks that drive senescence progression in MSCs, we first performed RNA velocity analysis to trace cell fate and reconstruct cell lineage direction precisely (Fig. 2g). In line with the Monocle results (Supplementary Fig. 1e), cells from C1 to C3 underwent continuous self-renewal cycling, with some of the cells from C3 progressing to the transition state (cluster C4). C5, C6 and C7 are three separate and independent populations derived from C4 cells. Based on the results of RNA velocity analysis, we next constructed a generalized linear model (GLM; see Methods) to characterize regulatory events responsible for the transitions between consecutive MSC subpopulations (Supplementary Data 6). Coarse-grained clustering revealed five regulon groups with distinct activation patterns, including one early activated, one activated in transition state C4, and three specifically activated in the senescent state (Fig. 2h). During the transition of non-senescent clusters (C1, C2, C3) to C4, we observed the dramatic loss of regulatory networks such as E2F1, HMGA2 and BRCA1, which are indispensable for cell cycle progression, anti-senescence function and DNA repair, suggested a potential functional change between C4 and C3 (Fig. 2h). Genes regulated by ZEB1[31], HMGA2, SOX11 and THAP11[32], which reported to contribute to stem cell self-renewal and pluripotency factors, were highly upregulated in C1, C2 and C3 and started to be downregulated in cluster C4 (Fig. 2h). This phenomenon may be attributed to the activation of the DNA damage response, as genes regulated by EGR1, FOXO1 and FOXO3 started to be upregulated in C4 and peaked at C7, with transcription factor networks regulating SASP production (CEBPB, NFKB1)[7] were also activated in C5, C6 and C7 (Fig. 2h). Furthermore, transcription factor genes (XBP1, ATF6) involved in the UPR were found to activate the transcription of genes in C5 (Fig. 2h).

Interestingly, we also identified several regulons that are seldom reported in MSCs (Supplementary Fig. 2f), including transcription factors contributing to nervous system development (CUX1, EMX2), embryonic stem cell self-renewal (THAP11) and hematopoiesis (GATA2)[33]. These regulons are likely to regulate senescence progression in MSCs. Collectively, our data offer valuable insights into the underlying biochemical mechanisms and molecular regulatory networks that may be involved in MSC senescence.

## Perinatal tissue-derived MSCs are less senescent than MSCs isolated from adult tissues

MSCs derived from various human tissues may be heterogenous in their cellular senescence state. Although RNA velocity analysis suggests that MSCs from adult and perinatal tissues share a similar cellular transition pattern (Fig. 3a), we found that MSCs from different tissue sources differed substantially in relative cell frequency per cluster (Supplementary Fig. 3a). Non-aging clusters C1, C2, and C3 exhibited a significantly increased frequency in perinatal MSCs, whereas the frequency of C7 was significantly decreased (Fig. 3b; Supplementary Fig. 3a). Furthermore, the heterogeneity in aging levels is reflected in an increase in the proportion of C7 in adult MSCs, but there was no significant expansion of C5 and C6 (Fig. 3b; Supplementary Fig. 3a). Consistently, the results from the CCK-8 proliferation assay showed that the absorbance at OD 450 nm was decreased in BM and AD MSCs (Fig. 3c), which indicated that cell proliferation was significantly impaired with the increase in senescence. These results suggest that compared to perinatal MSCs, MSCs derived from adult tissue exhibit higher levels of cellular senescence.

We next explored the regulatory mechanisms underlying the different aging status of adult and perinatal MSCs. Although MSCs from adult and perinatal tissue retain a similar set of core signature genes among subclusters (Supplementary Fig. 3b), many aging related genes were differentially expressed in each population (Supplementary Fig. 3c). Furthermore, gene set scoring also indicated a higher degree of senescence in MSCs from aging tissues (Fig. 3d), while genes related to pluripotency stem cell signature, DNA repair and proliferation pathways were transcriptionally downregulated in adult MSCs (Supplementary Fig. 3d, Fig. 3d). Finally, we verified the aging-related gene expression differences by quantitative real-time PCR (Supplementary Fig. 3e), which corroborated the downregulation of anti-senescence genes (HMGA2, EZH2, and DNMT1) in adult MSCs. Pluripotent stem cell factors (NANOG, OCT4, and SOX2) have been reported in several studies to be better selection markers for adult[34] and perinatal[35] MSCs. The results obtained from qPCR showed downregulated expression of NANOG, OCT4, and SOX2 in adult MSCs (Supplementary Fig. 3e), indicating that senescence may diminish the multipotent differentiation capacity of MSCs. Collectively, these analyses suggest a possible molecular regulatory mechanism that may contribute to the observed differences in aging levels between MSCs derived from adult and perinatal sources.

To validate our finding at the transcriptome level that adult MSCs exhibit higher levels of senescence than perinatal MSCs, we conducted experiments to examine aging markers, proliferation ability and functional alterations caused by the senescence process. Consistent with this bioinformatic inference, our MSC products from AD and BM exhibited an enlarged morphology and an increased level of senescence-associated beta-galactosidase (SA-β-gal) staining during in vitro culture (Fig. 3e). Furthermore, MSCs from adult tissues were also observed to have increased phosphorylated γH2AX foci via immunofluorescence (Supplementary Fig. 3f); these foci coordinate unresolved genotoxic stress and double-strand breaks. Altogether, these extensive bioinformatic analyses and experimental validations provide evidence that perinatal tissue-derived human MSCs are less senescent than adult MSCs.

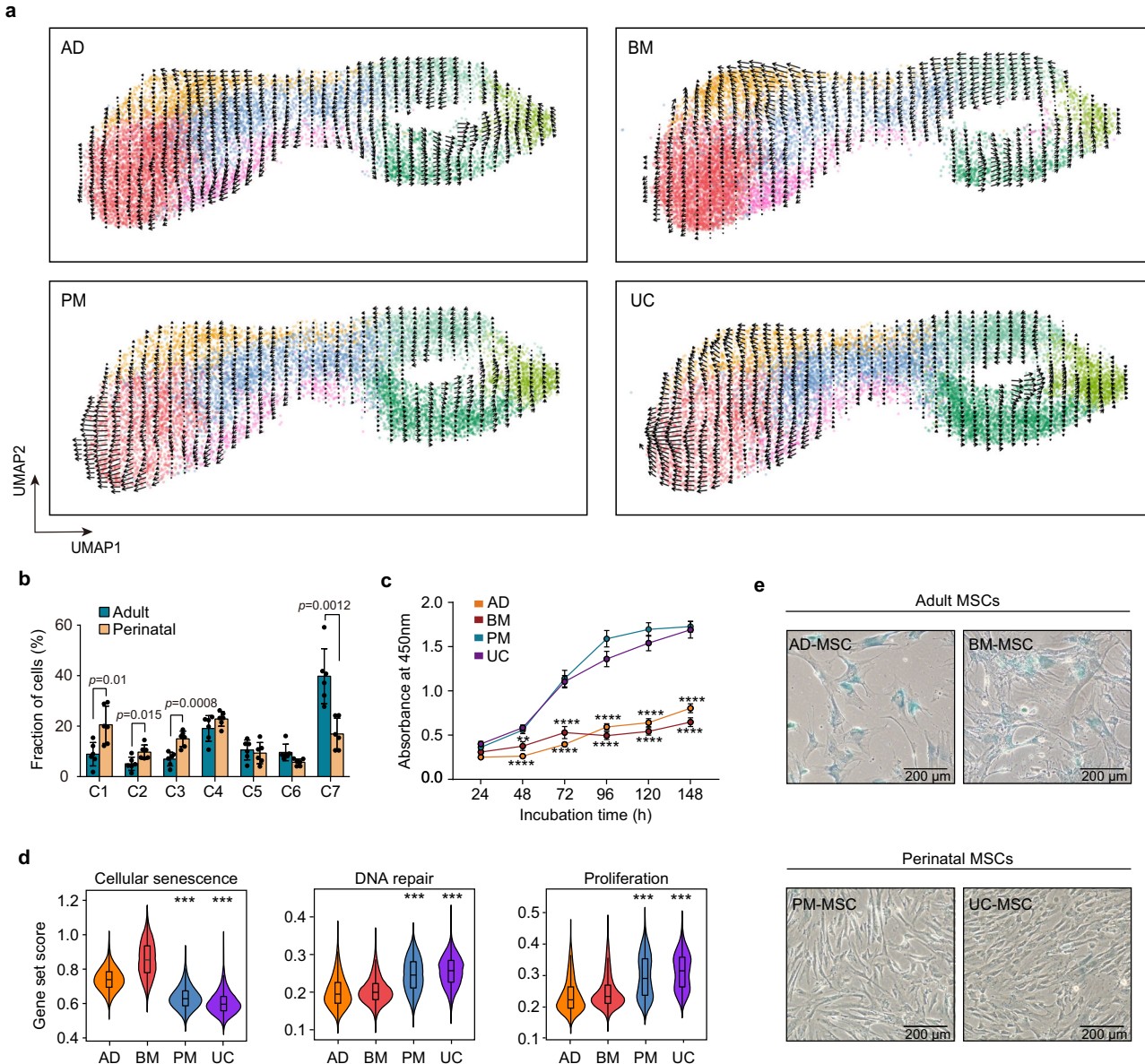

**Fig. 3 | Perinatal MSCs has less senescent cells as compared to adult MSCs.** **a** RNA velocity analysis of MSCs from adult and perinatal tissues. **b** Fractions of subpopulations in six adult versus six perinatal MSC samples (Error bar: means ± SD), *p* values were generated by two-tailed *t*-test with Welch's correction (*n* = 6 independent samples per group). **c** Proliferation of MSCs from 4 tissue origins assessed by CCK-8 assay. Data are means ± SEM and representative of one independent experiment. The *p* values were generated by two-sided one-way ANOVA with Tukey's multiple comparisons test. Asterisks on adult group represent there were statistical differences compared with perinatal group. (**p* < 0.0038; ****p* < 0.0001; *n* = 15 independent samples each group) **d** Violin plots of the cellular senescence score, DNA repair score and proliferation score of cells from each tissue origin. Box plots within each violin plot indicate median values and the 25th to 75th percentiles. Asterisks on specific groups represent significant differences compared with cells derived from AD and BM. The *p* values were generated by two-sided one-way ANOVA with Tukey's multiple comparisons test. (****p* < 2.2 × 10⁻¹⁶; *n* = 45,955 biologically independent cells) **e** SA-β-gal staining in human MSCs derived from adipose tissue, bone marrow, placenta membrane and umbilical cord. Data are representative of one independent experiment. Scale bar, 200 μm. Source data are provided as a Source Data file.

## Cellular senescence status determines the immunosuppressive activity and PD-L1 expression of MSCs

MSCs from different tissue resources vary in their immunomodulatory functions[36], and the underlying cellular and molecular mechanisms remain elusive. Moreover, it is also unknown whether the varying immunosuppressive activities of therapeutic MSC products accounts for the dissonant results of various MSC-based studies. Given that our results show that different tissue-derived MSCs are heterogeneous in terms of their cellular senescence status, we hypothesized that cell senescence may regulate the immunosuppressive function of MSCs. To systematically assess the effect of senescence on the immunomodulatory capacity of MSCs, we summarized multiple factors (Supplementary Data 3) previously reported to perform immunosuppressive functions in MSCs[3] and used them for gene set analysis (see Methods). Consistent with previous reports, we found that MSCs from non-senescent clusters (C1, C2, and C3) showed higher expression of immunosuppressive genes (Fig. 4a). Next, we examined the underlying mechanism by which senescence inhibits the immunosuppressive function of cultured MSCs. *PD-L1*[37] (*CD274*) was selected as an important candidate, as it was significantly upregulated in perinatal MSCs and gradually downregulated during the MSC senescence process (Fig. 4b; Supplementary Fig. 4a). To directly determine the role of

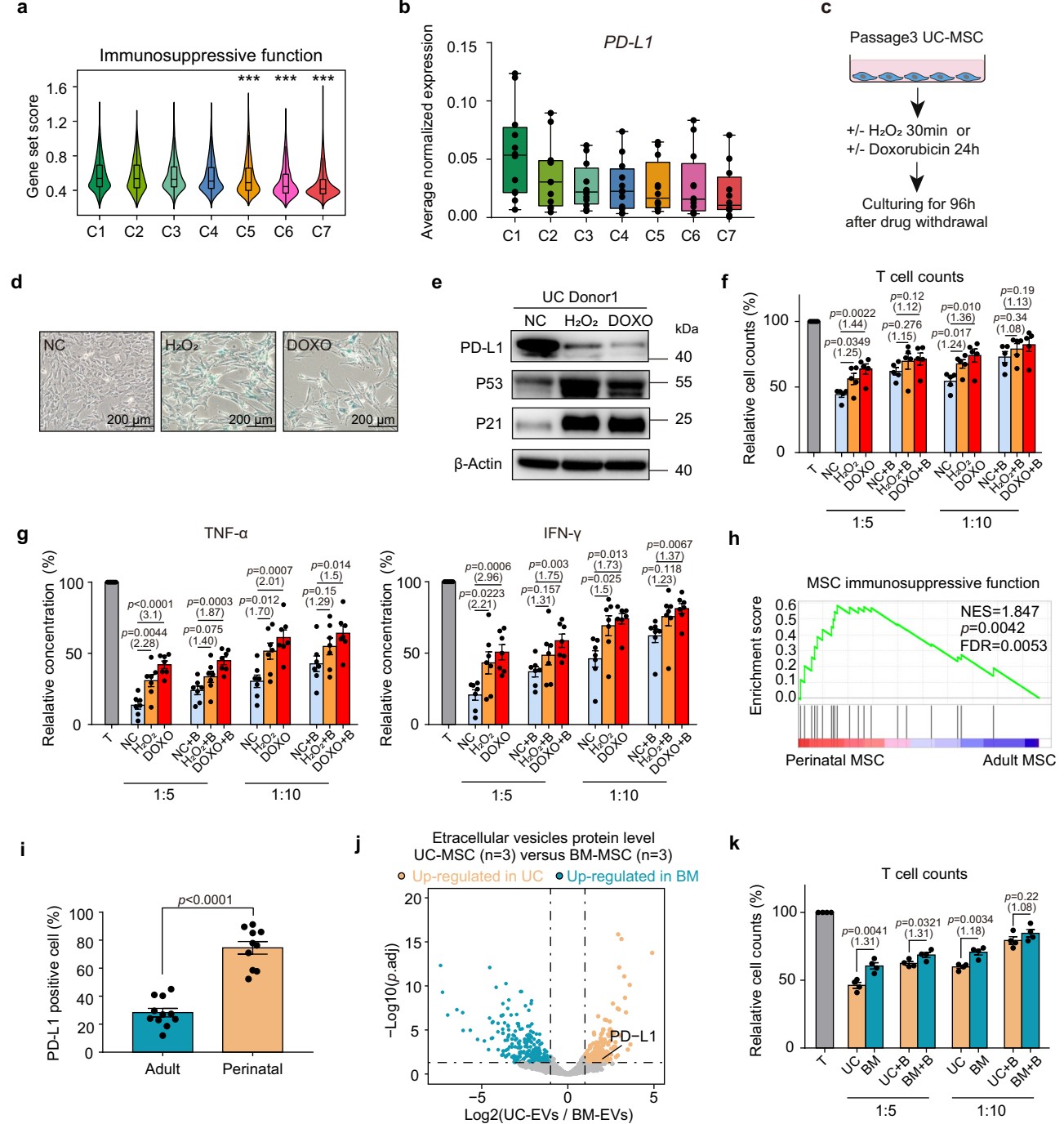

**Fig. 4 | Cellular senescence status determines the PD-L1 expression and immunosuppressive function of human MSCs. a** Violin plots showing the average expression of immunosuppressive functional genes for each cluster. Asterisks on specific groups represent significant differences compared with clusters C1, C2 and C3, *p* values were generated by two-sided one-way ANOVA with Tukey's multiple comparisons test. (***p* < 2.2 × 10⁻¹⁶, *n* = 45,955 biologically independent cells) **b** Box plot showing the average expression of *PD-L1* in each cell cluster of all samples (*n* = 12 independent samples per group). Each dot represents the average normalized expression value of *PD-L1* from Seurat RNA assay. **c** Experimental design. **d** SA-β-gal staining in MSCs. Scale bar, 200 μm. Data represent one independent experiment. **e** Western blot analysis of PD-L1, P53 and P21 protein in MSCs, represent one independent experiment. **f** Bar plots represent cell counts of CD4⁺ T cells cocultured with normal control (NC), H₂O₂-treated or doxorubicin-treated MSCs (*n* = 5 independent experiments). **g** ELISA of TNF-α (left) and IFN-γ (right) levels in the coculture supernatant. Data represent 5 independent experiments (*n* = 7 independent samples each group). **h** GSEA indicating the

significantly enriched immunosuppressive function of perinatal MSCs. The *p*-value was determined by one-sided permutation test, comparing observed NES with a null distribution. FDR was controlled using the Benjamini-Hochberg method. **i** Flow cytometry analysis showing PD-L1 expression among adult (nAD = 6 independent samples; nBM = 5 independent samples) and perinatal MSCs (nPM = 5 independent samples, nUC = 5 independent samples). **j** Volcano plot showing the differentially expressed proteins in UC- and BM-EVs (Log-transformed fold change >1; *p*.adj <0.05). **k** Cell counts of CD4⁺ T cells cocultured with BM- or UC-MSCs (*n* = 4 independent experiments). All box plots indicate median values, and the 25th to 75th percentiles. All bar plots represent the means ± SEM. For **f**, **g** and **k**, CD4⁺ T cells cocultured with MSCs at a 5:1 or 10:1 ratio in the presence of isotype or PD-L1 blocking antibody (+ B), multiples of the mean values between the two groups are marked in parentheses below the *p*-values. For **f**, **g**, **i** and **k**, *p*-values were determined by two-tailed unpaired Student's *t*-test. Source data are provided as a Source Data file.

senescence in regulating MSC PD-L1 expression, we induced cellular senescence in UC-MSCs via the application of doxorubicin or $H_2O_2$ (Fig. 4c) and verified the senescence status of the drug-induced senescence model. (Fig. 4d; Supplementary Fig. 4b). Next, we set to examine PD-L1 expression at the protein level. The flow cytometry results indicated that compared to the normal control (NC) group, DOXO- or $H_2O_2$-treated MSCs had a lower percentage of PD-L1-positive cells (Supplementary Fig. 4c). Similarly, protein bands on Western blots showed that drug-treated senescent MSCs exhibited reduced PD-L1 expression and upregulation of the aging markers P53 and P21 (Fig. 4e; Supplementary Fig. 4d, e). Consistent with this, the downregulation of PD-L1 was also observed in replicative senescent UC MSCs from high culture passages (Supplementary Fig. 4f). Collectively, these results indicate that senescent MSCs downregulate the expression of the immunosuppressive molecule PD-L1 at both the transcriptome and protein levels.

To determine whether the downregulation of PD-L1 protein is functionally important, the immunomodulatory capacity of MSCs was examined by analyzing their ability to inhibit the proliferation of anti-CD3/CD28-activated $CD4^+$ T cells (see methods). As expected, NC MSCs showed a much stronger suppressive effect, as evidenced by lower $CD4^+$ T cell counts (Fig. 4f) and a decreased relative proliferation rate (Supplementary Fig. 4g, h). After PD-L1 blockade with specific antibodies, the inhibition of $CD4^+$ T cell proliferation by NC MSCs was largely reversed, while T cell proliferation in the presence of senescent MSCs was changed only slightly after PD-L1 blockade (Fig. 4f; Supplementary Fig. 4h), indicating relatively low endogenous PD-L1 expression in senescent MSCs. Furthermore, the culture medium from the coculture system was collected to detect the proinflammatory factors (TNF-α and IFN-γ) secreted by $CD4^+$ T cells using ELISA (Fig. 4g). Analysis of spent media revealed that proinflammatory cytokines were secreted at a higher level in the coculture group with relatively senescent MSCs (Fig. 4g). The addition of an anti-PD-L1 blocker stimulated TNF-α and IFN-γ production in all MSC/T cultures, while the enhancement was generally greater with the NC MSCs than with the senescent MSCs (Fig. 4g). These results collectively demonstrated that downregulated PD-L1 expression is one of the molecular mechanisms underlying the impaired immunosuppressive function of senescent MSCs.

Our results showed that adult-derived MSCs had a higher degree of aging, and we next examined whether MSCs gradually lost their immunosuppressive capacity and downregulated PD-L1 expression during the aging process. We first applied GSEA to compare the immunosuppressive function of adult and perinatal MSCs (Fig. 4h), revealing the superior immunosuppressive function of perinatal MSCs. We next verified the enrichment of PD-L1 expression in perinatal MSCs at the protein level. As determined by flow cytometry, the proportion of PD-L1+ MSCs was significantly higher in perinatal tissue-derived MSCs than in adult MSCs (Fig. 4i, Supplementary Fig. 4i, j). The Western blot results also confirmed that there was a significantly higher level of PD-L1 expression in perinatal MSCs, accompanied by downregulation of the senescence markers P53 and P21 (Supplementary Fig. 4k, l). In addition to their cell-surface expression, MSCs were able to secrete PD-L1, which is potentially important in modulating contact-independent mechanisms of immunosuppression[38]. To determine whether cellular senescence attenuates PD-L1 secretion by MSC, we collected extracellular vesicles (EVs) from the culture medium of UC- and BM-MSCs (Supplementary Fig. 4m) and applied mass spectrometry sequencing to investigate their protein expression level. As compared to EVs secreted from the more senescent BM-MSCs, UC-MSC EVs specifically upregulated PD-L1 expression level (Fig. 4j; Supplementary Data 7). We next aimed to determine whether the upregulated PD-L1 in perinatal MSCs was functionally important. We found that compared to UC-MSCs, BM-MSCs from aging tissues lost the capacity to significantly inhibit T cell proliferation when coculturing

(Fig. 4k; Supplementary Fig. 4n, o). In contrast, the addition of anti-PD-L1 Fab to the coculture system with UC-MSCs substantially increased T cell proliferation (Fig. 4k; Supplementary Fig. 4n, o), indicating that the aging-associated decline in MSC immunomodulatory capacity is related to the downregulation of PD-L1. Furthermore, the results of ELISA also suggested that perinatal MSCs showed stronger immunomodulatory ability, as T cells cocultured with UC MSCs secreted less TNF-α and IFN-γ, which was partially rescued by PD-L1 blockade (Supplementary Fig. 4p, q). Collectively, this evidence demonstrates that cellular senescence is a key determinant of the PD-L1-mediated immunosuppressive activity of human MSCs.

## Altered multi-lineage differentiation ability and impaired immunosuppressive capacity were observed in BM-MSCs from aged donors

In order to further demonstrate that the decline in immunomodulatory function caused by the downregulation of PD-L1 is result from senescence, rather than the tissue origin of MSC, we collected BM-MSCs from aged (age>45) and young (age≤20) donors for research (Fig. 5a). We first examined the degree of senescence of BM-MSCs. SA-β-gal staining showed that BM-MSCs from aged donors had a significantly higher proportion of senescent cells when cultured in vitro (Fig. 5b). Consistent with previous reports[39], we found that adipogenesis of aged BM-MSCs was enhanced, while osteogenic differentiation was weakened under in vitro induction (Supplementary Fig. 5a), indicating a skewed differentiation potential. The results from the CCK-8 assay also showed that the proliferation was decreased in aged BM-MSCs (Supplementary Fig. 5b). BM-MSCs characterized for their degree of senescence were then sent for single-cell RNA sequencing, the final 31,907 BM-MSC cells were projected as a query to the clusters identified from the integrated clustering of scRNA-seq data from adult and perinatal MSC samples (Fig. 1b) and divided into five clusters (Fig. 5c). We proved the purity of BM-MSC samples by detecting MSC markers defined by ISCT criteria (Supplementary Fig. 5c). To characterize the functional characteristics of each BM-MSC cluster, we conducted GO analysis based on the DEGs of each cluster (Supplementary Fig. 5d, Supplementary Data 8). Compared to cluster BM1, BM2 and BM3, cells in BM5 and BM6 were projected into senescent cluster C6 and C7 with their upregulated genes enriched in senescence related pathways, including 'protein folding' and 'signal by p53 class mediator', while cells from BM1, BM2 and BM3 highly expressed genes related to proliferation and anti-senescence functions (Supplementary Fig. 5d). Consistently, the gene set scoring analysis showed cells from BM5 and BM6 have upregulated expression of cellular senescence genes (Fig. 5d), accompanied by impaired DNA repair and proliferative function (Supplementary Fig. 5e, f), the downregulation of pluripotency stem cell signature genes was also observed in BM5 and BM6 (Supplementary Fig. 5g). Furthermore, BM-MSCs from senescent (BM5-6) or non-senescent (BM1-3) clusters possessed a similar set of marker genes (Fig. 5e) and activated regulons (Supplementary Fig. 5g, Supplementary Data 9) to those found in adult and perinatal MSC dataset. Our Pearson correlation analysis also suggested that cells from BM1-3 have high correlation value with C1-3, while BM5-6 show more similarity to aging cluster C5-7 (Fig. 5f). We also conducted RNA velocity analysis, which suggested that the transition trajectory of BM-MSCs begins from non-aging clusters (BM1, BM2, and BM3) and ends in senescent clusters BM5 and BM6 (Fig. 5g). Overall, based on another single-cell dataset, we inferred that the cellular senescence process during in vitro expansion may result in the emergence of MSC clusters with different aging statuses.

We next compared the MSCs derived from aged and young BM-MSC at single cell level. We found that the frequency of non-aging clusters BM1 and BM2 was significantly higher in young BM-MSCs, while the frequency of BM6 significantly decreased (Fig. 5h). To further investigate the impact of cellular senescence on the multi-lineage

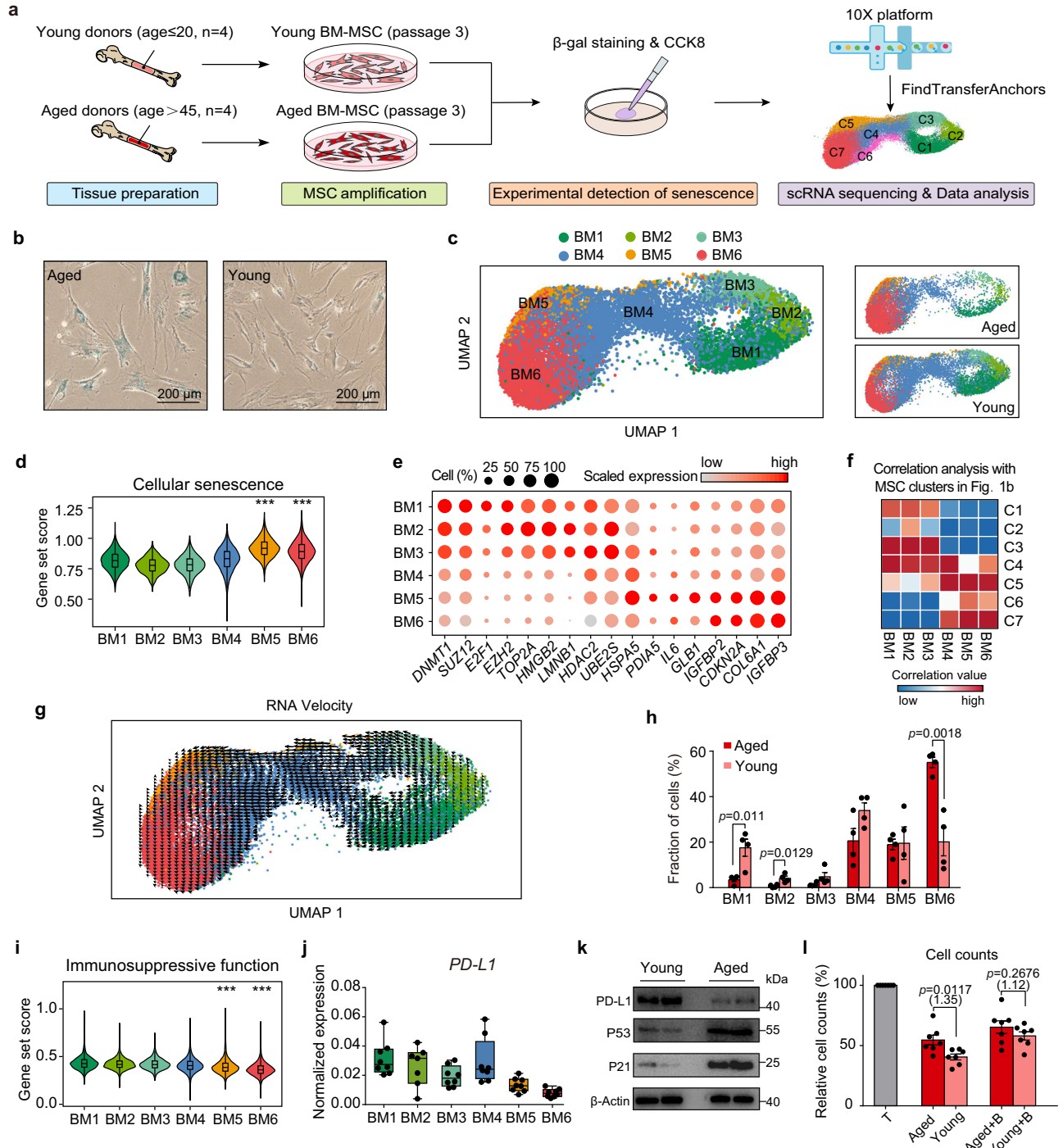

**Fig. 5 | Cellular senescence reduced the immunomodulatory capacity of BM-MSCs derived from aged donors. a** Experimental design. **b** SA-β-gal staining in young and aged BM-MSCs, representative of one independent experiment. Scale bar, 200 μm. **c** UMAP showing six clusters of BM-MSCs. MSCs were projected together by UMAP (left) and displayed separately by age of donors (right). **d** Violin plots showing average expression of cellular senescence genes for each cluster. **e** Dot plots showing the scaled expression of representative senescent-related genes for each cluster. Color represents the scaled expression values from Seurat RNA assay. **f** Correlation matrices showing the Pearson correlation coefficients of the 6 BM-MSC clusters and 7 MSC clusters as shown in Fig. 1b. **g** RNA velocity analysis of BM-MSCs from young and aged donors. **h** Fractions of subpopulations in aged and young MSC samples, *p* values were generated by two-tailed *t*-test with Welch's correction (*n* = 4 independent samples each group). **i** Violin plots showing average expression of immunosuppressive function genes for each cluster. **j** Box plot showing the pseudobulk *PD-L1* expression in each cell cluster (*n* = 8

independent samples per cluster). Each dot represents the average normalized expression value of *PD-L1* from Seurat RNA assay. **k** Western blot analysis of PD-L1, P53 and P21 protein in young (*n* = 2) and aged (*n* = 2) BM-MSCs, represent one independent experiment. **l** Cell counts of CD4⁺ T cells cocultured with young and aged BM-MSCs in the presence of isotype or PD-L1 blocking antibody (+ B). Data are representative of 3 independent experiments (*n* = 7 independent samples each group). The *p* values were generated by two-tailed unpaired Student's *t*-test. Multiples of the mean values between the two groups are marked in parentheses under the *p*-values. All box plots indicate median values, and the 25th to 75th percentiles. All bar plots represent the means ± SEM. For **d** and **i**, *p* values were generated by two-sided one-way ANOVA with Tukey's multiple comparisons test, asterisks on specific group represent there were statistical differences compared with cluster BM1, BM2 and BM3. (***$p < 2.2 \times 10^{-16}$; *n* = 31,907 biologically independent cells). Source data are provided as a Source Data file.

differentiation capacity of MSCs, we calculated the gene set scores for adipogenic genes, osteogenic genes and chondrogenic genes for each BM-MSC cluster (Supplementary Fig. 5i, Supplementary Data 3). The results suggested that the average expression of adipogenesis-related genes increased in the aging clusters (BM5, BM6), while the expression of osteogenesis-related genes decreased. Besides, we found that BM-MSCs derived from aged donors exhibited upregulation of adipogenic genes but decreased expression of osteogenic genes (Supplementary Fig. 5j), which was consistent with the results of the in vitro differentiation experiments (Supplementary Fig. 5a). Although no significant difference in chondrogenic ability was observed during the differentiation induction experiment of young and aged BM-MSCs (Supplementary Fig. 5a), our transcriptome-level analysis suggested that the expression of chondrogenic-related genes was downregulated in aged BM-MSCs. (Supplementary Fig. 5i, j). In addition, we also compared the expression of significant regulatory factors for MSC adipogenic (CEBPB, PPARG), osteogenic (RUNX2), and chondrogenic (SOX9) processes[40]. Although osteogenic and chondrogenic regulators did not exhibit significant differences between aging and non-aging clusters, we found that the expression of transcription factors regulating adipogenic differentiation was significantly upregulated in aging BM-MSC clusters (Supplementary Fig. 5k). While not statistically significant, we observed that CEBPB, which is indispensable for triggering adipogenesis[40], was upregulated in aged samples at the pseudobulk level, whereas the expression of RUNX2, an essential regulator that controls the initial stages of osteoblastogenesis[41], was decreased in aged BM-MSCs (Supplementary Fig. 5l). Together, these findings suggest that senescence may disrupt the balance of MSC multi-lineage differentiation ability, leading to weakened osteogenesis and enhanced adipogenic tendency.

To further explore the effect of aging on the immunosuppressive capacity of MSCs, we next calculated the immunosuppressive function score for each single cell in BM-MSC dataset. Consistent with previous analysis, we found that MSCs from non-senescent clusters (BM1, BM2, BM3) showed higher expression of immunosuppressive genes (Fig. 5i), cells from young BM-MSC samples also exhibit an overall higher immunosuppressive score compared to aged BM-MSCs (Supplementary Fig. 5m). To examine the effect of aging on PD-L1 expression, we detected the PD-L1 gene expression in BM-MSCs dataset, the result showed that PD-L1 was downregulated in senescent clusters (Fig. 5j) or aging MSC samples (Supplementary Fig. 5n). Downregulation of PD-L1 expression at the transcript level of aged BM-MSCs resulted in a decrease in PD-L1 at the protein level, as we detected by flow cytometry (Supplementary Fig. 5o) and western blotting (Fig. 5k). In addition, the levels of PD-L1 protein secreted by senescent BM-MSCs were also reduced, as we detected in culture supernatant using ELISA (Supplementary Fig. 5p). In order to confirm whether the decreased expression of PD-L1 in aging donor MSCs affects the immunosuppressive function, we co-cultured CFSE labeled activated CD4+ T cell with young and aged BM-MSC (Supplementary Fig. 5q). From the results of CFSE proliferation assay and ELISA analysis, we found that compared to young BM-MSCs, BM-MSCs from aging donors lost the capacity to significantly inhibit activated T cells, while this gap was largely reversed after anti-PD-L1 Fab addition (Fig. 5l; Supplementary Fig. 5r–u). Collectively, these results provided further evidence without confounding by tissue-specific functional differences that cellular senescence determines the downregulation of PD-L1 expression on MSCs, which leads to a decrease in their immunomodulatory capacity.

**Integration of single-cell transcriptome profiling and proteomic analysis identifies GATA2 as a key regulator of MSC senescence**
We next applied mass spectrometry sequencing to generate proteomic data for MSCs to validate the completeness of our single-cell RNA-sequencing data and capture senescence-dependent alterations in both mRNA and protein content for the MSCs (Fig. 6a). From bulk proteomic data (BM: $n = 3$; UC: $n = 3$), we quantified 3854 proteins at the cellular level (Fig. 6a). To compare the bulk MSC proteome data with single-cell transcriptomic data, we generated in silico bulk samples from the BM- and UC-MSC scRNA-seq data (see Methods) and performed principal component analysis (PCA) (Fig. 6b). The first principal component showed clustering by data modality, while the second component separated BM-MSCs from UC-MSCs across two data modalities (Fig. 6b). We also conducted correlation analysis to confirm the PCA results (Supplementary Fig. 6a). The results indicated that UC-MSCs exhibited intragroup homogeneity and were distinct from BM-MSCs in terms of their transcriptomic and proteomic profiles. To investigate common or distinct regulation of gene annotation categories at the transcriptomic or proteomic level, we performed a two-dimensional annotation enrichment analysis using Perseus software (Fig. 6c, Supplementary Data 11). We found that most gene categories were modulated the same direction in the transcriptome and proteome, so the positive Spearman's correlation of the annotation enrichment scores was highly significant (Fig. 6c). Proteins and mRNAs highly expressed in perinatal MSCs were enriched in the categories of cell proliferation, metabolic process and stem cell function maintenance, whereas adult MSCs showed a strong increase in cellular senescence and extracellular collagen synthesis in both datasets. Furthermore, we compared the multiomics data between adult MSCs and perinatal MSCs to detect significantly differentially expressed genes and proteins ($p$.adj < 0.05) for each comparison (Fig. 6d, Supplementary Data 10). The observed expression changes in the transcriptome and proteome data indicated comparable results (Fig. 6d, Supplementary Fig. 6b, c). These results suggest a significant correlation between our transcriptome and proteome data for MSCs and provide evidence from a multiomics perspective to demonstrate that adult tissue-derived MSCs have a higher degree of senescence at baseline than perinatal MSCs.

To identify key modulators of MSC senescence, we performed integrated analysis of transcriptomic data and proteomic data. We applied two criteria to narrow down our candidates: candidate TFs should be positively correlated with upregulated non-TF genes at the transcriptomic and proteomic levels in UC-MSCs, and their regulon should be activated in single-cell clusters C1, C2 and C3 (Supplementary Data 12). Overlaying the core TFs (Cor>0.9; $p$ value < 0.05) from the correlation analysis with the top upregulated regulon ($t$-value > 30; $p$ value < 0.05) in C1, C2 and C3 revealed 21 candidate TFs (Fig. 6e, Supplementary Data 12), including the transcription factor GATA2, which was observed to downregulated during the MSC senescent progress (Supplementary Fig. 6d). GATA2 has been reported to promote self-renewal in stem cells[33], but the antiaging function of GATA2 has not been reported in MSCs. To explore whether GATA2 might regulate the expression of anti-aging genes, we screened non-TF genes with a strong positive correlation with GATA2 expression (cor > 0.8; $p$.adj < 0.05) that were highly expressed in the perinatal MSC in both the transcriptome and protein data (Fig. 6f). Interestingly, these genes are responsible for inhibiting aggregation of misfolded proteins (PSMD4, PDIA6, HSPD1), telomere maintenance (CCT7, CCT3), and regulation of cell cycle (TOP1) (Fig. 6f, g), which collectively provided the evidence that GATA2 has the potential to upregulate anti-senescence gene expression in human MSC.

**GATA2 enhances the anti-senescence ability of MSC and restores their immunosuppressive function through the upregulation of PD-L1**
To prove the role of GATA2 in regulating MSC senescence, we used lentivirus vectors (Supplementary Fig. 7a) to overexpress GATA2 in adult BM-MSCs, and successful overexpression was verified by qPCR (Fig. 7a). Compared to MSCs infected with empty vector (EV), GATA2 overexpressed (GATA2-OE) BM-MSCs have weaker β-glucosidase expression at low (Fig. 7b) and high culture passage (Fig. 7c;

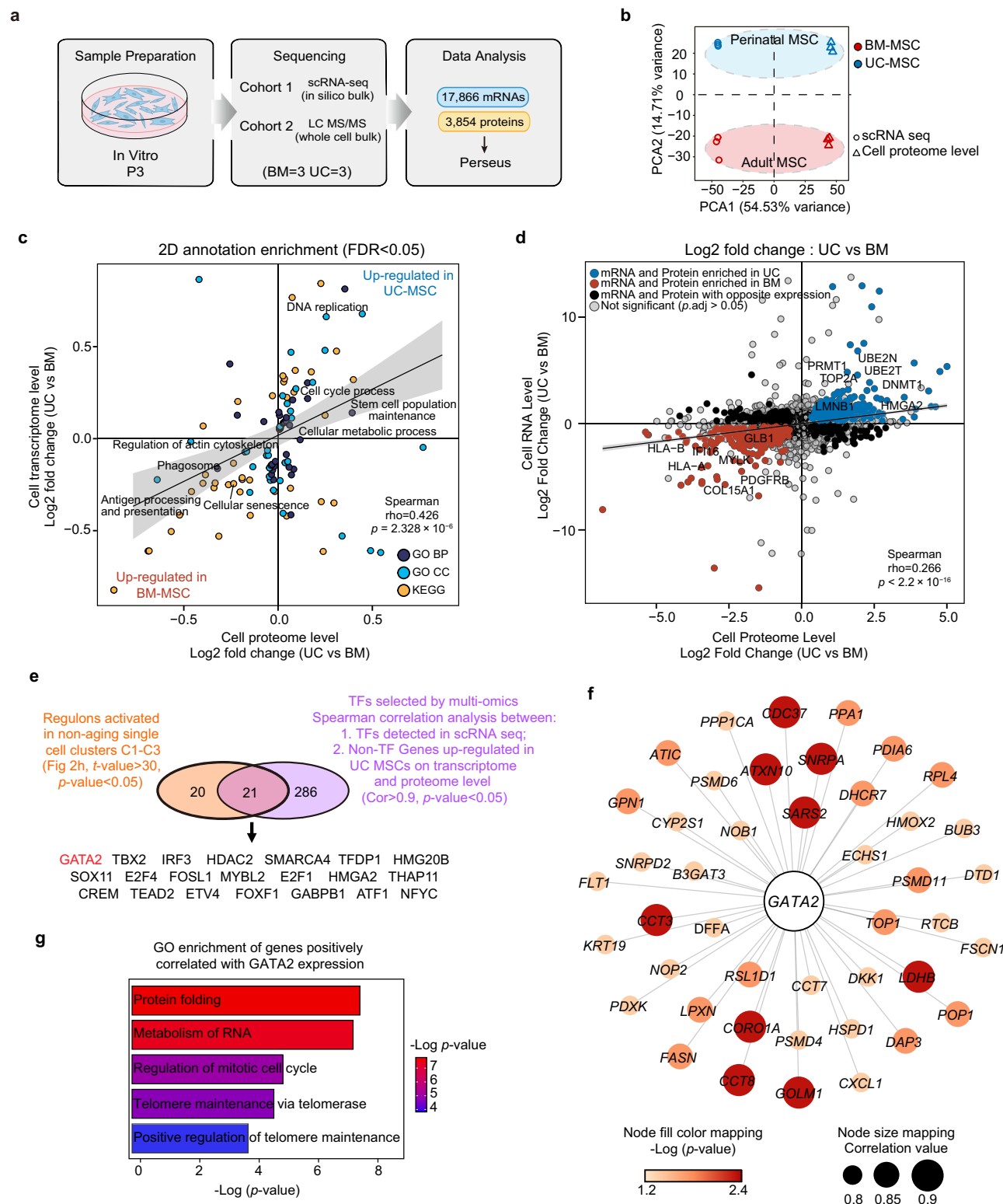

Supplementary Fig. 7b), which indicates the enhanced anti-senescence function of these cells. To further explore whether the overexpression of *GATA2* can improve the immunomodulatory capacity of MSCs alongside their anti-senescence effect, we explored the publicly available GATA2 ChIP-seq data from stromal cells and revealed that GATA2 peaks mapped to the promoter region of *PD-L1* at about ±500 bp around the TSS region (Fig. 7d). Thereafter, we explored all the GATA2 factor binding sites around 500 bp upstream and downstream of the TSS region of the *PD-L1* gene, and designed three promoter sequences,

namely region1 (−534bp to 0 bp), region2 (0 bp to 593 bp) and region3 (−207bp to 134 bp), which all contained GATA2 binding sites inside the sequences (Fig. 7e). The result of dual luciferase assay showed that the relative luciferase activity was upregulated only by co-transfected *GATA2*-plasmid with vector containing promoter region 1 or region 3 (Fig. 7f), which indicates that the binding sites enable GATA2 to drive *PD-L1* expression are likely to located at the overlap sequence of region1 and region3. Therefore, we next analyzed GATA2 enrichment at *PD-L1* promoter by quantitative real-time PCR using primers

**Fig. 6 | Integrated analysis of single-cell transcriptomic and proteomic data identifies GATA2 as a key regulator of MSC senescence. a** Experimental workflow used to analyze the transcriptome and proteome of MSCs. **b** In silico bulk (scRNA-seq) data were merged with proteome data (mass spectrometry) and subjected to principal component analysis. The first principal component shows clustering by data modality. The second principal component separates perinatal and adult MSCs across two data modalities. **c** Scatter plot showing the result of the two-dimensional annotation enrichment analysis based on fold changes in the proteome (x-axis) and transcriptome (y-axis). The significant positive correlation of both datasets was calculated by Spearman's test, *p* values were computed using two-sided algorithm AS 89 (*n* = 114 pathways; Supplementary Data 11). **d** Scatter plot showing the results of differential expression analysis based on fold changes in the proteome (x-axis) and transcriptome (y-axis). The differential

expressed genes or proteins were calculated by two-tailed Wald test and adjusted using Benjamini-Hochberg FDR correction (*n* = 3 independent samples for each group). The significant positive correlation of both datasets was calculated by Spearman's test, *p* values were computed using two-sided asymptotic t approximation (*n* = 3,664 genes; Supplementary Data 10). **e** Venn diagram showing the overlap of TFs between the top activated regulons in clusters C1-C3 (*t*-value > 30; *p* value < 0.05) and TFs significantly correlated with upregulated proteins in UC-MSCs (Cor > 0.9, *p* value < 0.05). **f** Protein−protein interaction (PPI) network representing *GATA2* positively correlated non-TF genes, which were upregulated in UC-MSCs on both transcriptome and proteome level. The color of the nodes represents -Log(*p*-value) and the size of the nodes represents correlation value. **g** Bar plot representing enriched GO pathways of the *GATA2* positively correlated non-TF genes.

designed to detect GATA2 binding fragments at −153 bp (primer 1) and 93 bp (primer 2) of *PD-L1* (Supplementary Data 13). The ChIP-PCR results confirmed the predictive binding of GATA2 at the above two genomic loci (Fig. 7g). In addition, compared with BM-MSCs, young UC-MSCs detected more GATA2 recruitment at the *PD-L1* promoter region (Fig. 7g), which provided a mechanistic explanation for higher *PD-L1* expression in younger MSCs. Altogether, these results indicates that the downregulation of GATA2 during senescence may be an important mechanism mediating the decline of PD-L1.

To confirm this finding, we next examined PD-L1 expression using western blot (Fig. 7h) and flow cytometry (Supplementary Fig. 7c), which validated that the overexpression of *GATA2* upregulates the expression of PD-L1 on protein level. To determine whether the PD-L1 upregulation induced by *GATA2* overexpression enhances immunosuppressive function, we cocultured *GATA2*-OE MSCs with activated CD4$^+$ T cells (Supplementary Fig. 7d) and tested the effect of *GATA2* overexpression on human MSC-mediated T cell suppression by evaluating the capacity to inhibit CFSE-labeled CD4$^+$ T cell proliferation (Fig. 7i; Supplementary Fig. 7e, f) and proinflammatory factors secretion (Supplementary Fig. 7g, h). The *GATA2*-OE MSCs showed a significant improvement in their capacity to suppress CD4$^+$ T cells activation, while the *GATA2*-OE MSC/T cultures treated with anti-PD-L1 Fab showed a pronounced increase in T cell proliferation and TNF-α/IFN-γ production compared with that of control MSC/T-treated cultures, which indicates that upregulated PD-L1 mediates the enhanced immunosuppressive capacity of *GATA2*-OE MSCs. To provide further evidence that GATA2 regulates MSC senescence and PD-L1 expression, we used lentivirus vectors (Supplementary Figure 7i) to knockdown *GATA2* in perinatal UC-MSCs (Fig. 7j). Compared to MSCs infected with shRNA scramble control (CT), UC-MSCs after *GATA2* knockdown (KD) showed senescence phenotypes (Fig. 7k) and downregulations of PD-L1 at protein level (Fig. 7l; Supplementary Fig. 7j). As expected, we also proved that the decreased PD-L1 expression mediates the impaired immunosuppressive capacity of *GATA2*-KD MSCs through the MSC and CD4$^+$ T cells co-culturing assay (Fig. 7m, Supplementary Fig. 7k−o). Collectively, these results identify GATA2 as a crucial regulator of MSC senescence and PD-L1 expression that enhances the immunomodulatory capacity of MSCs.

## Discussion

MSCs are promising cellular therapy products with potent immunomodulatory and pro-regenerative activity. However, various tissue origin and nonstandardized isolation and culture techniques have created obstacles to MSC product preparation and quality management[15,36,42], which has made MSCs the focus of biological studies directed not only at elucidating their intrinsic heterogeneity and unique properties but also developing more homogeneous and effective cellular products for clinical therapies. An increasing number of reports have demonstrated that many surface marker proteins are differentially expressed in MSC populations, as cultured MSCs meeting the ISCT criteria for classic phenotypes can be sorted into different

functional subsets[43]. Recent single-cell transcriptomic studies have investigated the heterogeneity of MSC products, but the results of these studies were limited by the small number of sequenced cells[44], insufficient data analysis[45], single tissue origin[45,46] and an inappropriate data integration method that may fail to correct the batch effect of single-cell sequencing[47].

In this study, we used single-cell transcriptome profiling to systematically characterize in vitro expanded MSC cellular therapy products derived from AD, BM, PM and UC. Our single-cell atlas indicated that cellular senescence is the main cause of MSC heterogeneity at the transcriptome level. Although the differences caused by cell aging inferred through single-cell analysis in this study have been extensively validated by experiments, we cannot exclude the possibility that the cell cycle effect may contribute to cell clustering, as our analysis result characterizes all transcriptomic heterogeneity of the cells. In many single-cell transcriptomic data analyses, it is common to select only cells in a specific cell cycle stage or to use specific algorithms to remove cell cycle effect on cell clustering. However, in some cases, it may be more appropriate to retain the influence of the cell cycle on clustering. In this study, we chose to retain the influence of the cell cycle on clustering for the following reasons: Firstly, after we removed the effect of the cell cycle, there were no significant changes in cell clustering, and the results of the cell cycle regression did not affect our major conclusions of this study. Secondly, cellular senescence is defined as a state of permanent cell cycle arrest, the changes in cell cycle during cellular senescence are also an important characteristic. Thirdly, cell cycle genes may be involved in the proliferation and self-renewal of stem cells. Therefore, the clustering generated by cell cycle genes in stem cells may have biological significance[48]. Additionally, the proliferation capacity is an important factor for measuring MSC function, and retaining the influence of the cell cycle on clustering may better reflect the functional state of MSCs. Many studies using single cell technology on stem cell research retain the influence of the cell cycle on clustering[49,50], which is in line with common practice and may provide more comprehensive analysis results. The results of our analysis indicated that clusters of non-senescent MSCs (C1, C2, and C3) are present in MSC colonies from all tissue origins; these cells highly expressed immunomodulatory signatures and cell cycle regulatory genes. We found that unresolved genomic damage and the unfolded protein response are the two major causes of MSC senescence during extensive in vitro expansion. The accumulation of damage and stress at the genomic or proteomic level causes senescence in three different MSC clusters (C5, C6, and C7) with distinct biological phenotypes and secretory profiles. We speculate that the variation in the proportions of senescent cells caused by different culture passages or differences among individual donors may result in inconsistent product effects. We also identified thousands of differentially expressed genes underlying cellular senescence and described the underlying transcription regulatory networks that drive the dynamics between subclusters. Consistent with previous reports, we found that HMGA2-PI3K-AKT signaling[25,30] and changes in epigenomic modification genes[22,51]

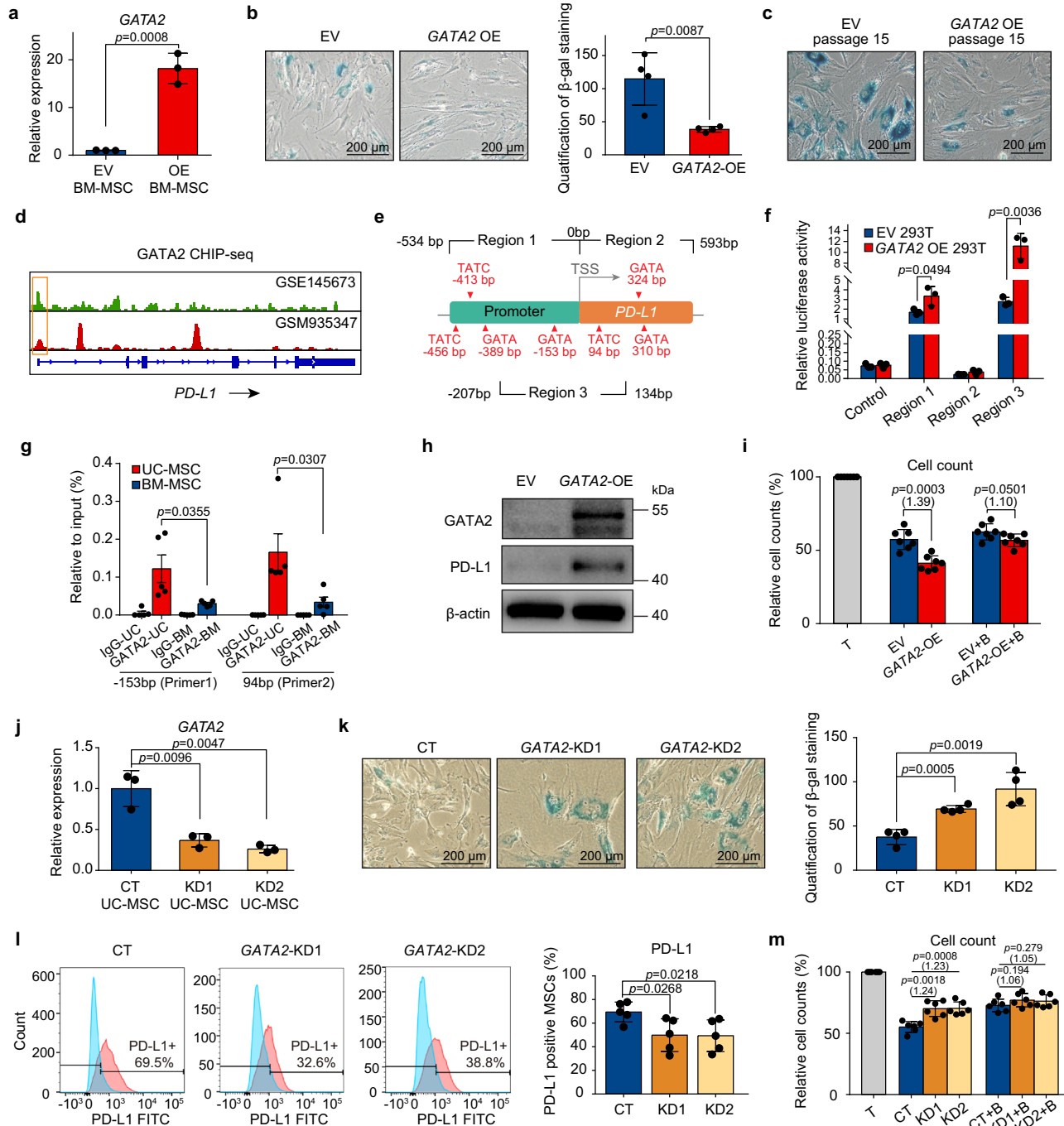

**Fig. 7 | GATA2 enhance the anti-senescence ability of MSC and restore their immunosuppressive function. a** qPCR analysis of *GATA2* expression in BM-MSCs upon lentivirus-mediated gene overexpression (*n* = 3 independent experiments). **b** (left) SA-β-gal staining in BM-MSCs, represent one independent experiment. Scale bar, 200 μm. (right) Quantification of SA-β-gal-positive MSCs, represent three independent experiments (*n* = 4 independent samples per group). **c** SA-β-gal staining in EV or *GATA2*-OE BM-MSCs cultured to passage 15. Scale bar, 200 μm. Data represent one independent experiment. **d** Representative GATA2 occupancy plots of *PD-L1* genes from the ChIP-seq datasets. **e** Experimental design. **f** Bar plot showing the luciferase activity of 293 T cells co-transfected with dual-luciferase reporter, as well as EV or *GATA2* (*n* = 3 independent experiments). **g** qPCR results were used to quantify enrichment of GATA2 at the *PD-L1* promoter using ChIP-assay, pooled from three independent experiments. (*n* = 5 independent samples each group). **h** Western blot analysis of GATA2 and PD-L1 protein in EV and *GATA2*-OE BM-MSCs, represent one independent experiment. **i** Cell counts of CD4⁺ T cells cocultured with EV or *GATA2*-OE MSCs (*n* = 7 independent samples each group).

**j** qPCR analysis of *GATA2* expression in UC-MSCs after transfection with scramble- or *GATA2*-shRNA. (*n* = 3 independent experiments). **k** (left) SA-β-gal staining in UC-MSCs, representative of one independent experiment. Scale bar, 200 μm. (right) Quantification of SA-β-gal-positive MSCs are shown in the bar plot, represent three independent experiments (*n* = 4 independent samples each group). **l** (left) Representative flow cytometry plots. (right) Bar plot showing PD-L1 expression on CT or *GATA2*-KD UC-MSCs, pooled from three independent experiments (*n* = 5 independent samples each group). **m** Cell counts of CD4⁺ T cells cocultured with CT or *GATA2*-KD MSCs (*n* = 6 independent samples each group). For **g**, bar plots represent the means ± SEM. All other bar plots represent the means ± SD. For **i** and **m**, data represent three independent experiments, T cells were co-cultured with MSCs in the presence of isotype or PD-L1 blocking antibody (+ B), multiples of the mean values between the two groups are marked in parentheses. All *p*-values were generated by two-tailed unpaired Student's *t*-test. Source data are provided as a Source Data file.

(*DNMT1*, *HDACs*, *EZH2*, *SUZ12*) may play significant roles in regulating MSC senescence.

MSC products derived from aging donors or tissues have impaired immunosuppressive function[52]. Our computational integration of single-cell transcriptomics and mass spectrometry-based proteomics indicates that, compared to MSCs from perinatal tissue, adult tissue-derived MSCs are more prone to undergo senescence during in vitro expansion and show aging-associated increases in proinflammatory factor levels and the proportion of senescent cells. Senescent MSCs are reported to have impaired immunosuppressive function[11,12], which can explain the observation that adult MSCs exhibit decreased efficacy in immunotherapy. However, the specific molecular changes accounting for the attenuated immunosuppression resulting from senescence have yet to be explored. In this study, we found that the immunosuppressive molecule PD-L1 (CD274) showed higher expression in perinatal-derived MSCs than in adult MSCs. In addition, the downregulation of PD-L1 is also observed during induced senescence in UC-MSCs, which may explain the downregulation of PD-L1 in MSCs from aging tissues. Subsequently, we confirmed that increased PD-L1 levels in non-senescent or perinatal MSCs were functionally important for the suppression of CD4+ T cell proliferation, which suggests that decreased PD-L1 expression may be one of the important molecular mechanisms responsible for the reduced immunomodulatory capacity of senescent MSCs. This finding supports the possible application of PD-L1 in the treatment of inflammatory diseases.

Stemness of MSCs has always been a controversial issue[53], studies based on single-cell clonal assays have speculated that not all cells possess stem cell properties in MSCs during in vitro culture[54]. In this study, we have also investigated the effect of cell aging on MSC pluripotency gene expression and multi-lineage differentiation capacity. Research has shown that aging can have a negative impact on the pluripotency and self-renewal capacity of stem cells[9]. Our findings support this notion, as we observed that non-aging MSC clusters exhibit higher expression of pluripotency stem cell signature genes, and young MSCs from perinatal tissues highly expressed stem cell pluripotency markers, which suggest that the heterogeneity in pluripotency gene expression among MSC populations could be related to differences in the senescence states of individual cells. It is worth noting that although pluripotency stem cell signature genes are widely used to demonstrate the self-renewal ability and multipotent differentiation capacity of MSCs[21,34,35], whether these genes can fully stand for MSC stem cell capacity in the aging process is still controversial[55] and needs to be proved in future work. In addition, our experiments provided additional evidence that senescence impairs the stem cell proliferation capacity and alters tri-lineage differentiation ability. Although the consensus is that a stronger differentiation potential represents better differentiation capacity of stem cells, human MSCs exhibited disrupted balance between differentiations to the osteogenic versus adipogenic lineages during the aging process, and the direction of this shift is still controversial[10]. In this work, we found that senescence decreases osteogenesis and increases adipogenesis of MSCs from aged donors, which is consistent with several previous reports[56] and indicate that aging may bias the multipotent differentiation capacity of MSCs towards adipogenesis. Based on our dataset, we also identified thousands of differentially expressed genes underlying stem cell senescence, and we described the underlying transcription regulatory networks that drive the dynamics between subclusters. The putative senescence-associated transcription factors and the underlying regulatory molecular mechanisms outlined here warrant further investigation. Our combined multiomics analysis of the single-cell transcriptome and multitissue protein profiles helped us detect key regulators of MSC senescence in a more precise way. This information about transcription factors provides new opportunities for reprogramming cells to suit a specific need with consistent therapeutic outcomes or improved product preparation efficiency. The transcription factor GATA2 was proven to be involved in the regulation of stem cell senescence. Overexpression and knockdown experiments of *GATA2* in MSCs prove its important role in inhibiting senescence and restoring *PD-L1* expression. *GATA2* may be used as a potential biomarker at the RNA level to predict the senescence and functional status of MSC products or generate MSC products in cell lines with stronger anti-senescence ability.

## Methods

### Study approval

In this study, MSCs from BM and AD samples were collected from healthy volunteer donors, while the UC and PM samples were from tissues following eutocia or normal caesarean section. All the donors or their guardians provided written informed consent for sample collection and data analysis, and we have obtained consent from donors to publish identifying information, including the biological sex and age information. All procedures in this study were approved and monitored by the Ethics Committee of the Institute of Hematology, Blood Diseases Hospital, Peking Union Medical College and Chinese Academy of Medical Sciences (PUMC/CAMS; approval number: SC2019002-EC-1).

### Tissue collection and isolation of MSCs

In this study, MSCs were isolated from 4 different sources: bone marrow, adipose tissue, umbilical cord and the placental chorionic membrane. The age range, sex and other basic information of donors is provided in Supplementary Data 1.

### Bone marrow MSCs

BM-MSCs were obtained from three donors by iliac crest bone marrow aspiration. Approximately 5–10 mL of BM cells were collected into vacuum blood collection tube which with heparin sodium (ST750SH, INSEPACK). Mononuclear cells containing BM MSCs were isolated by density gradient centrifugation at $650 \times g$ for 15 min via centrifugation (Thermo Scientific Sorvall ST 16 Centrifuge) at room temperature using Ficoll–Hypaque (1.077 g/mL) (LTS1077, TIAN JIN HAO YANG BIOLOGICAL MANUFACTURE). Cells from whitish buffy coat were aspirated and washed with PBS twice.

### Adipose tissue MSCs

Adipose tissues were obtained from three donors undergoing liposuction surgery and washed by PBS (SH30256.01, HyClone) solution with 100 U of penicillin/streptomycin (10378016, GIBCO). The tissues were then cut into small fragments (approximately 1 mm2) and digested in 0.1% collagenase (17101015, GIBCO) with gently agitation for 45 min in 37 °C water bath. The digestion was stopped by DMEM (11995081, GIBCO) medium diluting and undigested tissue was removed by filtration. AD-MSCs from the supernatant were collected by centrifugation at $700 \times g$ for 20 min via centrifugation (Thermo Scientific Sorvall ST 16 Centrifuge) at room temperature.

### Umbilical cord MSCs

Umbilical cords were obtained from neonatal tissue after the delivery and fully rinsed by PBS solution with 100 U of penicillin/streptomycin. The tissues were then cut into small fragments (approximately 1 mm$^2$) and digested in 0.1% collagenase with gently agitation for 30 min in 37 °C water bath. The tissue suspension was transferred into the centrifuge tube and centrifuged at $650 \times g$ for 30 min to discard the supernatant. The remaining tissues were resuspended with 0.125% trypsin, digested with gently agitation for 30 min in 37 °C water bath. The digestion was stopped by adding fetal bovine serum (FBS, 16000-044, GIBCO) and undigested tissues was removed by filtration. UC-MSCs were collected by centrifugation at $700 \times g$ for 20 min at room temperature. Cell centrifugation is performed using Thermo Scientific Sorvall ST 16 Centrifuge.

## Placental chorionic membrane MSCs

Placentas were obtained from neonatal tissue after the delivery and fully rinsed by PBS solution with 100 U of penicillin/streptomycin. The placental chorionic membrane was carefully separated and cut into small fragments (approximately 1 mm²), which were later digested in 0.1% collagenase with gently agitation for 30 min in 37 °C water bath. The tissue suspension was transferred into the centrifuge tube and centrifuged at $650 \times g$ for 30 min to discard the supernatant. The remaining tissues were resuspended with 0.125% trypsin, digested with gently agitation for 30 min in 37 °C water bath. The digestion was stopped by adding FBS and undigested tissue was removed by filtration. PM-MSCs were collected by centrifugation at $650 \times g$ for 20 min at room temperature. Cell centrifugation is performed using Thermo Scientific Sorvall ST 16 Centrifuge.

## Cell culture and expansion

The collected MSCs from different tissue origins were separately seeded into 75 cm² cell culture flask at the density of $1 \times 10^6/cm^2$ with DMEM/F12 (11320033, GIBCO) medium plus 10% FBS, 2 mM L-glutamine and 1% penicillin/streptomycin solution. Cells were cultivated at 37 °C in a saturated humidified atmosphere containing 5% $CO_2$ in a Forma Steri-Cycle $CO_2$ Incubator (Thermo Fisher Scientific, Inc.). After 72 h, the non-adherent cells were removed by washing with PBS solution, after then the medium was changed every three days. The MSCs cultured to the density of 80% following treatment with 0.05% trypsin (27250018, GIBCO) and 0.02% EDTA (AM9912, Invitrogen) for 3 min at 37 ˚C. The cells were washed by centrifugation at $350 \times g$ for 5 min, then replanting to a new T75 cell culture flask at a lower density. MSCs were cultured to the third passage (P3) and prepared for the next step experiment.

## In vitro differentiation

During the third passage (P3), BM-MSC, AD-MSC, UC-MSC and PM-MSC samples were placed in six-well plates at the density of $5 \times 10^4$ cells/well for trilineage differentiation.

Osteogenic differentiation of cells was induced by feeding them with DMEM-LG osteogenic induction medium consisting of 10% FBS, 100 U of penicillin/streptomycin, 100 nM dexamethasone, 10 mM-glycerophosphate (G9422, Sigma-Aldrich), 1 mM sodium pyruvate, 2 mM L-glutamine and 50 µg/mL ascorbate (A4034, Sigma-Aldrich), the media was changed every three day. The osteogenic differentiation was confirmed by positive staining of the extracellular calcium matrix using 2% Alizarin Red S (A5533, Sigma-Aldrich) staining after fixing the cells by 4% polyformaldehyde (P6148, Sigma-Aldrich) at room temperature.

Adipogenic differentiation of cells was induced by feeding them with DMEM-LG (12320032, GIBCO) adipogenic induction medium consisting of 10% FBS, 100U of penicillin/streptomycin, 100 nM dexamethasone (D4902, Sigma-Aldrich), 1 mM sodium pyruvate (11360070, GIBCO), 2 mM L-glutamine (25030081, GIBCO), 0.5 mM indomethacin (I7378, Sigma-Aldrich), 0.5 mM isobutylmethy-lxanthine (I5879, Sigma-Aldrich) and 10 µM insulin (12585014, GIBCO), The media was changed every two day. The adipogenic differentiation was confirmed by the deposition of lipid droplets in the cytoplasm using 0.5% Oil Red O staining (G1260, Solarbio) after fixing the tissues by 4% paraformaldehyde (158127, Sigma-Aldrich) at room temperature.

Chondrogenic differentiation of cells was induced by feeding them with DMEM-HG (11965092, GIBCO) chondrogenic induction medium consisting of 100 nM dexamethasone, 100 µg/mL sodium pyruvate, 50 µg/mL ascorbate, 40 µg/mL proline (P5607, Sigma-Aldrich), 10 ng/mL TGF-β1 (100-21, Peprotech), 1:100 ITS (41400045, GIBCO), 12.5 µg/mL BSA (A8020, Solarbio) and 53.5 µg/mL linoleic acid (L8134, Sigma-Aldrich), the media was changed every two day. To confirm chondrogenic differentiation, cells were stained with Alcian Blue 8GX (A5268, Sigma-Aldrich) staining solution to identify proteoglycans.

## Induction of the senescent MSC model

MSCs were cultured to the third passage (P3) at 70% cell density, then MSCs were treated with DMEM with $H_2O_2$ (300 µM for 40 min) or doxorubicin (1 µM for 24 h) (D8740, Solarbio Life Science), and an equal volume of PBS solution was added to the control group. After drug withdrawal, MSCs were cultured in a 37 °C incubator containing 5% $CO_2$ for 96 h and then used for the next experiment.

## Cell proliferation analysis

MSCs from P3 were planting into 96-well plates at a density of $10^3$ cells/well and cultured in 37 °C incubator with 5% $CO_2$. From day1 (first day implanted into 96-well plates) to day6, we added 10 µL CCK-8 (CK04, Dojindo) solution to each well daily and incubated in 37 °C incubator for 2 h, the absorbance at 450 nm was then measured to determine the cell viability using the microplate reader.

## Senescence associated β-galactosidase (SA-β-gal) staining

At indicated times, cultured MSCs were washed with PBS and fixed in staining fixatives for 15 min at room temperature. Fixed cells were stained with fresh SA-β-gal staining solution 37 °C overnight according to manufacturer's protocol (G1580, Solarbio Life Science). Quantifications of SA-b-Gal+ MSCs were performed with Nikon NIS Element software (version 5.21.00).

## Immunofluorescence

MSCs were fixed with 4% formaldehyde in PBS at room temperature (RT) for 15 min. After fixation, cells were treated with 0.25% Triton X-100 in PBS for 10 min at RT. After blocked with 1% BSA for 1 h, cells were incubated with the anti-γ-H2AX antibody (80312 S, Cell signaling technology, 1:100 dilution) with the indicated dilutions at 4 °C overnight, followed by washing in PBS for three times and incubation at RT for 30 h with the Alexa Fluor goat-anti-mouse IgG (H + L) antibody (A11001, Invitrogen, 1:100 dilution). Then cells were stained with DAPI (D1306, Invitrogen, 1:250 dilution) for 5 min. Images were acquired with a Spinning Disk Confocal Microscope (Perkinelmer).

## T cell proliferation assay

Peripheral blood from healthy donors was collected into EDTA anticoagulant tubes. The Ethics Committee of Tianjin Blood Disease Hospital approved the study protocol, and the donors provided written informed consent for sample collection and data analysis. Mononuclear cells were isolated by density gradient centrifugation at $700 \times g$ for 15 min at room temperature using Ficoll solution. CD4⁺ T cells were collected using CD4 magnetic microbeads (130-045-101, Miltenyi Biotec) and then labeled with CFSE (C34554, Invitrogen) according to the manufacturer's protocol. MSCs from P3 or pretreated cells were then digested with trypsin and seeded in 24-well culture plates at a density of $2 \times 10^4$ cells/well (5:1) or $1 \times 10^4$ cells/well (10:1) with culture medium. After 6 h of culturing MSCs, CD4⁺ T cells were added to the 24-well plates at a density of $1 \times 10^5$ cells/well and stimulated with IL-2 (200-02-10UG, PEPROTECH) (10 ng/mL) and anti-CD3/CD28 precoated Dynabeads beads (11132D, Gibco) ($8 \times 10^4$/well). An anti-human PD-L1 blocking antibody (329716, Biolegend, 1:400 dilution, 5 µg/mL) was added to the culture medium in half of the wells, while an equal volume of isotype antibody was added to the other wells as a control. The coculture ended after 72 h of culture in a 37 °C incubator containing 5% $CO_2$, and the supernatant and the cells were collected separately for further experiments. CD4⁺ T cells from the coculture system were analyzed by flow cytometry to quantify the proportion of CFSE low cells.

## Flow cytometry

Cell surface antigen phenotypic analysis of MSCs or T cells was performed with a FACS Canto II flow cytometer (BD Biosciences) using the following antibodies: CD11b-APC (561015, BD Biosciences, 1:100 dilution), CD14-APC (555399, BD Biosciences, 1:100 dilution), CD19-APC (555415, BD Biosciences, 1:100 dilution), CD34-APC (560940, BD Biosciences, 1:100 dilution), CD45-APC (560915, BD Biosciences, 1:100 dilution), HLA-DR-FITC (555560, BD Biosciences, 1:100 dilution), CD73-PerCP/Cyanine5.5 (344013, Biolegend, 1:100 dilution), CD90-APC (559869, BD Biosciences, 1:100 dilution), CD105-PE (560839, BD Biosciences, 1:100 dilution), PD-L1-FITC (393606, Biolegend, 1:50 dilution), CD4-APC (317416, Biolegend, 1:50 dilution), and FITC Isotype (555748, BD Biosciences, 1:100 dilution). DAPI (D1306, Invitrogen, 1:250 dilution) was used to exclude dead cells. The protocols used were those provided by the manufacturers. Data were analyzed using FlowJo software (Tree Star, Ashland, OR, version 10.0.7). The proportion of PD-L1-positive MSCs was calculated by comparing the same MSC sample labeled with PD-L1 FITC and the FITC isotype control, with detection at the same FITC channel voltage.

## Real-time PCR

Total RNA was extracted from cells using Trizol Reagent (15596018, Invitrogen) to detect the expression of genes by real-time PCR. Reverse transcription was performed using 2 μg RNA to synthesize cDNA at 50 °C for 15 min and inactivated at 80 °C for 5 min. The reaction mixture of qPCR contained 5 μL 2×SYBR green quantitative PCR (Takara), 10 μM of each primer, and 1 μL of cDNA from the reverse transcription product. The thermal cycler parameters for PCR amplification were run as follows: 94 °C for 5 min, 40 cycles of PCR [94 °C for 30 s, 60 °C for 60 s. At the end of the amplification reaction, the specificity of the product was detected by melting curve analysis, and the expression abundance of each gene was quantified by using β-actin as the internal reference. The primer sequences are listed in Supplementary Data 13.

## ELISA assay

Secreted cytokines levels in the collected supernatant were measured using the human ELISA Kit in accordance with the manufacturer's instructions: TNF-α (EHC103a.96, NeoBioscience), IFN-γ (EHC102g.96, NeoBioscience). The secreted PD-L1 was measured using Adipogen PD-L1 (human) ELISA Kit as per the manufacturer's instructions (AG-45B-0016-KI01, AdipoGen Life Sciences). For secreted PD-L1 measurement, MSC culture media samples were concentrated approximately 4-fold using 10 K cut-off Amicon Ultra centrifugal filters (UFC801008, Millipore AB). The absorbance was measured at 450 nm using the BioTek Synergy H4 Hybrid Microplate Reader. The concentration of cytokines was determined from the standard curve generated in each experiment.

## Extracellular vesicles isolation

MSCs at passage 3 were washed three times with PBS and switched to extracellular vesicle (EV) isolation and serum-free medium (ME000-N023, ExCell Biotech). MSCs were conditioned for 48 h prior to EVs isolation. The collected culture medium was first centrifuged at 2000 × g for 10 min to remove cell debris, the supernatant was then being spun at 110,000 × g for 70 min via ultracentrifugation (Type 45Ti rotor, Beckman Coulter, USA). The pellet was then washed with 3 mL of PBS and spun again at 110,000 × g for 70 min again. The pellets were resuspended in 400–500 μL PBS, sterilized by filtration through a prerinsed 0.22 μm filter and stored at −80 °C until use. Visualization of EVs was assessed by HT7700 transmission electron microscope (TEM) (HITACHI, Japan).

## Western blotting

MSCs were lysed for SDS−PAGE. After electrophoresis, the denatured protein samples were transferred onto nitrocellulose membranes for incubation with primary antibody. Antibodies against PD-L1 (ab213524, Abcam, 1:1000 dilution), TP53 (2527 S, Cell Signaling Technology, 1:1000 dilution), P21 (ab109520, Abcam, 1:1000 dilution), GATA2 (ab109241, Abcam, 1:1000 dilution) and β-Actin (abs137975, Absin Bioscience Inc, 1:1000 dilution). After incubation with the primary antibodies overnight at 4 °C, the membranes were washed with TBST, followed by another incubation with HRP-conjugated secondary Goat Anti-Mouse IgG (H+L) antibody (115-035-003, AffiniPure, 1:1000 dilution) or Goat Anti-Rabbit IgG (H+L) antibody (111-035-003, AffiniPure, 1:1000 dilution) for 2 h at room temperature. The specific signals from the HRP-ECL reaction were visualized with a ChemiDoc imaging system (Bio-Rad). Relative quantification of protein bands from the Western blot images was performed with ImageJ software (version 1.52p).

## Overexpression and knockdown of *GATA2* in MSCs

The full-length human *GATA2* ORF was cloned into a lentiviral vector, fused with mScarlet and puro for positive selection. Knockdown of *GATA2* was performed using lentiviral plasmids containing shRNA against human *GATA2* or scramble control. Lentiviral packaging was performed according to the manuals of SBI Company, with slight modifications. In brief, 12–15 μg lentiviral vector was cotransfected with 10 μg PAX2 and 10 μg pLP-VSVG into 293 T cells using PEI 40 K (Polyscience). The supernatant containing the virions produced over the next three days was collected, pushed through a 0.45 μm PVDF filter (Millipore), and centrifuged at 50,000 × g for 2 h to precipitate the lentivirus particles. The empty vector (EV) or *GATA2* OE/*GATA2* KD viruses were then resuspended in DMEM, followed by titer measurement and cell transduction. For sequential lentiviral transduction, MSCs were seeded overnight before the virus was added. The individual viral stocks were added to cells with fresh medium containing 4 μg/mL polybrene. The medium was aspirated after 24 h, and the cells were allowed to recover in normal growth medium. Three days later, bulk populations of infected MSCs were selected in growth medium containing 0.75 μg/mL puromycin. After 120 h of selection, the cells were passaged in puromycin-free medium for further experiments.

## Luciferase assays to test for *PD-L1* promoter activity

Human *PD-L1* promoter region1 (−534/0 bp), region2 (0/593 bp) and region3 (−207/+134 bp) was synthesized and subcloned into the promoterless basic pGL4.10[luc2] vector (Promega) to generate pGL4.10-*PD-L1*p-4.0-[luc2]. Promoterless pGL4.10 [luc2] and pGL4.74[hRluc] vectors (Promega) served respectively as negative and internal controls. HEK293T cells were plated on 24-well plates at $1 \times 10^5$ cells/well one day before transient transfection. Co-transfection groups included basic pGL4.10[luc2] vector or pGL4.10-*PD-L1*p-4.0-[luc2] (800 ng/well), empty vector (EV) or *GATA2* overexpression lentiviral vector (800 ng/well), and pGL4.74[hRluc] (50 ng/well). These vectors were cotransfected into 293 T cells using PEI 40 K (Polyscience). After 48 h, the luciferase activities were determined using a dual-luciferase reporter assay kit (RG088S, Beyotime Biotechnology) following the manufacturer's instructions and measured by BioTek Synergy H4 Hybrid Microplate Reader. The firefly luciferase activity was normalized to Renilla luciferase for each well, and each sample was analyzed in duplicate.

## ChIP-PCR

ChIP assay was performed using the SimpleChIP Plus Enzymatic Chromatin IP Kit (#9005, Cell Signaling Technology). Chromatin was fragmented using the Bioruptor Pico sonication device (Diagenode). The fragmented chromatin was immunoprecipitated with either 3 μg antibody against GATA2 (ab109241, Abcam, 1:50 dilution) or nonspecific rabbit IgG as a negative control, at 4 °C for 12–16 h. Two primers were designed to detect GATA2 binding fragments at −153 bp (primer 1) and 93 bp (primer 2) of PD-L1. Primers for *PD-L1* promoter binding site were listed in Supplementary Data 13.

## Single-cell library construction and sequencing

The collection of MSCs from each sample was performed at P3 when the cells reached 80% confluence. Samples were treated with 0.05% trypsin for 3 min at 37 °C, and the digestion was then stopped by washing twice and resuspending in PBS containing 0.04% BSA. Cell viability and counts were assessed by microscopy with trypan blue, and single-cell suspensions with viabilities >85% were used for library preparation.

For MSC derived from adult and perinatal tissue (as show in Fig. 1), each sample was adjusted to an appropriate volume for a target capture of 5000 cells. scRNA-seq libraries were prepared using the Chromium Single Cell 3′ Reagent Kit V2 (10x Genomics; Pleasanton, CA, USA) according to the manufacturer's protocol. The generated indexed complementary DNA (cDNA) libraries of individual cells were run on a NovaSeq 6000 (Illumina), and 150 bp paired-end reads were generated. We constructed 3 batches of single-cell libraries for sequencing due to the sample collection interval (details in Supplementary Data 1). Samples from the same batch were processed for libraries in parallel and pooled for sequencing in separate lanes of the same flowcell at the same time. The BCL format files generated by Illumina sequencer were converted into FASTQ files using bcl2fastq (version 2.1.9).

For BM-MSC derived from young (*n* = 4) and aged donors (*n* = 4), each sample was adjusted to an appropriate volume for a target capture of 8000 cells. scRNA-seq libraries were prepared using the Chromium Single Cell 3′ Reagent Kit V3 (10x Genomics). The indexed libraries of individual cells were then run on a NovaSeq 6000 (Illumina) with 150 bp paired-end reads generated. All eight BM-MSC sample were processed in one independent experiment, and eight samples were processed for libraries in parallel and pooled for sequencing in separate lanes of two flowcells at the same time.

## Pre-processing of scRNA-seq data

The quality of raw sequencing data was evaluated by FastQC. The Cellranger toolkit (10X Genomics) was used to perform sample demultiplexing and mapping reads to the human genome reference data GRCh38. Cellranger (version 2.2.0) was used to analyze raw counts of 12 MSC samples from perinatal and adult tissue, while Cellranger (version 5.0.1) was utilized in analyzing BM-MSC from young and aged donors. Each sample was generated with a gene expression matrix which records the UMI counts per gene associated with each cell barcode. Unless otherwise stated, further analysis (data filtering and normalizing, identification of highly variable genes, dimensionality reduction and clustering, discovery of differentially expressed genes) was performed using R (version 3.6.1) and the Seurat R package[57] (version 3.1.0). Cells with low-quality and very high mitochondrial genome transcripts were removed according to a sample specific cutoff (details in Supplementary Data 1). After filtering out unwanted cells, we used NormalizeData function to normalize the data for scaling the sequencing depth to a total of 10,000 molecules per cell.

## Integrated analysis of single-cell datasets

The Seurat alignment method for data integration can be used to identify shared sources of variation between datasets. To identify shared correlation structures across MSC datasets from different tissues and detect common subcluster types (as shown in Fig. 1. b), a total of 12 MSC datasets were integrated directly with the FindIntegrationAnchors function using the union of the top 2000 variable genes and 30 dimensions from the CCA. The integrated datasets were used for further analysis.

## Dimension reduction and clustering of MSC single cell transcriptome data

In the next step, the default assay of Seurat object was set to 'integrated', and the ScaleData function was used to regress out the number of UMIs and mitochondrial read count of integrated dataset. To reduce dimensionality, the top 2000 variably expressed genes generated previously were summarized by principal component analysis (PCA), and the top 30 PCA components were further used for visualization by UMAP dimensionality reduction using the RunUMAP function with number of neighboring points set to 12 L. For the MSC subpopulation clustering, we used the FindClusters function which identifies clusters of cells by a shared nearest neighbor (SNN) modularity optimization using the Louvain algorithm. The number of principal components is set to 30, for an 'elbow' was observed around PC 30 through the ElbowPlot function, suggesting the majority of true signal is captured in the first 30 PCs. The resolution is set to 0.4, as when the resolution rate is greater than 0.4, the DEGs between the newly found clusters started to overlap, making it difficult to define their functional characteristics Importantly, data from the 'integrated' assay was only used for dimensional reduction (PCA), UMAP visualization and cell clustering. The other analyses (like differential expression analysis and gene set enrichment) were based on the normalized data from the 'RNA' assay of Seurat object.

## Reference mapping

The single-cell transcriptome data from young and aged BM-MSC (Fig. 5a) were projected as a query to the clusters identified from the integrated clustering of scRNA-seq data from adult and perinatal MSC samples (Fig. 1b). To project the query onto the reference data, anchors between two datasets were calculated through the FindTransferAnchors function implemented in Seurat v4.2.0. Seurat v4.2.0 was installed and executed in R version 4.2.1. The cell type of each cell in the query data was predicted based on those anchors through the TransferData function, which using the top 30 principal components from the reference data for the anchor weighting procedure. Only predicted cluster with more than 100 cells were retained. The query data were then projected to the reference data with MapQuery with reference.reduction set to 'pca' and reduction.model set to 'umap', generating the UMAP plot in Fig. 5c.

## Differential expression analysis of gene signatures and Gene Ontology analysis

For differential expressed genes (DEGs) analysis at single cell level, we applied a "pseudobulk" strategy for differential expression testing, which models variation across biological rather than single cell replicates. For each single cell cluster within each sample, we generated the "pseudobulk" cluster by summing UMI counts across all cells within this cluster. The "pseudobulk" clusters of the same type from different tissues or age groups constitute biological replicates. The DEG analysis between different "pseudobulk" replicates was performed using the R package DESeq2[58] (version 1.24.0, Benjamini-Hochberg adjusted *p*-values < 0.05). For the comparison of MSCs from different tissue sources in Fig. 1, we used DEseq2 to further correct the batch effect in differential gene calculation (design = ~ batch + pseudobulk cluster). DEGs were further used to perform Gene Ontology analysis using Metascape (version v3.5.20211218; http://metascape.org/gp/index.html).

## Gene set activity

The activity of gene sets in MSC clusters was calculated using the GSVA package[59] (version 1.36.2). Like our strategy for calculating DEGs, we sum gene expression values in each cluster to 'pseudobulk' and calculated the GSVA score for each pseudobulk sample. We utilized the limma package[60] (version 3.40.6) to perform differential gene set scoring analysis. The parameter of the model.matrix function was set to '~0+pseudobulk_cluster+batch' to further remove the influence of batch effects on limma differential analysis. The mean GSVA scores for all pseudobulk samples per cluster were displayed in the heatmap. The

reference gene sets were collected from the Molecular Signatures Database (details in Supplementary Data 3 and Supplementary Data 4; https://www.gsea-msigdb.org/gsea/msigdb).

### SCENIC analysis

We employed the SCENIC R-package[61] (SCENIC version 1.1.3; Rcis-Target version 1.6.0; AUCell version 1.8.0; with RcisTarget hg19 motif databases) to carry out transcription factor network analysis. Data subsampled by randomly selecting 1000 cells from each tissue was then analyzed using GENIE3 package (version 1.8.0) to infer potential transcription factor targets. The activity of the regulatory network, trained on this subset of cells, was then evaluated on all the cells in the dataset with AUCell (Step 3).

### Pseudotime trajectory inference

Trajectory analysis was performed using Monocle package[62] (version 2.26.0) to infer the developmental trajectory of MSCs. Ordering genes were identified by the differentialGeneTest function, and the top 1250 genes with lower $q$-value were selected as significant and used to order cells. The nonlinear reconstruction algorithm DDRTree method was used to reduce data to two dimensions and visualized through the plot_cell_trajectory function. The orderCells function was then utilized to sort and map the cells along the trajectory.

### PAGA analysis

To assess the global connectivity topology between the MSC cell clusters, we applied Partition-based graph abstraction (PAGA) from Python Scanpy package[63] (version 1.8.2). The weighted edges represent a statistical measure of connectivity between the partitions.

### Analysis of differential pathway or regulon activities

The 'pseudobulk' strategy were also used to assess differential activities of pathway (scored by GSVA) between 'pseudobulk' clusters using limma package. Besides, a generalized linear model[64] (GLM) was constructed to compare the regulon activity between subclusters during the process of MSC development. The result of GLM is further selected by $p$-value and $t$-values were shown by heatmap.

### Scoring of biological processes

Functional scores were calculated as the average normalized expression of the corresponding gene sets, with the full gene list provided in Supplementary Data 3. The immunoregulation-related genes[3], stem cell property-related genes[20,21] and DNA damage-related genes[65] were summarized from the previous literature. Other functional gene sets were collected from the online database (Gene Ontology, KEGG and Reactome).

### RNA velocity

We verified the trajectory and its directionality using the Velocyto program[66], which predicts individual cell transitions from their spliced and unspliced mRNA content. Velocyto.py (version 0.17.17) was used to generate annotated spliced and unspliced reads from the 10X BAM files. The downstream calculation was performed using Velocyto. R package (version 0.6). Genes were filtered if they had an average spliced variant expression of less than 0.5 or an average unspliced variant expression of 0.05 in at least one cluster. Aggregate velocity fields were estimated for each cell using the gene.relative.velocity.estimates function and visualized via UMAP.

### Cell cycle analysis

For cell cycle analysis, we used the gene set of G1/S and G2/M phase specific genes stored in 'cc.genes' of Seurat package. Each cell was assigned a score by CellCycleScoring function based on the expression of phase-specific marker genes.

### Label-free proteomic analysis using LC–MS/MS

The protein samples stored at −80 °C were thawed and treated with lysis buffer (8 M urea, 1% protease inhibitor, 50 mM NH4HCO3) for 5 min at room temperature. After ultrasonic lysis on ice and centrifugation at $14,000 \times g$ for 10 min at 20 °C, the amount of total protein in each MSC and EV sample was determined by bicinchoninic acid assay (Beijing Solarbio Science & Technology Co., Ltd., Beijing, China) using a NanoDrop ND-1000 spectrophotometer (Thermo Fisher Scientific, Inc., Sunnyvale, CA, USA). Protein digestion was carried out by a filter-aided sample preparation technique. Briefly, 50 µg sample was loaded onto a Microcon filter unit YM-30 (Millipore) and washed with 8 M urea, followed by protein alkylation with 50 mM iodoacetamide in 8 M urea in the dark for 20 min. After alkylation, the filters were washed twice with 8 M urea and twice with 40 mM ammonium bicarbonate. Trypsin (Promega, Madison, WI, USA) was added to the samples at a 100:1 protein-to-enzyme ratio and incubated overnight at 37 °C. Later, the same quantity of trypsin was added, followed by incubation for 2 h. The resulting peptides were collected from the filters by centrifugation.

The proteomic analysis was carried out using an EkspertnanoLC 415 (AB Sciex, Framingham, MA, USA) coupled with a Triple TOF 6600 mass spectrometer (AB Sciex, Framingham, MA, USA). The peptides were dissolved in mobile phase A solvent (0.1% formic acid, 2% CAN, 97.9% water) and loaded onto an AB SCIEX trap column (10 × 0.3 mm; C18 packing specification is 5 µm, 120 A) at a speed of 10 µL/min. Phase B solution (97.9% acetonitrile, 2% water, 0.1% formic acid) was used to elute the trap column at a flow rate of 5 µl per minute. The eluted peptide segments were subjected to an analytical column (150 × 0.3 mm; C18 packing specification is 3 µm, 120 A) and electrospraying. The TOF MS accumulation time was set as 0.25 s with a mass range of 300–1500 Da. The charge state was set from +2 to +5. The mass tolerance was less than 50 ppm, and 60 candidate ions were submitted for each cycle. The LC eluent was directed to a nanoflow electrospray source for MS/MS analysis in an information-dependent acquisition mode. The SWATH Variable Window Calculator (software version 1.1) was used to calculate the mass range for each variable window[67].

MS/MS data were searched against the human UniProt database (released Mar 2020) using ProteinPilot software (version 2.0, Applied Biosystems, USA) with trypsin set as the digestion enzyme. The ProteinPilot search results were imported into SWATH software (version 2.0) as a database to quantify the proteomic data collected by the data independent acquisition (DIA) method. Six transitions were selected for each peptide segment, peptide segment confidence was set to 99%, and FDR was set to 1%. The peak extraction window was set to 10 min, and the quality deviation was within 50 ppm. Two endogenous peptides were selected every 10 min for retention time correction, and the output peak area was used as a quantitative value.

### Integrated analysis of scRNA-seq and bulk proteomic data

In silico bulk samples were generated by summing UMI counts across all cells within one single-cell MSC sample. Differential gene expression analysis of in silico bulk samples was performed using the R package DESeq2[58] (version 1.24.0). As for proteomic sequencing data (cell level or EV level), the differentially expressed proteins between UC- and BM-MSCs were also calculated by DESeq2.

The integration of scRNA-seq data and MSC protein data was performed through the following methods: raw counts from the in silico bulk and MSC bulk proteomic data at the cell level were normalized using the voom() function of the limma[60] (version 3.40.6) R package. Next, in silico bulk and protein data were merged on a set of genes present in all datasets and quantile normalized. This merged and quantile normalized expression matrix was then subjected to PCA, as shown in Fig. 6b. The 2D annotation enrichment analysis used to compare the proteome and transcriptome was performed with

Perseus software[68] (version 1.6.14.0). This test is based on a two-dimensional generalization of the nonparametric two-sample test, and only pathways with a false discovery rate less than 0.05 were considered statistically significant.

### GSEA analysis

GSEA was applied to screen pathways enriched in given cell clusters or MSC products from different tissue origin with default parameter. Gene sets used in GSEA were downloaded from Molecular Signatures Database (https://www.gsea-msigdb.org/gsea/msigdb/genesets.jsp) or summarized from previous published researches. The plots were generated using GSEA software (version 4.0.3) from the Broad Institute (https://www.gsea-msigdb.org/gsea/downloads.jsp). The average normalized expression matrix generated by AverageExpression() function of Seurat R package were input to GSEA analysis in Fig. 4h, in which each pseudobulk sc-RNA data were treated as independent sample. Only results with NES > 1 and FDR and $p$ value < 0.05 was considered statistically significant.

### ChIP-sequencing analysis

We reanalyzed the published GATA2 ChIP-seq data for stromal cells from the GEO dataset (GSE145673, GSM935347). The SRA data were transferred to Fastq with fastq-dump (version 2.5.7). The raw reads were processed through Trimgalore (version 0.6.6) to cut the adapters with the default settings. Then, we aligned the high-quality reads to the GRCh38 reference genome using Bowtie2 (version 2.3.5.1) with the default settings. Subsequently, peak calling was performed with MACS2 (version 2.2.7.1) to select the candidate peaks. BigWig files of GATA2 occupancy profiles were generated using deepTools (version 3.5.1). The occupancy plots were generated via Integrative Genomics Viewer (IGV, version 2.4.16).

### Data visualization, statistics and reproducibility

Bar plots, Violin plots and Box plots displayed in this paper were generated using ggplot2 R package (version 3.2.1). All the heatmaps were generated by pheatmap R package (version 1.0.12). The expression values of differentially expressed genes (DEGs) presented in figures for comparison across clusters or groups were all normalized or scaled values obtained from the Seurat RNA assay. DEGs were calculated using DEseq2 to model variation across biological replicates rather than single-cell replicates. Furthermore, to ensure that only significant genes were selected for plotting and that they were not the result of batch effects, we used DESeq2 to correct for batch effects in the DEG analysis (design=~batch+ pseudobulk cluster).

Comparisons between two groups were done using unpaired two-tailed $t$-tests or two-tailed Wilcox rank sum test. One-way ANOVA with Tukey's multiple comparisons tests were used for multiple group comparisons. All experiments were repeated at least three times independently, yielding similar conclusions. All statistical analysis described in this article were performed using R or GraphPad Prism (version 7.00).

### Reporting summary

Further information on research design is available in the Nature Portfolio Reporting Summary linked to this article.

## Data availability

The scRNA-seq data and read counts data of MSCs were deposited at the National Center for Biotechnology Information's Gene Expression Omnibus with accession number: GSE200161. The raw scRNA sequence data of young and aged BM-MSC have been deposited in the Genome Sequence Archive in National Genomics Data Center, China National Center for Bioinformation / Beijing Institute of Genomics, Chinese Academy of Sciences (GSA-Human: HRA003258) that are publicly accessible at (https://ngdc.cncb.ac.cn/gsa-human/browse/ HRA003258). The mass spectrometry proteomics data of MSCs at both the cellular (dataset identifier: PXD033812) and extracellular vesicle (dataset identifier: PXD042977) levels have been deposited to the ProteomeXchange Consortium (http://proteomecentral. proteomexchange.org). The RDS files, including meta data, labeled assays, and reduction map information have been made publicly available via Zenodo (https://zenodo.org/record/8026174) in order to ensure the reproducibility of the scRNA-seq data. The GRCh38 reference genome used to map the single-cell RNA-seq data and ChIP-seq data was downloaded from the 10× genomics website (http://cf. 10xgenomics.com/supp/cell-exp/refdata-cellranger-GRCh38-1.2.0.tar. gz). The publicly available auxiliary input databases for SCENIC analysis were downloaded from the cisTarget resources website (https:// resources.aertslab.org/cistarget/). The MS/MS data were searched against the human UniProt database (https://www.uniprot.org/). All other data are available in the article and its Supplementary files or from the corresponding author upon request. Source data are provided with this paper.

## Code availability

All analyses were performed using publicly available software as described in the Methods section. Codes related to data screening and major analysis are deposited in the GitHub repository[69] (https://github. com/GaoYuchenPUMC/MSC_paper, https://zenodo.org/record/ 8026091).

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

## Acknowledgements

This study was supported by National Key Research and Development Program of China (2021YFA1101603 to Y.Z.), CAMS Innovation Fund for Medical Sciences (CIFMS) (2022-I2M-2-003 to L.Z., 2020-I2M-C&T-B-086 to Y.C., 2021-I2M-1-041 to J.W.), National Natural Science Foundation of China (81900126 to Y.C., 81970121 to L.Z., 82000136 to T.S., 82070125 to H.L., 82170127 to R.Y., 82171042 to X.Z., 82100151, 82270152 to L.Z., 32230055 to J.W., 81870651 to X.Z., 82101933 to W.Z.), National Key Research and Development Program of China (2019YFA0110802), Non-profit Central Research Institute Fund of Chinese Academy of Medical Sciences (2021-RC310-011 to J.W., 2022-RC320-01 to W.Z.), and CIFMS (2021-I2M-1-003, 2021-I2M-1-073), Haihe Laboratory of Cell Ecosystem Innovation Fund (22HHXBSS00022 to L.Z.), the Special Research Fund for Central Universities, Peking Union Medical College (3332021056 to J.W.), the Distinguished Young Scholars of Tianjin (22JCJQJC00070 to J.W.), and Tianjin Key Medical Discipline (Specialty, to X.Z.). The funders had no role in the study design, data collection and analysis, decision to publish or preparation of the manuscript. We appreciate all donors and their families for participating in this study.

## Author contributions

L.Z., J.W. and X.Z. conceived the project and designed experiments; Y.G., Y.C., W.W., H.L. and W.Z. performed experiments and analyzed the data; Y.G. and J.W. performed bioinformatic analyses, analyzed the data and wrote the manuscript; Y.C. and L.Z. revised the article with the great help of R.Y., T.S., X.L., F.X., W.L. and R.F.; X.Z., J.A. and Y.D. conducted proteomic sequencing and related analyses; H.L., Y.C. and Z.H. provided patient-associated resources and/or patient samples for the studies; P.Z. and Y.Z. assisted with the bioinformatic and experimental workflow; T.C. assisted with the review of the manuscript and provided overall direction.; L.Z., J.W. and X.Z. supervised the study and revised the manuscript before submission.

## Competing interests

The authors declare no competing interests.
