## [Peer Review File · Nature Communications]

Multi-omics analysis of human mesenchymal stem cell shows cell aging that alters immunomodulatory activity through the downregulation of PD-L1REVIEWER COMMENTS

Reviewer #1 (Remarks to the Author):

Summary

Using single-cell RNA sequencing (scRNAseq) and bioinformatics analyses of passage 3 human mesenchymal stem/stromal cells (MSCs), the authors proposed to demonstrate that the 2 adult and 2 perinatal sources—adult sources being from adipose tissue (AD) and bone marrow (BM), and perinatal sources being from placental chorionic membrane (PM) and umbilical cord (UC)—can model the senescent process and allow for identification of ‘stemness’ factors, which the authors identified as GATA2 and PD-L1. Perinatal MSCs were less able to differentiate than adult MSCs, which the authors interpreted as having higher stemness.

While the human scRNAseq profiles are very useful and the bioinformatics analyses is quite compelling, the study ignores well-demonstrated tissue-specific functional differences in human MSCs, especially the fact that perinatal MSCs are more immunomodulatory than adult MSCs and that BM MSCs differentiate more readily into 2 of the 3 required lineages than other sources (i.e. see recent review Yen et al, FEBS J 2022), and thus confusing tissue-specific differences as senescence-related changes.

Major

1. Very similar information has been published in Reinisch et al, Blood 2015, and the interpretation in that study was opposite from this study. In the Blood paper, the “lower differentiation capacity” of non-BM MSCs was interpreted as less MSC-like, i.e. less stemness (which is in line with ISCT Minimal Criteria) and not MORE stemness as the authors here strangely concluded. Moreover, in the same Blood paper, several animal models were used to demonstrate in vivo trilineage differentiation capacity + hematopoiesis, which are highly compelling evidence.
2. If the authors wish to demonstrate chronological senescence w/o the confounding issue of tissue-specific functional differences in differentiation and immunomodulation, the authors could have easily used BM and/or AD MSCs from donors of different ages to focus on this, since immunomodulation is well-reported with both these sources and thus can be evaluated in progressively older donor samples. Please see Park et al, Mech Ageing Dev 2005 & Zhou et al, Aging Cell 2008.
3. General consensus in the field is that both chronological as well as replicative MSC senescence decreases osteogenesis and increases adipogenesis, or even decreases all 3 trilineage capacity. So why in this paper is less differentiation capacity considered more youthful/better “stemness”? More functional evidence is needed to overturn the overwhelming volume of publications to the contrary.
4. It is unclear why passage 3 cells are already senescent/b-gal+. There is much information in the literature that human MSCs usually need to be passaged at least 3 or more passages before a more homogenous population—without WBC/non-MSC contamination—could be even seen, much less utilized, i.e. please see flow cytometry marker Table 5 in Mitchell et al, Stem Cells 2005, and also Brooke et al, Br J Hematol 2008 for clinical use.
5. MSC stemness is a quite controversial issue even to this day (please see numerous commentaries/reviews on this in Cell Stem Cell—latest is Soliman et al 2021, Nature series journals), so rigorous proof-of-concept evidence (at least in vitro, if not in vivo) is necessary, i.e. Hong et al, Science 2005.
6. The inclusion of pluripotency factors to stand for MSC stemness is especially controversial, i.e. Lengner et al, Cell Stem Cell 2007 & Piazzolla et al, Nat Commun 2014, and so functional relevance should be demonstrated.
7. Key references relevant to tissue-specific functions, senescence, MSC PD-L1 research, etc. are not cited.

Specific comments:

1. It is unclear by what criteria PD-L1 was selected to be the gene of interest: a) no PD-L1 at the protein level was seen on when in the same cluster (C7), a number of related and also well-published MSC-immunomodulation genes—TGFB1, HGF, and HMOX1—are also increased; b) expression profiles of PD-L1 in all seven clusters (Figure 4b) did not correlate well with their gene set scores for

immunosuppressive function(Figure 4a)

2. PD-L1 at the protein/surface level is constitutively expressed on human BMMSCs, i.e. please see Davies et al (Le Blanc lab), Stem Cells 2017 which focuses specifically on this topic.
3. This same paper also reports that PD-L1 can be secreted by BMMSCs, so this should be assayed.
4. The link between GATA2 and PD-L1 required functional validation, i.e. promoter assay, at the very least.
5. Along the same lines, in vitro overexpression of GATA2 in adult MSCs or knockdown of GATA2 in perinatal MSCs is necessary for the conclusions the authors are trying to make. In vivo evidence is strongly advised whenever stemness is being hypothesized.
6. Reference #16 (Turinetto et al) is not appropriately cited: it does not mention a tendency of aged MSCs to undergo spontaneous differentiation. Rather, the review paper states that generally osteogenesis is compromised while adipogenesis is not affected or increased.
7. BM-3 appears to have significantly lower cell counts—why is this, and does this have any effect on the bioinformatics analyses?
8. Differentiation functional data is not well correlated to the microarray data, i.e. adipogenesis-related genes are upregulated in aged MSCs/C7.
9. The UMAP in Figure S1i is in the opposite direction as all other similar UMAPs.
10. In Table S2, genes in the the list of distinct expression signatures for C7 cluster are duplicated.

Reviewer #2 (Remarks to the Author):

In their manuscript, “Multi-omics analysis of human mesenchymal stem cell reveals cell aging determining their immunomodulatory activity through the downregulation of PD-L1,” Gao et al. use single cell RNA sequencing (scRNA-Seq) along with mass spectrometry proteomics to characterize mesenchymal stromal cells (MSC) from different tissue sources (adipose tissue (AD), the bone marrow (BM), the placental chorionic membrane (PM) and the umbilical cord (UC)). The authors report that MSCs from different tissue sources (and correspondingly different ages) exhibit different degrees of cellular senescence, with MSCs from “younger” tissue sources exhibiting less senescence than MSCs from “older” tissue sources. Moreover, the authors report that these differences correspond to differences in PD-L1 expression by MSC, which in turn affects the immunosuppressive potential of the MSCs. Finally, they report that expression of PD-L1 in this context is mediated through GATA2 activity.

MSCs have shown great potential in a variety of clinical applications, but, as noted by the authors, there are many incongruous molecular, cellular and functional definitions of MSCs across studies, species, protocols, and tissue sources. There is an unmet need for carefully and thoroughly characterizing MSC resources, making the present study potentially significant. The application of cutting-edge technologies, such as scRNA-Seq and mass spec proteomics, to carefully defined MSC specimens from a variety of tissue sources is a scientifically valuable undertaking, and likely to uncover important MSC biology. Moreover, support of these characterizations with functional experiments evaluating PD-L1 expression, immunosuppressive activity, and GATA2-mediated regulation are important components that extend the studies beyond general “profiling” characterizations. Despite the potential impact of the study, there are significant concerns regarding analysis strategies and corresponding results interpretations, particularly with regard to scRNA-Seq analyses. Issues in these analyses makes it somewhat difficult to evaluate the biological explanations and conclusions drawn from these experiments, which in turn formed the basis for many of the functional follow up experiments. Beyond specific analysis issues, although acknowledged and discussed by the authors, an experimental design in which MSC tissue source is (necessarily) confounded with age presents challenges in determining if potential differences in senescence are due to tissue source, age, or both. Specific comments and suggestions are listed below:

MAJOR POINTS

1. Many aspects of the scRNA-Seq experiment and analysis workflow were well-suited to the goals of characterizing MSC from different sources. For example, balanced sample batching, implementation of data integration through Seurat, cell cycle scoring, etc. were appropriately applied. However, there are several stages of the analysis that raise concerns about and/or could improve the ultimate results and biological interpretations - these are summarized below.

Differential gene expression testing: Throughout the study, a variety of inter-cluster and inter-condition differential gene expression analyses are performed. As helpfully reported by the authors, the samples examined were balanced across three batches (reported in Sup Table 1). To account for potential inter-batch variation, along with inter-source variation, Seurat integration was appropriately applied to 'harmonize' the 12 datasets under study. The authors note that "The other analyses (like differential expression analysis and gene set enrichment) were based on the normalized data from the 'RNA' assay of Seurat object." This is the correct approach. However, in the subsequent differential gene expression testing, gene set testing, etc. there are no steps taken to manage potential technical effects across batches. While this may not necessarily affect results (it represents an additional source of variation across replicates), differential expression results would likely be more robust when accounting for this source of variation.

For differential gene expression testing at the single cell level, the authors implement the FindAllMarkers() and FindMarkers() Seurat functions with Wilcoxon Rank-Sum Test or Student's T-test for statistics. First, with an integrated dataset, the FindConservedMarkers() function (multiple contrasts across each dataset), would likely be more robust. Furthermore, all of these functions can be called with different statistical frameworks, including those that implement linear modeling (e.g. DESeq2, MAST, etc.), in which batch (and other) factors can be included in model testing.

Perhaps a better strategy, given that the authors implemented a strong experimental design with true biological replicates ($n = 3$ for each tissue source), they might consider a 'pseudobulk' strategy for differential expression testing that models variation across biological rather than single cell "replicates." As presently applied, while sometimes necessary, single cell differential expression testing has been shown to be prone to high false positive rates (Squair JW, et al. Confronting false discoveries in single-cell differential expression. Nat Commun. 2021 Sep 28;12(1):5692). Given the fully replicated design here, differential gene expression with pseudobulk biological replicates would likely provide more robust results.

An improvement in differential gene expression results would also be expected to improve results in downstream analyses that used these results as input (e.g. Metascape, etc.).

Cell cycle phase assignments and interpretations: Cell cycle scores were assigned per cell using the associated functionality in Seurat. This approach predicts a likely cell cycle phase (output as G1, S, or G2M) for each cell based on expression patterns of associated gene sets. While a useful tool, there are some concerns as to these data are interpreted in the text. For example, the authors note that (line 132), "MSCs from C1 were predicted to be in S phase, MSCs from C2 and C3 were assigned to G2/M phase, and cell cycle arrest occurred in, MSCs from C5 to C7, as nearly all cells were predicted to be in G1 phase." A "G1" assignment does not indicate "cell cycle arrest." Indeed, in our experience, this is a typical observation in scRNA-Seq data for cells that are not actively in the S/G2M proliferating process; in the Seurat classification function, there are no other options (e.g. "G0", etc.), so the G1 label often acts as a "catch all" for non-S, non-G2M. We tend to observe large numbers of "G1" classified cells even in scRNA-Seq datasets of universally proliferating cell lines.

Related observations such as (line 142) "phenotype space between nonproliferative (C5, C6, C7) and proliferative clusters (C1, C2, C3) is relatively weak compared to the connectivity within either proliferative or nonproliferative subpopulations (Fig. 1f)" may not be fully explanatory, as the S/G2M gene expression patterns result in considerable variation across cells that can "drive" the PCA and consequent clustering and UMAP, without necessarily indicating distinct underlying phenotypes of the cells.

By extension, it is somewhat unclear how much of the gene expression differences in the “proliferative” vs “non-proliferative” clusters are driven simply by differences in cell cycle phase versus differences in senescence. While many of the genes highlighted by the authors are associated with senescence, alternatively, these differences in expression could be associated with cell cycle phases. It would be most helpful to have a “ground truth” scRNA-Seq dataset that included definitive senescent cells for direct comparison. These and related observed gene expression differences in “proliferative” vs. “non-proliferative” clusters may overinterpret the biological significance of these patterns, and do not necessarily support definitive statements such as “These results collectively indicate that the attenuated MSC proliferation can be ascribed to cellular senescence” (Line 152), and “Thus, our data at single-cell resolution indicate that the loss of proteostasis plays an important role in driving the cellular senescence of C5.” (Line 227), and “The abundant expression of genes such as EGR1, TP53, CDKN1A, CDKN2A and FOXO3 provided solid evidence that senescence of C6 and C7 is linked to genomic stress (Fig. 2a, b, Supplementary Fig. 2a) (Line 191).”

Gene set scoring by GSVA: GSVA, a widely used tool for gene set scoring and enrichment analysis in “bulk” RNA-Seq studies, was used to assign per cell gene set scores, which in some cases were subsequently used to test for differential set enrichment across samples. Unlike standard “bulk” RNA-Seq data, scRNA-Seq datasets are extremely sparse and have many different characteristics than bulk datasets. Is there any indication that GSVA has been tested and found to be appropriate/robust for single cell RNA-Seq data? There appears to be a github package under development (<https://github.com/guokai8/scGSVA>), but there do not seem to be any peer reviewed indications that this tool performs appropriately for scRNA-Seq data, raising concerns about these results. If the authors wish to use the GSVA tool for scRNA-Seq data, they must demonstrate its robust performance with appropriate controls and ground truth benchmarking, and/or cite such a demonstration.

Trajectory analysis: Trajectory analysis was performed using Monocle v2. These types of analyses can present potential pitfalls for integrated multisample data. However, there are no details provided on how the trajectory analysis was implemented for the integrated datasets. Furthermore, while most of the analysis code is helpfully available via the provided github link, it appears that this repository does not contain the code for the Monocle trajectory analyses, which made it challenging to evaluate this aspect of the study. In particular, Monocle analysis allows for specifying several parameters regarding the expected number of roots/branches, etc. from trajectory data. Without knowing how these parameters were set, the interpretation that “mesenchymal clusters occurred on a onefold trajectory, starting from cells in the active 139 proliferating phase (C1, C2 and C3) and ending with C7 (Fig. 1e)” may not be appropriate.

RNA velocity: RNA velocity analysis can also present potential pitfalls for integrated multisample data, especially when attempting to compare velocity results across different conditions (as in Fig 3). It is not clear how the different samples (and different batches) were managed in RNA velocity analysis which makes it challenging to evaluate the results.

2. The inclusion of functional assays for cell proliferation, immunosuppression by MSCs via PD-L1, and GATA2 regulation of PD-L1 expression are a strength of the paper, following up on hypotheses generated by the scRNA-Seq and proteomics data. However, while significant, the differences in immunosuppressive effects via PD-L1 appear to be somewhat modest – would this difference be sufficient to explain some of the differences observed in vivo?

ADDITIONAL MINOR POINTS

The text notes that genes associated with dissociation-induced artifact were removed to test for such

effects. Were these genes removed for the final analysis? While it's good that they were checked for dissociation gene expression patterns, removing these genes is not appropriate — the genes could also be involved in the MSC biological pathways under study.

The text references "Supplemental Table 10", but this table does not appear in the supplied Excel workbook.

In the Methods section, there are no details provided as to how the number of principal components were selected for scRNA-Seq dimensionality reduction or how the cluster resolution was determined.

Reviewer #3 (Remarks to the Author):

The authors have performed an exhaustive gene and protein expression analysis of mesenchymal stroma cells (MSC). The goal of the single cell based analysis was to describe the cellular phenotype variation in MSC and compare MSC from bone marrow, adipose tissue, placenta and cord blood. The data confirmed that MSC are not homogenous in culture, and that the characteristics of the different sources organs are maintained. Fewer proliferative and stem like cells are present in adult MSC than in perinatal MSC, while more senescence markers and in general G1/2 phase markers are reciprocally present. This is not new, except for the now very thorough analysis showing the shift in senescence and the cell source specific relative proportions of senescence and non-senescent cells. The data also confirm many of the previous findings on factors influencing or accompanying proliferation, senescence, immunosuppressive function or differentiation. It remains unclear whether the comparison is biologically sound as the MSC from different tissue sources may as well represent different cell types, which are compared here. The transcriptomic analysis hints in this direction. The finding that PD-L1 is associated with senescence and immunosuppression was confirmed, but is not new (e.g. non referenced paper by Davies, 2017). New is the finding that GATA2, by regulating for example PD-L1, is important for maintaining a rather non-senescent phenotype and maintaining a better immune suppression in the used biological assays. It would be of interest whether this function can be shown in long-term cultures after multiple passageing of cells in vitro - and not only in peroxide induced senescence. The authors claim that knowing the transcriptomic profiles and factors can be used to influence senescence and thus maintain a homogenous and strong immunosuppressive phenotype in clinical products.

This is a well written manuscript. However, the complex methodological procedure and amount of figures (6 Figures and 61 panels) are not needed to get the information out and should be reduced and focused on the main new findings. The specific points raised are:

- elucidate the point that the MCS from different source organs are intrinsically different and cannot be compared as potentially identical cell types. Can the data be analysed under this aspect? Are 'tissue origin related differences among subpopulations...' also found between populations?
- is it possible that the cell clusters just reflect different cell cycle stages, which could be misinterpreted as senescent vs. proliferative?
- donor age should be reorted (AD and BM MSC)
- reduce complexity of text and figues and focus on main novel findings. For example, most transcriptomic and proteomics data confirm what has been known, but novelty is limited (although the provided cell atlas is valuable).
- it should be discussed whether the detected functional differences are potentially clinically relevant (how predictive are the used suppression assays to predict clinical efficacy)?

Recommend to accept with major revision.

Point by point response to reviewers

We appreciate the encouraging and constructive comments from all reviewers. Following the suggestions by the reviewers, we have performed additional experiments and provided a substantial amount of new information in the revised manuscript. Among the highlights of the new results, we have improved the analysis of our single cell RNA sequencing data by adopting more rigorous strategies, which further enhanced the credibility of our analysis results. Besides, a single-cell transcriptomic analysis of bone marrow mesenchymal stem cells (BM-MSCs) from aged and young donors was complemented and verified by exhaustive functional experiments, which provided further evidence without confounding by tissue-specific functional differences that senescence is a key determinant of the PD-L1-mediated immunosuppressive activity of MSCs. We have also provided new data to reveal that PD-L1 can be detected in extracellular vesicles (EVs) of MSCs, and cellular senescence attenuates the ability of MSCs to secrete PD-L1. Furthermore, we elucidated the mechanism by which GATA2 drives the expression of PD-L1, and by further complemented the GATA2-overexpression and -knockdown system, we provided better functional evidence for GATA2 as a critical regulating factor in restraining senescence and maintaining immunomodulatory capacity of MSCs. Finally, we reported here a clinical trial (NCT04014166) of the use of umbilical cord MSCs (UC-MSCs) in treating refractory immune thrombocytopenia (ITP). The new data are highlighted in a blue font in the revised manuscript, including the legends and labels of figures and supplementary tables. With the new results, we believe we have addressed all of the reviewers' concerns, substantially improved the quality and mechanistic insight of our work, and broadened the impacts of our study.

A brief summary of the major new results is listed below, which are divided into five categories.

A. Adoption of more rigorous strategies to calculate DEGs, single-cell gene set score and pseudotime trajectory.

1. The DEG analysis between different “pseudobulk” replicates was performed using the R package DESeq2 (**Fig. 1h; Supplementary Fig. 1f; Fig. 2b, e; Supplementary Fig. 2e; Supplementary Table 2**).
2. The result of DEGs calculated by “pseudobulk” strategy were used to improve results in downstream GO enrichment analyses (**Fig. 2d; Supplementary Fig. 2a, b**).

3. The activity of gene sets in single cells was calculated using R package AUCell. The ‘pseudobulk’ strategy were also used to assess differential activities of pathway scored per cell by AUCell (**Fig. 1c; Supplementary Table 4**).
4. Pseudotime trajectory analysis of adult and perinatal MSCs was improved using updated new version of Monocle v2.26.0 package (**Supplementary Fig. 1e; Fig. 2f**).

B. Extensive analysis of BM-MSCs derived from young and aged donors provided further evidence that senescence is a key determinant of the PD-L1-mediated immunosuppressive activity of MSCs

1. SA- β -gal staining, trilineage differentiation and cell proliferation assay of BM-MSCs derived from aged donors (**Fig. 5b; Supplementary Fig. 5a, b**).
2. Single cell analysis of young and aged BM-MSCs dataset revealed 6 clusters of different senescent status (**Fig. 5c-f; Supplementary Fig. 5d-g**).
3. RNA velocity analysis indicated aged BM-MSCs have a stronger tendency to progress to senescent state during in vitro expansion and result in a significantly increased frequency of aging cells (**Fig. 5g, h**).
4. Gene set scoring analysis identified reduced immunomodulatory capacity in aging BM-MSCs, the decline in PD-L1 has also been observed during senescent progress (**Fig. 5i, j; Supplementary Fig. 5h-i**).
5. FACS analysis and western blots of PD-L1 expression in BM-MSCs from young and aged donors (**Fig. 5k; Supplementary Fig. 5j**).
6. Cell counts and relative proliferation rate of CD4⁺ T cells cocultured with young and aged BM-MSCs for 72 h in the presence of isotype or PD-L1 blocking antibody (**Fig. 5l; Supplementary Fig. 5l, m**).
7. ELISA analysis of TNF- α and IFN- γ levels in the cell culture supernatant secreted by T cells after 72 h of coculture with young and aged BM-MSCs in the presence of isotype or PD-L1 blocking antibody (**Supplementary Fig. 5o, p**).
8. Average expression level of GATA2 in MSC from adult and perinatal tissue; and the gradually reduced expression of GATA2 during aging process is showed in BM-MSCs derived from young and aged donors (**Supplementary Fig. 6d**)

C. Analysis of soluble PD-L1 in culture medium of in vitro expanded MSCs reveals that PD-L1 secretion was significantly suppressed in senescent MSCs.

1. Analysis of PD-L1 expression level in proteomic data of UC- and BM-MSCs

secreted EVs (**Fig. 4j; Supplementary Fig. 4m**).

2. ELISA analysis of PD-L1 level in the cell culture supernatant secreted by BM-MSCs from young and aged donors (**Supplementary Fig. 5k**)

D. Experimental evidence elucidated the mechanism by which GATA2 drives the expression of PD-L1, and by further complemented the GATA2-overexpression and -knockdown system, we provided better functional evidence for GATA2 to manipulate senescence progress and immunomodulatory capacity of MSCs.

1. Real-time PCR analysis of GATA2 expression in EV or GATA-OE BM-MSCs (**Fig. 7a**).
2. SA- β -gal staining in EV or GATA2-OE BM-MSCs at passage 3 or passage 15 (**Fig. 7b, c; Supplementary Fig. 7b**).
3. Relative luciferase activity of 293 T cells co-transfected with dual-luciferase reporter constructs containing PD-L1 promoter regions, as well as EV or GATA2 (**Fig. 7e, f**)
4. ChIP-qPCR analysis of GATA2 enrichment at PD-L1 promoter region (**Fig. 7g**).
5. Western blot and FACS analysis of PD-L1 protein in EV and GATA2-OE BM-MSCs (**Fig. 7h; Supplementary Fig. 7c**).
6. Cell counts and relative proliferation rate of CD4⁺ T cells cocultured with EV or GATA2-OE BM-MSCs for 72 h in the presence of isotype or PD-L1 blocking antibody (**Fig. 7i; Supplementary Fig. 7d-f**).
7. ELISA analysis of TNF- α and IFN- γ levels in the cell culture supernatant secreted by T cells after 72 h of coculture with EV or GATA2-OE BM-MSCs in the presence of isotype or PD-L1 blocking antibody (**Supplementary Fig. 7g, h**).
8. Real-time qPCR analysis of GATA2 expression in human UC-MSCs after transfection with scramble- or GATA2-shRNA (**Fig. 7j; Supplementary Fig. 7i**).
9. SA- β -gal staining in CT or GATA2-KD UC-MSCs (**Fig. 7k**).
10. Western blot and FACS analysis of PD-L1 protein in CT and GATA2-KD UC-MSCs (**Fig. 7l; Supplementary Fig. 7j**).
11. Cell counts and relative proliferation rate of CD4⁺ T cells cocultured with EV or GATA2-OE BM-MSCs for 72 h in the presence of isotype or PD-L1 blocking antibody (**Fig. 7m; Supplementary Fig. 7k-m**).
12. ELISA analysis of TNF- α and IFN- γ levels in the cell culture supernatant secreted by T cells after 72 h of coculture with EV or GATA2-OE BM-MSCs

in the presence of isotype or PD-L1 blocking antibody (**Supplementary Fig. 7n, o**).

E. A registered clinical trial (NCT04014166) was conducted to use the UC-MSCs in treating refractory ITP. The results of six participants who had no response to splenectomy were showed in the revised manuscript to demonstrated the safety and high efficacy of UC-derived perinatal MSC therapy.

1. Clinical trial design and baseline characteristics of the 6 immune thrombocytopenic purpura patients reported in the clinical trial (**Supplementary Fig. 8a, b**).
2. Study enrollment, UC-MSC administration, participant follow-up and data analysis are shown for the three study groups with escalated therapeutic dose (**Supplementary Fig. 8c**).
3. The longitudinal changes in platelet counts in the participants receiving UC-MSC treatment for 28 weeks (**Supplementary Fig. 8d**).
4. Summary of best responses in efficacy-evaluable patients (**Supplementary Fig. 8e**).

Apart from these, we have complemented in Supplementary Table 1 the biological age information of each MSC sample used for sequencing and functional experiments in this manuscript. According to the suggestion of reviewer #3, we have reduced the complexity of figures by transferring redundant panels to corresponding supplemental figures, which made manuscript more conducive to presenting the main new findings, besides, the complex methodological procedure descriptions were simplified in the result section.

We also present additional results to the reviewers as appendix figures at the end of our response letter (Appendix Fig. 1-6). We prefer not to include these data in the manuscript due to space limitations and the scope of our study, but we will be happy to include them if the reviewers or the editors think this necessary. Unless otherwise noted, all figure callouts below correspond to figures in the revised manuscript.

Reviewers' comments:

Reviewer #1 (Remarks to the Author):

Summary

Using single-cell RNA sequencing (scRNAseq) and bioinformatics analyses of passage 3 human mesenchymal stem/stromal cells (MSCs), the authors proposed to demonstrate that the 2 adult and 2 perinatal sources—adult sources being from adipose tissue (AD) and bone marrow (BM), and perinatal sources being from placental chorionic membrane (PM) and umbilical cord (UC)—can model the senescent process and allow for identification of ‘stemness’ factors, which the authors identified as GATA2 and PD-L1. Perinatal MSCs were less able to differentiate than adult MSCs, which the authors interpreted as having higher stemness.

While the human scRNAseq profiles are very useful and the bioinformatics analyses is quite compelling, the study ignores well-demonstrated tissue-specific functional differences in human MSCs, especially the fact that perinatal MSCs are more immunomodulatory than adult MSCs and that BM MSCs differentiate more readily into 2 of the 3 required lineages than other sources (i.e. see recent review Yen et al, FEBS J 2022), and thus confusing tissue-specific differences as senescence-related changes.

We thank the reviewer for the insightful comments.

Major

1. Very similar information has been published in Reinisch et al, Blood 2015, and the interpretation in that study was opposite from this study. In the Blood paper, the “lower differentiation capacity” of non-BM MSCs was interpreted as less MSC-like, i.e. less stemness (which is in line with ISCT Minimal Criteria) and not MORE stemness as the authors here strangely concluded. Moreover, in the same Blood paper, several animal models were used to demonstrate in vivo trilineage differentiation capacity + hematopoiesis, which are highly compelling evidence.

We thank the reviewer for raising this question. We learned from the published articles that a tightly balanced core set of specific transcription factors are the major driving forces in stem cell maintenance, which are able to promote self-renewal by repressing transcription factors that initiate differentiation programs¹⁻⁴, and aging of stem cells are known to affect their function, leading to the loss of self-renewal capacity⁵, this is why we interpreted MSCs with lower differentiation capacity as being more stemness and less senescent in the first draft. However, following the suggestion of reviewer, we have recognized that MSC stemness is a quite controversial issue even to this day. We admitted if MSCs cannot be proved to be pluripotent stem cells with self-renewal ability, it is inappropriate to use differentiation ability to illustrate their senescence status, as

stronger differentiation capacity of MSCs was interpreted as ‘more stemness’ in many published studies. In this work, our core content and main innovations are to highlight the effect of cell senescence on the expression of PD-L1 in MSCs, and to prove the regulatory effect of GATA2 on the aging process and PD-L1 expression of MSCs. The stemness and differentiation function of MSCs are not the focus of our research, but the evidence we used to verify the aging status of cells. In addition, we have provided a large amount of evidence to prove cellular senescence, such as SA- β -gal staining (Fig. 3e), CCK8 proliferation assay (Fig. 3c), immunofluorescence staining for γ -H2AX (Supplementary Fig. 3e) and Western blot analysis (Supplementary Fig. 4k). We believe that removing differentiation-related content will not affect the quality and innovation of our work. Therefore, we have removed all content in the manuscript using MSC stemness and differentiation ability to verify its senescence status.

2. If the authors wish to demonstrate chronological senescence w/o the confounding issue of tissue-specific functional differences in differentiation and immunomodulation, the authors could have easily used BM and/or AD MSCs from donors of different ages to focus on this, since immunomodulation is well-reported with both these sources and thus can be evaluated in progressively older donor samples. Please see Park et al, Mech Ageing Dev 2005 & Zhou et al, Aging Cell 2008.

We thank the reviewer for the suggestions to provide further evidence to demonstrate our findings without confounding by tissue-specific functional differences that senescence is a key determinant of the PD-L1-mediated immunosuppressive activity of MSCs. We have followed the reviewer’s suggestion and performed another single cell RNA sequencing of BM-MSCs from young (n=4; age \leq 20) and aged (n=4; age $>$ 45) donors. Our analysis revealed that BM-MSCs can be divided into five clusters (**Fig. 5c**), BM5 and BM6 are identified as senescent cluster, while cells from BM1-3 are non-relatively non-senescent MSCs (**Fig. 5d; Supplementary Fig. 5d-f**). Furthermore, by comparing single cell data from young and aged BM-MSCs, we found that BM-MSCs from aged donors were more senescent, which resulted in a decrease in their immunosuppressive capacity and PD-L1 expression (**Fig. 5i-k; Supplementary Fig. 5h-j**). At last, we have also co-cultured CFSE labeled activated CD4⁺ T cell with young and aged BM-MSCs (**Supplementary Fig. 5l**). Compared to aged BM-MSCs, MSCs derived from young donors showed a stronger capacity to suppress T cells activation, while this gap was significantly reversed after treated with anti-PD-L1 Fab (**Fig. 5l; Supplementary Fig. 5n-p**), which indicates that upregulated PD-L1 mediates the enhanced immunosuppressive capacity of young BM-MSCs. Collectively, by

comparing BM-MSCs from donors of different ages, we have provided further evidence that cellular senescence is the key determinant of downregulated PD-L1 in MSCs from aged donors, which leads to a decrease in their immunomodulatory capacity.

3. General consensus in the field is that both chronological as well as replicative MSC senescence decreases osteogenesis and increases adipogenesis, or even decreases all 3 trilineage capacity. So why in this paper is less differentiation capacity considered more youthful/better “stemness”? More functional evidence is needed to overturn the overwhelming volume of publications to the contrary.

We thank the reviewer for the suggestion. As we replied in reviewer question Major 1, MSC stemness is a quite controversial issue, and we shouldn't have regarded enhanced MSC differentiation ability as a sign of reduced self-renewal ability due to aging. In addition, the differentiation ability of MSCs is affected by various aspects, including the tissue origin of MSCs⁶. Therefore, we have removed all content in the manuscript using MSC stemness and differentiation ability to verify its senescence status.

In addition, we have followed the reviewer suggestion and compared the trilineage differentiation ability of BM-MSCs derived from young and aged donors. Consistent with previous reports, BM-MSCs from aged donors showed decreased osteogenesis and increased adipogenesis potential (**Supplementary Fig. 5a**), which indicates a more senescent status of MSCs derived from aged donors.

4. It is unclear why passage 3 cells are already senescent/b-gal+. There is much information in the literature that human MSCs usually need to be passaged at least 3 or more passages before a more homogenous population—without WBC/non-MSC contamination—could be even seen, much less utilized, i.e. please see flow cytometry marker Table 5 in Mitchell et al, Stem Cells 2005, and also Brooke et al, Br J Hematol 2008 for clinical use.

We thank the reviewer for raising this question and we hope to explain this question through two points. Firstly, all passage 3 MSC lines we used for sequencing and functional experiments in this work have been strictly quality controlled according to ISCT criteria, all samples are subject to FACS analysis to ensure no blood cell contamination and tested for trilineage differentiation potential (**Supplementary Fig. 1a, b; Supplementary Fig. 5a, c**). Therefore, the appearance of senescent phenotype at passage 3 does not mean that our MSC samples are contaminated or not

homogeneous under the ISCT criteria. Secondly, the AD-MSCs and BM-MSCs we used for SA- β -gal staining in Fig. 3e were donated by 45-year-old and 51-year-old donors, respectively, which are relatively older donors. We have also complemented the comparison of the SA- β -gal staining of BM-MSCs derived from young and aged donors (**Fig. 5b**), the result revealed that young BM-MSCs (age=18) were rarely stained, while aged BM-MSCs showed a senescent phenotype. Altogether, these results indicate MSCs from aged donors may have undergone senescent in vivo, so the higher proportion of senescent cells makes SA- β -gal staining positive in the third passage.

Also, we apologize for not specifying the biological age of each sample in our first draft and we have complemented in Supplementary Table 1 the biological age information of each MSC sample used for sequencing and functional experiments in this manuscript.

5. MSC stemness is a quite controversial issue even to this day (please see numerous commentaries/reviews on this in Cell Stem Cell—latest is Soliman et al 2021, Nature series journals), so rigorous proof-of-concept evidence (at least in vitro, if not in vivo) is necessary, i.e. Hong et al, Science 2005.

We thank the reviewer for raising this question. As detailed response to the reviewer's question above, the innovation of our work is that cellular senescence affects the immunomodulatory ability of MSC cell therapy products, and the stemness of MSC is not the focus of this research. Therefore, without affecting the innovation and main content of the article, we decided to remove all content related to MSC stemness in the article.

6. The inclusion of pluripotency factors to stand for MSC stemness is especially controversial, i.e. Lengner et al, Cell Stem Cell 2007 & Piazzolla et al, Nat Commun 2014, and so functional relevance should be demonstrated.

Following the reviewer's suggestion, the results of *NANOG*, *SOX4* and *OCT4* have been removed from Supplementary Fig. 3d.

7. Key references relevant to tissue-specific functions, senescence, MSC PD-L1 research, etc. are not cited.

We thank the reviewer for the suggestion.

(1) We have supplemented reference relevant to MSCs senescent research: reference 15 (Wang D et al, Cancer Res. 2009); reference 16 (Ko E et al, Stem Cells Dev. 2012); reference 17 (Sepúlveda JC et al, Stem Cells. 2014); reference 18 (Zhou et al, Aging Cell. 2008.).

(2) We have supplemented cited article study MSC PD-L1: reference 63 (Sheng, H., et al. Cell Res); reference 64 (Davies et al, Stem Cells); reference 65 (Li M et al, J Extracell Vesicles. 2021)

(3) We have also supplemented cited article study MSC tissue-specific functions: reference 72 (Reinisch et al, Blood 2015); reference 73 (Yen BL et al, FEBS J. 2022); reference 74 (Collins E et al, J Immunol. 2014).

Specific comments:

1. It is unclear by what criteria PD-L1 was selected to be the gene of interest: a) no PD-L1 at the protein level was seen on when in the same cluster (C7), a number of related and also well-published MSC-immunomodulation genes—TGFb1, HGF, and HMOX1—are also increased; b) expression profiles of PD-L1 in all seven clusters (Figure 4b) did not correlate well with their gene set scores for immunosuppressive function (Figure 4a)

We thank the reviewer for the suggestions to better explain the reason why PD-L1 was selected to be the gene of interest. We have performed analysis to identify key modulators of immunomodulatory function during MSC senescence. We applied two criteria to narrow down our candidates: candidate genes should be among the highly variable genes, which determine the PCA results and clustering of single cells, and have a positive correlation with gene set scores for immunosuppressive function, and candidates should be upregulated in perinatal MSCs and have a positive correlation with gene set scores for immunosuppressive function (**Appendix Fig. 1**). Overlaying two sets of genes revealed 69 candidates, only 4 genes have been reported to be associated with MSC immunosuppressive function, including PD-L1 (CD274), which revealed that: (a) PD-L1 is among the highly variable genes, demonstrating a marked expression change of PD-L1 during dynamic aging progress; (b) PD-L1 is not only downregulated in senescent cell cluster but also downregulated in adult MSC on pseudobulk level; (c) PD-L1 expression is positively correlated with scores for immunosuppressive function. Furthermore, the accuracy of our data screening was proved again in another single cell dataset of young and aged BM-MSCs (**Fig. 5j**) and

repeatedly verified by functional experiments and. We performed multiple sets of FACS analysis and western blots, demonstrating the phenotype of significantly decreased expression of PD-L1 in senescent MSCs. This downregulation was proved to be functional related not only by detecting the inhibition of T cell proliferation, but also by detecting the inhibition of T cell secretion of inflammatory factors.

2. PD-L1 at the protein/surface level is constitutively expressed on human BMMSCs, i.e. please see Davies et al (Le Blanc lab), Stem Cells 2017 which focuses specifically on this topic.

We thank the reviewer for raising this question. Although the expression level was lower than that of perinatal-derived MSCs, we did find PD-L1 expression on BM-MSCs through FACS analysis (**Supplementary Fig. 4i, j; Supplementary Fig. 5j**) and western blots (**Supplementary Fig. 4k; Fig. 5k**).

3. This same paper also reports that PD-L1 can be secreted by BMMSCs, so this should be assayed.

We thank the reviewer for these helpful suggestions to detect secreted PD-L1, which is potentially important in modulating contact-independent mechanisms of immunosuppression. Firstly, we have supplemented with the protein sequencing data of extracellular vesicles (EVs) derived from UC- and BM- MSCs (**Fig. 4j; Supplementary Fig. 4m**). The PD-L1 expression was detected in proteomic data of EVs, while the EVs derived from UC-MSCs possessed a significantly higher expression of PD-L1 as compared to BM-MSC EVs. Furthermore, the secreted PD-L1 in culturing medium of young and aged BM-MSCs were also analyzed by ELISA (**Supplementary Fig. 5k**). Consistently, the levels of PD-L1 protein secreted by senescent BM-MSCs were also reduced. These results collectively indicate that senescence also reduced the ability of MSCs to secrete PD-L1.

4. The link between GATA2 and PD-L1 required functional validation, i.e. promoter assay, at the very least.

We thank the reviewer for the valuable suggestion to explore the mechanisms by which GATA2 promote the PD-L1 expression. We have followed the suggestions and performed the following new experiments as detailed below.

(1) We first explored the publicly available GATA2 CHIP-seq data and found that GATA2 peaks mapped to the promoter region of PD-L1 at about ± 500 bp around the TSS region (**Fig. 7d**). Thereafter, GATA2 factor binding sites around 500bp of the TSS region of the PD-L1 gene was located, and three promoter sequences were designed, which were named as region1 (-534bp to 0bp), region2 (0bp to 593bp) and region3 (-207bp to 134bp). These three promoter regions all contained GATA2 binding sites inside the sequences (**Fig. 7e**). We then transiently cotransfected the empty vector or GATA2-expressing plasmids with the region1 to region5 reporter into HEK293T cells and found that the luciferase activity was upregulated only by co-transfected GATA2-plasmid with vector containing promoter region 1 or region 3 (**Fig. 7f**), which indicates that the binding sites enable GATA2 to drive PD-L1 expression are likely to located at the overlap sequence of region1 and region3.

(2) We next analyzed GATA2 enrichment at PDL1 promoter by ChIP-PCR. We designed primers to detect GATA2 binding fragments at -153bp (primer 1) and 93bp (primer 2) of PD-L1 (Supplementary table 13). The ChIP-PCR results confirmed the predictive binding of GATA2 at the above two genomic loci (**Fig. 7g**). In addition, compared with BM-MSCs, young UC-MSCs detected more GATA2 recruitment at the PD-L1 promoter region, which provided a mechanistic explanation for higher PD-L1 expression in younger MSCs.

5. Along the same lines, in vitro overexpression of GATA2 in adult MSCs or knockdown of GATA2 in perinatal MSCs is necessary for the conclusions the authors are trying to make. In vivo evidence is strongly advised whenever stemness is being hypothesized.

We thank the reviewer for this important suggestion. According to the detailed response to this reviewer's question above, the stemness of MSC is not the focus of this research. Therefore, without affecting the innovation and main content of the article, we have removed all content related to MSC stemness in the article. In addition, we have followed the reviewer's suggestion and further complemented the GATA2-overexpression and -knockdown system, we provided better functional evidence for GATA2 to manipulate senescence progress and immunomodulatory capacity of MSCs. Specifically, we found:

(1) We first overexpressed GATA2 in adult BM-MSCs. Compared to MSCs infected with empty vector (EV), GATA2 overexpressed (GATA2-OE) BM-MSCs have weaker

β -glucosidase expression at low (**Fig. 7b**) and high culture passage (**Fig. 7c; Supplementary Fig. 7b**), which indicates the enhanced anti-senescence function of these cells. Through the FACS analysis and western blots, we have also proved that the overexpression of GATA2 upregulated the PD-L1 expression on adult MSCs (**Fig. 7h; Supplementary Fig. 7c**). This upregulation of PD-L1 was proved to be functional significant, as GATA2-OE BM-MSCs showed an increased suppressive effect when co-culturing with activated CD4⁺ T cells as compared to EV-BMMSCs, while this gap was significantly reversed after treated with anti-PD-L1 Fab (**Fig. 7i; Supplementary Fig. 7e-h**).

(2) Furthermore, we used lentivirus vectors (**Supplemental Figure 7i**) to knockdown GATA2 in perinatal UC-MSCs (**Fig. 7j**). UC-MSCs after GATA2 knockdown (KD) showed senescence phenotypes as compared to counterpart infected with shRNA scramble control (CT). The downregulations of PD-L1 at protein level of GATA2-KD UC-MSCs were also verified by FACS and western blots (**Fig. 7i; Supplementary Fig. 7j**). The GATA2-KD MSCs showed a decline in their capacity to suppress CD4⁺ T cells activation, besides, the GATA2-KD MSC/T cultures treated with anti-PD-L1 Fab showed almost none increase in T-cell proliferation and TNF- α /IFN- γ production compared with that of control group UC-MSC/T-treated cultures (**Figure. 7m, Supplementary Fig. 7k-o**), which indicates GATA2 as an important modulator of PD-L1 mediated immunosuppressive effect of UC-MSCs.

6. Reference #16 (Turinetto et al) is not appropriately cited: it does not mention a tendency of aged MSCs to undergo spontaneous differentiation. Rather, the review paper states that generally osteogenesis is compromised while adipogenesis is not affected or increased.

We thank the reviewer for the suggestion. We have followed the suggestion and reference #16 have been removed.

7. BM-3 appears to have significantly lower cell counts—why is this, and does this have any effect on the bioinformatics analyses?

We thank the reviewer for raising this question. Due to the difference in cell viability of each sample and batches of 10X cell capture reagents, single-cell sequencing on the 10X platform inevitably encounters batch effects and differences in the sequencing depth of single-cell data nowadays. We carried out strict sample data quality control for

each sample to remove the low-quality cells with low UMI counts and high mitochondrial gene ratio. In BM3, the 2,740 single cells with a total UMI count similar to that of other samples were included for downstream analysis. We appropriately applied the widely published Seurat integration method to remove batch differences between samples. Besides, the Seurat 'NormalizeData' function can further eliminated the influence of different sequencing depths in each sample. The results showed that each cell of BM3 is evenly distributed in the same 7 subpopulations as the rest of the samples (**Appendix Fig .2a**), and the marker genes of each subpopulation are the same as in the overall analysis (**Appendix Fig .2b**). This suggests that although our strict quality control may result in a lower cell counts in BM3, this was necessary and would not have a significant impact on our bioinformatics analysis.

8. Differentiation functional data is not well correlated to the microarray data, i.e. adipogenesis-related genes are upregulated in aged MSCs/C7.

We thank the reviewer for the suggestion. As detailed in our response to this reviewer's question above, we have removed all the content using MSC differentiation ability to assess their senescent status.

9. The UMAP in Figure S1i is in the opposite direction as all other similar UMAPs.10. In Table S2, genes in the list of distinct expression signatures for C7 cluster are duplicated.

We apologize for the confusion. In order to make the article more concise, we removed the information related to quality control from the supplementary figure 1 and transferred it to the Appendix Figure 3. During the quality control of our single cell data, we tried to explore whether dissociation has an effect on cell clustering. After removing dissociation-related genes, we re-performed the data normalization, and the new results affected the calculation of PCA dimensionality reduction analysis, resulting in different UMAP shapes and direction (**Appendix Fig. 3d**). After confirming that the dissociation-related genes do not affect our MSC clustering, the dissociation-related genes were kept in the dataset and were not removed from formal analysis, so the UMAP direction in the manuscript figures does not match the figure Appendix 3d.

Besides, we apologize for our mistake that genes in the list of distinct expression signatures for C7 cluster are duplicated. We have performed the "pseudobulk" differential gene analysis and updated the results in Supplementary Table 2.

Reviewer #2 (Remarks to the Author):

In their manuscript, “Multi-omics analysis of human mesenchymal stem cell reveals cell aging determining their immunomodulatory activity through the downregulation of PD-L1,” Gao et al. use single cell RNA sequencing (scRNA-Seq) along with mass spectrometry proteomics to characterize mesenchymal stromal cells (MSC) from different tissue sources (adipose tissue (AD), the bone marrow (BM), the placental chorionic membrane (PM) and the umbilical cord (UC)). The authors report that MSCs from different tissue sources (and correspondingly different ages) exhibit different degrees of cellular senescence, with MSCs from “younger” tissue sources exhibiting less senescence than MSCs from “older” tissue sources. Moreover, the authors report that these differences correspond to differences in PD-L1 expression by MSC, which in turn affects the immunosuppressive potential of the MSCs. Finally, they report that expression of PD-L1 in this context is mediated through GATA2 activity.

MSCs have shown great potential in a variety of clinical applications, but, as noted by the authors, there are many incongruous molecular, cellular and functional definitions of MSCs across studies, species, protocols, and tissue sources. There is an unmet need for carefully and thoroughly characterizing MSC resources, making the present study potentially significant. The application of cutting-edge technologies, such as scRNA-Seq and mass spec proteomics, to carefully defined MSC specimens from a variety of tissue sources is a scientifically valuable undertaking, and likely to uncover important MSC biology. Moreover, support of these characterizations with functional experiments evaluating PD-L1 expression, immunosuppressive activity, and GATA2-mediated regulation are important components that extend the studies beyond general “profiling” characterizations. Despite the potential impact of the study, there are significant concerns regarding analysis strategies and corresponding results interpretations, particularly with regard to scRNA-Seq analyses. Issues in these analyses makes it somewhat difficult to evaluate the biological explanations and conclusions drawn from these experiments, which in turn formed the basis for many of the functional follow up experiments. Beyond specific analysis issues, although acknowledged and discussed by the authors, an experimental design in which MSC tissue source is (necessarily) confounded with age presents challenges in determining if potential differences in senescence are due to tissue source, age, or both. Specific comments and suggestions are listed below:

We thank the reviewer for the encouraging comments and pointing out the existing problems. In order to prove that decreased immunomodulatory ability of MSCs is caused by senescence without confounded by tissue resource, we have followed the reviewer's suggestion and performed another single cell RNA sequencing of BM-MSCs from young (n=4; age \leq 20) and aged (n=4; age $>$ 45) donors (**Fig. 5a**). Our analysis revealed that BM-MSCs from aged donors were more senescent as compared to young BM-MSCs (**Fig. 5g, h**). The immunosuppressive function score of senescent BM-MSCs was also decreased and PD-L1 expression was also found to be significantly downregulated in BM-MSCs from aged donors (**Fig. 5i, j; Supplementary Fig. 5h, i**). we have also co-cultured CFSE labeled activated CD4⁺ T cell with young and aged BM-MSCs (**Supplementary Fig. 5l**). BM-MSCs derived from young donors showed a stronger capacity to suppress T cells activation, while inhibitory effect was significantly weakened after treated with anti-PD-L1 Fab (**Fig. 5l; Supplementary Fig. 5n-p**), which indicates that upregulated PD-L1 mediates the enhanced immunosuppressive capacity of young BM-MSCs. Collectively, by comparing BM-MSCs from donors of different ages, we have provided further evidence that cellular senescence is the key determinant of downregulated PD-L1 in MSCs from aged donors, which leads to a decrease in their immunomodulatory capacity.

MAJOR POINTS

1. Many aspects of the scRNA-Seq experiment and analysis workflow were well-suited to the goals of characterizing MSC from different sources. For example, balanced sample batching, implementation of data integration through Seurat, cell cycle scoring, etc. were appropriately applied. However, there are several stages of the analysis that raise concerns about and/or could improve the ultimate results and biological interpretations - these are summarized below.

Differential gene expression testing: Throughout the study, a variety of inter-cluster and inter-condition differential gene expression analyses are performed. As helpfully reported by the authors, the samples examined were balanced across three batches (reported in Sup Table 1). To account for potential inter-batch variation, along with inter-source variation, Seurat integration was appropriately applied to 'harmonize' the 12 datasets under study. The authors note that "The other analyses (like differential expression analysis and gene set enrichment) were based on the normalized data from the 'RNA' assay of Seurat object." This is the correct approach. However, in the subsequent differential gene expression testing, gene set testing, etc. there are no steps

taken to manage potential technical effects across batches. While this may not necessarily affect results (it represents an additional source of variation across replicates), differential expression results would likely be more robust when accounting for this source of variation.

For differential gene expression testing at the single cell level, the authors implement the `FindAllMarkers()` and `FindMarkers()` Seurat functions with Wilcoxon Rank-Sum Test or Student's T-test for statistics. First, with an integrated dataset, the `FindConservedMarkers()` function (multiple contrasts across each dataset), would likely be more robust. Furthermore, all of these functions can be called with different statistical frameworks, including those that implement linear modeling (e.g. DESeq2, MAST, etc.), in which batch (and other) factors can be included in model testing.

Perhaps a better strategy, given that the authors implemented a strong experimental design with true biological replicates ($n = 3$ for each tissue source), they might consider a 'pseudobulk' strategy for differential expression testing that models variation across biological rather than single cell "replicates." As presently applied, while sometimes necessary, single cell differential expression testing has been shown to be prone to high false positive rates (Squair JW, et al. Confronting false discoveries in single-cell differential expression. *Nat Commun.* 2021 Sep 28;12(1):5692). Given the fully replicated design here, differential gene expression with pseudobulk biological replicates would likely provide more robust results.

An improvement in differential gene expression results would also be expected to improve results in downstream analyses that used these results as input (e.g. Metascape, etc.).

We thank the reviewer for the suggestions to adopt a more rigorous strategy to calculate DEGs, we have applied a "pseudobulk" strategy for differential expression testing, which models variation across biological samples rather than single cell replicates. The codes for performing "pseudobulk" analysis have been uploaded to Github (https://github.com/GaoYuchenPUMC/MSC_paper/tree/main/code_for_paper_Revision), and we have updated the method of 'pseudobulk' DEG analysis in the manuscript. For each single cell cluster within each sample, we generated the "pseudobulk" cluster by summing UMI counts across all cells within this cluster. All "pseudobulk" clusters from the same tissue or age group constitute "pseudobulk" biological replicates. Next, we applied DESeq2 to calculate DEGs between different "pseudobulk" replicates. The

DEGs calculated by “pseudobulk” strategy were showed in supplementary table 2. The results of DEGs showed in dot plots and heatmaps are updated with the results calculated by “pseudobulk” strategy (**Fig. 1h; Fig. 2b, e; Supplementary Fig. 1f; Supplementary Fig. 2e**). Although the expression levels of many originally ‘false positive’ genes were no longer significantly different ($p_{adj} > 0.05$) after switching to the “pseudobulk” strategy, the key aging markers (TP53, CDKN1A, GLB1, CDKN2A) were still found to be differentially upregulated in aging clusters. Besides, the results of “pseudobulk” DEGs were further used to perform Gene Ontology analysis (**Fig. 2d; Supplementary Fig 2a, b**), which still indicates the senescent phenotype of cluster 5-7. Furthermore, we have also implemented the “pseudobulk” strategy to calculate DEGs of BM-MSCs from young and aged donors (**Fig. 5e; Supplementary Fig. 5d**), the results revealed that the biochemical basis for BM5 and BM6 senescence are misfolded protein response and DNA damage, respectively. Collectively, these results indicate that differences in cellular senescence at the single-cell level persist after switching to a more rigorous analysis strategy.

Cell cycle phase assignments and interpretations: Cell cycle scores were assigned per cell using the associated functionality in Seurat. This approach predicts a likely cell cycle phase (output as G1, S, or G2M) for each cell based on expression patterns of associated gene sets. While a useful tool, there are some concerns as to these data are interpreted in the text. For example, the authors note that (line 132), “MSCs from C1 were predicted to be in S phase, MSCs from C2 and C3 were assigned to G2/M phase, and cell cycle arrest occurred in, MSCs from C5 to C7, as nearly all cells were predicted to be in G1 phase.” A “G1” assignment does not indicate “cell cycle arrest.” Indeed, in our experience, this is a typical observation in scRNA-Seq data for cells that are not actively in the S/G2M proliferating process; in the Seurat classification function, there are no other options (e.g. “G0”, etc.), so the G1 label often acts as a “catch all” for non-S, non-G2M. We tend to observe large numbers of “G1” classified cells even in scRNA-Seq datasets of universally proliferating cell lines.

Related observations such as (line 142) “phenotype space between nonproliferative (C5, C6, C7) and proliferative clusters (C1, C2, C3) is relatively weak compared to the connectivity within either proliferative or nonproliferative subpopulations (Fig. 1f)” may not be fully explanatory, as the S/G2M gene expression patterns result in considerable variation across cells that can “drive” the PCA and consequent clustering and UMAP, without necessarily indicating distinct underlying phenotypes of the cells.

We thank the reviewer for raising this question. We have relearned the CellCycleScoring function of Seurat, which stores S and G2/M scores in object meta data, along with the predicted classification of each cell in either G2M, S or G1 phase. The CellCycleScoring function only assign each cell a S or G2M score, based on its expression of G2/M and S phase markers, cells expressed low levels of G2M and S phase genes were assigned G1 phase. Thus, cells predicted to be in the G1 phase are more like cells with relatively weak proliferative ability or cells that are not in the cell cycle state, and it is inappropriate for us to interpret it as “cell cycle arrest” or “non-proliferative”. Therefore, we have changed the interpretation of cell cycle analysis in the manuscript to: “MSCs from C1, C2 and C3 were assigned high G2/M or S phase score based on its high expression of G2/M and S phase markers. Cells from C5, C6 and C7 were likely not in active cycling and were assigned G1 phase, indicating a relatively low proliferative potential as compared to cells from C1, C2 and C3. C4 appeared to be a subset of cells in a transition state” (line 129). Besides, the interpretation of PAGA analysis was changed to “PAGA analysis revealed transcriptional similarity between active proliferating clusters (C1, C2 and C3), and strong connective structures were observed in phenotype space between C5, C6 and C7 (Fig. 1d). Consistent with the Monocle analysis, the PAGA path also suggested that active proliferating clusters gave rise to cells in C5, C6 and C7, while C4 appeared as a node that linking this transition” (line 138).

By extension, it is somewhat unclear how much of the gene expression differences in the “proliferative” vs “non-proliferative” clusters are driven simply by differences in cell cycle phase versus differences in senescence. While many of the genes highlighted by the authors are associated with senescence, alternatively, these differences in expression could be associated with cell cycle phases. It would be most helpful to have a “ground truth” scRNA-Seq dataset that included definitive senescent cells for direct comparison. These and related observed gene expression differences in “proliferative” vs. “non-proliferative” clusters may overinterpret the biological significance of these patterns, and do not necessarily support definitive statements such as “These results collectively indicate that the attenuated MSC proliferation can be ascribed to cellular senescence” (Line 152), and “Thus, our data at single-cell resolution indicate that the loss of proteostasis plays an important role in driving the cellular senescence of C5.” (Line 227), and “The abundant expression of genes such as EGR1, TP53, CDKN1A, CDKN2A and FOXO3 provided solid evidence that senescence of C6 and C7 is linked to genomic stress (Fig. 2a, b, Supplementary Fig. 2a) (Line 191).”

We thank the reviewer for raising this important question. We want to explain from the following three aspects that the cell clustering of MSCs data was more attributable to senescence, not the cell cycle.

(1) We first set to remove cell cycle effect on data dimensional reduction and cell clustering. Two methods are used in this section to remove the effect of cell cycle related genes⁷. The first method removed the cell cycle related genes from highly variable genes calculated by FindVariableFeatures, this method was reported before⁸ and can avoid cell cycle genes effect the downstream PCA analysis. However, after removal of cell cycle related genes, MSCs are still divided into seven clusters (**Appendix Fig. 4a**) with similar differentially expressed genes as before (**Appendix Fig. 4b**). The Pearson correlation analysis provided further evidence that MSCs were still divided as senescent and non-senescent cluster (**Appendix Fig. 4d**). Besides, perinatal tissue derived MSCs show higher percentage of non-aging clusters and lower percentage of aging cluster as compared to adult MSCs. In the second method, we first assigned each single cell a cell cycle score through the CellCycleScoring function, and then regressed out cell cycle score using ScaleData function. After regressed out cell cycle related effect (**Appendix Fig. 4e-f**), we found two non-aging clusters still highly expressed anti-senescence genes (DNMT1, EZH2, LMNB1), while three aging clusters were distinguished by upregulated senescent markers (GLB1, TP53, CDKN1A). Collectively, these results reveal that removal of cell cycle related effect make no change on cell clustering, which indicate the differentially expressed cell cycle genes are not major determinants of cell clustering. The codes of this part of analysis have been uploaded to Github.

(2) Secondly, while many of the genes highlighted as senescence marker could be associated with cell cycle phases, many aging marker genes (IL6, GLB1, HSPA9) and various SASPs factors (COL1A1, COL5A2, IGFBP5, FGF2) were not cell cycle related. These genes only highly expressed in senescent clusters and showed relatively low expression in non-aging clusters (**Appendix Fig. 4b, f**), if cell cycle alone determined cell clustering, the expression of these genes should not vary appreciably across cell populations. After removing the effect of cell cycle, the expression difference of these senescent related genes can still determine the cell grouping.

(3) At last, each young and aged BM-MSC sample was tested for senescence prior to single-cell sequencing (**Fig. 5b**) and high proportion of senescent MSCs was identified within adult-derived BM-MSCs at early passage 3, which indicates that MSCs from

aged donors may have experienced senescence before leaving the bone marrow niche. Our single cell analysis results also showed that aged BM-MSCs possessed a significantly higher frequency of senescent cells (**Fig. 5h**) expressing aging markers and SASPs factors (**Fig. 5e**). These results may provide further evidence that cell clustering of MSCs data was more attributable to senescence, not the cell cycle.

Gene set scoring by GSVA: GSVA, a widely used tool for gene set scoring and enrichment analysis in “bulk” RNA-Seq studies, was used to assign per cell gene set scores, which in some cases were subsequently used to test for differential set enrichment across samples. Unlike standard “bulk” RNA-Seq data, scRNA-Seq datasets are extremely sparse and have many different characteristics than bulk datasets. Is there any indication that GSVA has been tested and found to be appropriate/robust for single cell RNA-Seq data? There appears to be a github package under development (<https://github.com/guokai8/scGSVA>), but there do not seem to be any peer reviewed indications that this tool performs appropriately for scRNA-Seq data, raising concerns about these results. If the authors wish to use the GSVA tool for scRNA-Seq data, they must demonstrate its robust performance with appropriate controls and ground truth benchmarking, and/or cite such a demonstration.

We thank the reviewer for the suggestion to improve our analysis of gene set activity. Based on previous reports⁹, we agree that GSVA is not designed for single-cell analysis. We decided to choose R packages AUCell (version 1.8.0) for gene set analysis, which was recommended for single-cell analysis⁹ and used in many published works^{10,11}. The AUCell was used to convert the cell-by-gene matrix into a cell-by-gene-set matrix based on normalized data with default parameters, the reference gene sets were the same as we used in GSVA analysis (**Supplementary Table 3**). The ‘pseudobulk’ strategy were also used to assess differential activities of pathway (scored per cell by AUCell) between ‘pseudobulk’ replicates (**Supplementary Table 4**), only gene sets score with significant expression differences were shown (**Fig. 1c**). The code used for AUCell gene set scoring have been uploaded to Github.

Trajectory analysis: Trajectory analysis was performed using Monocle v2. These types of analyses can present potential pitfalls for integrated multisample data. However, there are no details provided on how the trajectory analysis was implemented for the integrated datasets. Furthermore, while most of the analysis code is helpfully available via the provided github link, it appears that this repository does not contain the code for the Monocle trajectory analyses, which made it challenging to evaluate this aspect of

the study. In particular, Monocle analysis allows for specifying several parameters regarding the expected number of roots/branches, etc. from trajectory data. Without knowing how these parameters were set, the interpretation that “mesenchymal clusters occurred on a onefold trajectory, starting from cells in the active 139 proliferating phase (C1, C2 and C3) and ending with C7 (Fig. 1e)” may not be appropriate.

We thank the reviewer for the suggestion to improve our analysis of pseudotime trajectory analysis. In the first edition of manuscript, trajectory analysis was performed using Monocle version 2.12.0. We have reperformed the analysis after upgrading the version of Monocle from 2.12.0 to 2.26.0 (**Supplementary Fig. 1e**), which is the new version fixed existing problems and improved performance and stability. The result of monocle analysis revealed 4 branch points along the trajectory (**Appendix Fig. 5a**), which divided MSCs along the trajectory to 8 states, with non-aging cluster 1-3 mainly resides at state 6 and 7, while aging clusters mainly resides at state 1 (**Appendix Fig. 5b**). Therefore, state 7 was set as the “root_state” in orderCells function, as nonsenescent MSCs highly expressed proliferative genes (**Appendix Fig.5c, d**). The code used for Monocle trajectory analysis have been uploaded to Github.

RNA velocity: RNA velocity analysis can also present potential pitfalls for integrated multisample data, especially when attempting to compare velocity results across different conditions (as in Fig 3). It is not clear how the different samples (and different batches) were managed in RNA velocity analysis which makes it challenging to evaluate the results.

We apologize for not providing the complete code due to our mistake. The new code has been completed and uploaded to Github. Besides, in order to prove our RNA velocity results were not affected by integrated analysis, we have performed RNA velocity analysis in each 12 MSC sample from adult and perinatal tissue respectively (**Appendix Fig. 6**). All MSC samples from adult tissue bearing longer vectors from C3 to C4 as compared to MSCs from perinatal tissue, which provide further evidence that adult MSCs have a stronger tendency to progress to senescent clusters during in vitro expansion.

2. The inclusion of functional assays for cell proliferation, immunosuppression by MSCs via PD-L1, and GATA2 regulation of PD-L1 expression are a strength of the paper, following up on hypotheses generated by the scRNA-Seq and proteomics data. However, while significant, the differences in immunosuppressive effects via PD-L1

appear to be somewhat modest – would this difference be sufficient to explain some of the differences observed in vivo?

We thank the reviewer for raising this question. We have performed multiple experimental groups (perinatal and adult MSCs; aged and young donor BM-MSCs; GATA2-modified MSCs) to detect PD-L1-mediated immunosuppression effect. In each experimental group, we have not only detected the proliferation of T cells, but also evaluated the change of their function by detecting the inflammatory factors secreted by T cells. These exhaustive experimental data provided solid evidence that the downregulation of PD-L1 is sufficient to attenuate the immunosuppressive capacity of senescent MSCs. Furthermore, we agree that in vivo experiments are important to verify the function of PD-L1, we have reported here a clinical trial (NCT04014166) of the use of PD-L1 abundant perinatal UC-MSCs in treating refractory immune thrombocytopenia (ITP), which is an autoimmune disorder characterized by immune-mediated destruction of circulating platelets (PLTs) and suppression of PLT production. These six participants all received splenectomy and showed no durable response, indicating a complicated pathogenic mechanism of these patients and the research on new therapies for them are necessary. The results showed that participants in high dose group (2×10^6 MSCs/kg) had great odds of response after the first 4 weeks after MSC transfusion with all six participants showed no severe adverse effect (**Supplementary Fig. 8**). Our single-cell study on MSCs and this phase I trial are the early phase research of clinical MSC therapy, the follow-up data of this clinical trial will be reported in the future work, we are also preparing for a clinical trial of PD-L1 overexpressed MSCs in the treatment of refractory ITP. However, due to the various limitations of human clinical trial, we cannot conduct extensive validation, so the current results cannot provide direct evidence for the role of PD-L1 in vivo. For this part of the clinical data, we are willing to hear the opinions from the reviewer. If the reviewer think that this part of in vivo clinical trial data is not convincing, we also agree with this point of view and will delete it.

ADDITIONAL MINOR POINTS

The text notes that genes associated with dissociation-induced artifact were removed to test for such effects. Were these genes removed for the final analysis? While it's good that they were checked for dissociation gene expression patterns, removing these genes is not appropriate — the genes could also be involved in the MSC biological pathways under study.

We thank the reviewer for raising this question. During the quality control of our single cell data, we have found that the dissociation-related genes do not affect our MSC clustering (**Appendix Fig. 3d-f**), the dissociation-related genes were then kept in the dataset and were not removed from formal analysis.

The text references “Supplemental Table 10”, but this table does not appear in the supplied Excel workbook.

We apologize for the mistake. We have removed the wrong text and revised the supplementary tables.

In the Methods section, there are no details provided as to how the number of principal components were selected for scRNA-Seq dimensionality reduction or how the cluster resolution was determined.

We thank the reviewer for raising this question. We supplemented the details in the methods section of our manuscript (line 989). The number of principal components is set to 30, for an ‘elbow’ was observed around PC 30 through the ElbowPlot function, suggesting the majority of true signal is captured in the first 30 PCs. The resolution is set to 0.4, as small clusters appeared when the resolution is higher than 0.4, which possessed DEGs with low Log2foldchange value, and most of these DEGs overlapped with other clusters, making it hard to define their functional characteristics.

Reviewer #3 (Remarks to the Author):

The authors have performed an exhaustive gene and protein expression analysis of mesenchymal stroma cells (MSC). The goal of the single cell based analysis was to describe the cellular phenotype variation in MSC and compare MSC from bone marrow, adipose tissue, placenta and cord blood. The data confirmed that MSC are not homogenous in culture, and that the characteristics of the different sources organs are maintained. Fewer proliferative and stem like cells are present in adult MSC than in perinatal MSC, while more senescence markers and in general G1/2 phase markers are reciprocally present. This is not new, except for the now very thorough analysis showing the shift in senescence and the cell source specific relative proportions of senescence and non-senescent cells. The data also confirm many of the previous

findings on factors influencing or accompanying proliferation, senescence, immunosuppressive function or differentiation. It remains unclear whether the comparison is biologically sound as the MSC from different tissue sources may as well represent different cell types, which are compared here. The transcriptomic analysis hints in this direction. The finding that PD-L1 is associated with senescence and immunosuppression was confirmed, but is not new (e.g. non referenced paper by Davies, 2017). New is the finding that GATA2, by regulating for example PD-L1, is important for maintaining a rather non-senescent phenotype and maintaining a better immune suppression in the used biological assays. It would be of interest whether this function can be shown in long-term cultures after multiple passaging of cells in vitro - and not only in peroxide induced senescence. The authors claim that knowing the transcriptomic profiles and factors can be used to influence senescence and thus maintain a homogenous and strong immunosuppressive phenotype in clinical products. This is a well written manuscript. However, the complex methodological procedure and amount of figures (6 Figures and 61 panels) are not needed to get the information out and should be reduced and focused on the main new findings. The specific points raised are:

We thank the reviewer for the encouraging comments. Following the suggestion of the reviewer, we have used lentivirus vectors to overexpress GATA2 in adult BM-MSCs at passage 3, and passaged these MSCs to passage 15. We then complemented SA- β -gal staining and quantification of EV and GATA2-OE BM-MSCs at passage 15 (**Fig. 7c; Supplementary Fig. 7b**). As compared to the GATA2-OE counterparts, EV BM-MSCs at passage 15 exhibited an enlarged size and higher percentage of staining positive cell, which indicates that GATA2 overexpression enhanced the anti-senescence function of BM-MSCs under in vitro expansion.

- elucidate the point that the MSC from different source organs are intrinsically different and cannot be compared as potentially identical cell types. Can the data be analysed under this aspect? Are 'tissue origin related differences among subpopulations...' also found between populations?

We thank the reviewer for raising this question. We do understand that the function of MSCs is tissue-differentiated, which presents challenges in determining if potential differences in senescence are due to tissue source or age. Therefore, we have followed the reviewer's suggestion and performed another single cell RNA sequencing of BM-MSCs from young (n=4; age \leq 20) and aged (n=4; age $>$ 45) donors. Our analysis revealed

that BM-MSCs can be divided into five clusters (**Fig. 5c**). The analysis of DEGs revealed that MSCs from BM1-3 expressed anti-senescence genes, while BM5 and BM6 highly expressed aging markers (**Fig. 5d; Supplementary Fig. 5d-f**). Furthermore, MSCs from older donors were found to be more senescent, with reduced immunosuppressive capacity due to PD-L1 decline (**Fig. 5i-k; Supplementary Fig. 5h-j**). At last, we have also co-cultured CFSE labeled activated CD4+ T cell with young and aged BM-MSCs (**Supplementary Fig. 5l**). Compared to aged BM-MSCs, MSCs derived from young donors showed a stronger capacity to suppress T cells activation. After PD-L1 blockade with specific antibodies, the inhibition of CD4+ T-cell activation by young BM-MSCs was largely reversed, while the proliferation and TNF- α / IFN- γ secretion in the presence of aged BM-MSCs was changed only slightly after PD-L1 blockade, indicating relatively low endogenous PD-L1 expression in BM-MSCs from aged donors. (**Fig. 5l; Supplementary Fig. 5n-p**), which indicates that upregulated PD-L1 mediates the enhanced immunosuppressive capacity of young BM-MSCs. Collectively, by comparing BM-MSCs from donors of different ages, we have provided further evidence without confounded with tissue resource that cellular senescence is the key determinant of downregulated PD-L1 in MSCs from aged donors, which leads to a decrease in their immunomodulatory capacity.

- is it possible that the cell clusters just reflect different cell cycle stages, which could be misinterpreted as senescent vs. proliferative?

We thank the reviewer for raising this important question. We would like to answer your questions through the following points and clarify that the cell clustering of MSCs data was more attributable to senescence, not the cell cycle.

(1) We first set to remove cell cycle effect on data dimensional reduction and cell clustering. Two methods are used in this section to remove the effect of cell cycle related genes⁷. The first method removed the cell cycle related genes from highly variable genes calculated by FindVariableFeatures, this method was reported before⁸ and can avoid cell cycle genes effect the downstream PCA analysis. However, after removal of cell cycle related genes, MSCs are still divided into seven clusters (**Appendix Fig. 4a**) with similar differentially expressed genes as before (**Appendix Fig. 4b**). The Pearson correlation analysis provided further evidence that MSCs were still divided as senescent and non-senescent cluster (**Appendix Fig. 4d**). Besides, perinatal tissue derived MSCs show higher percentage of non-aging clusters and lower percentage of aging cluster as compared to adult MSCs. In the second method, we first

assigned each single cell a cell cycle score through the CellCycleScoring function, and then regressed out cell cycle score using ScaleData function. After regressed out cell cycle related effect (**Appendix Fig. 4e-f**), we found two non-aging clusters still highly expressed anti-senescence genes (DNMT1, EZH2, LMNB1), while three aging clusters were distinguished by upregulated senescent markers (GLB1, TP53, CDKN1A). Collectively, these results reveal that removal of cell cycle related effect make no change on cell clustering, which indicate the differentially expressed cell cycle genes are not major determinants of cell clustering. The codes of this part of analysis have been uploaded to Github.

(2) Secondly, while many of the genes highlighted as senescence marker could be associated with cell cycle phases, many aging marker genes (IL6, GLB1, HSPA9) and various SASPs factors (COL1A1, COL5A2, IGFBP5, FGF2) were not cell cycle related. These genes only highly expressed in senescent clusters and showed relatively low expression in non-aging clusters (**Appendix Fig. 4b, f**), if cell cycle alone determined cell clustering, the expression of these genes should not vary appreciably across cell populations. After removing the effect of cell cycle, the expression difference of these senescent related genes can still determine the cell grouping.

(3) At last, each young and aged BM-MSC sample was tested for senescence prior to single-cell sequencing (**Fig. 5b**) and high proportion of senescent MSCs was identified within adult-derived BM-MSCs at early passage 3, which indicates that MSCs from aged donors may have experienced senescence before leaving the bone marrow niche. Our single cell analysis results also showed that aged BM-MSCs possessed a significantly higher frequency of senescent cells (**Fig. 5h**) expressing aging markers and SASPs factors (**Fig. 5e**). These results may provide further evidence that cell clustering of MSCs data was more attributable to senescence, not the cell cycle.

-donor age should be reported (AD and BM MSC)

We apologize for not specifying the biological age of each sample. We have complemented in Supplementary Table 1 the biological age information of each MSC sample used for sequencing and functional experiments in this manuscript.

- reduce complexity of text and figures and focus on main novel findings. For example, most transcriptomic and proteomics data confirm what has been known, but novelty is limited (although the provided cell atlas is valuable).

We thank the reviewer for the suggestion to present data in a clearer way. Following the reviewer's suggestion, we have reduced the complexity of figures by transferring redundant panels to corresponding supplemental figures (now is 7 figures and 64 panel after new experimental data being complemented), which made the manuscript more conducive to presenting the main new findings. Furthermore, we have simplified the complex methodological procedure descriptions (line 121, line 145, line 305, line 320, line 442), and we have reduced the description of known background knowledge in our data analysis (line 159, line 171, line 193, line 196, line 206, line 228, line 461).

- it should be discussed whether the detected functional differences are potentially clinically relevant (how predictive are the used suppression assays to predict clinical efficacy)?

We thank the reviewer for raising this question. We agreed that in vivo experiments are important to verify the functional difference caused by PD-L1 in young and aged MSCs. However, we have reported here a phase I clinical trial (NCT04014166) of the use of UC-MSCs in treating refractory immune thrombocytopenia (ITP), which is an autoimmune disorder characterized by immune-mediated destruction of circulating platelets (PLTs) and suppression of PLT production. Here we have described the research methodology, study design and patient outcomes in detail (**Supplementary Fig. 8**), and discussed here the promising effectiveness and safety of UC-MSCs for the treatment of refractory ITP. Our single-cell study on MSCs and this phase I trial are only the early phase research of clinical MSC therapy, the follow-up data of this clinical trial will be reported in the future work, we are also preparing for a clinical trial of PD-L1 overexpressed MSCs in the treatment of refractory ITP. However, various limitations of human clinical trial have prevented us from conducting experimental validation, so the current results cannot provide direct evidence for the immunosuppressive role of PD-L1 in vivo. Besides, for this part of the clinical data, we are willing to listen to the opinions from the reviewer. If the reviewer think that this part of in vivo clinical trial data is not convincing, we also agree with this point of view and will delete it.

Recommend to accept with major revision.

We thank the reviewers for giving us the opportunity to revise our work, we have supplemented detailed experiments and revised the manuscript as required.

References

- 1 Bernstein, B. E. *et al.* A bivalent chromatin structure marks key developmental genes in embryonic stem cells. *Cell* **125**, 315-326, doi:10.1016/j.cell.2006.02.041 (2006).
- 2 Boyer, L. A. *et al.* Core transcriptional regulatory circuitry in human embryonic stem cells. *Cell* **122**, 947-956, doi:10.1016/j.cell.2005.08.020 (2005).
- 3 Boyer, L. A. *et al.* Polycomb complexes repress developmental regulators in murine embryonic stem cells. *Nature* **441**, 349-353, doi:10.1038/nature04733 (2006).
- 4 Lee, T. I. *et al.* Control of developmental regulators by Polycomb in human embryonic stem cells. *Cell* **125**, 301-313, doi:10.1016/j.cell.2006.02.043 (2006).
- 5 Oh, J., Lee, Y. D. & Wagers, A. J. Stem cell aging: mechanisms, regulators and therapeutic opportunities. *Nature medicine* **20**, 870-880, doi:10.1038/nm.3651 (2014).
- 6 Al-Nbaheen, M. *et al.* Human stromal (mesenchymal) stem cells from bone marrow, adipose tissue and skin exhibit differences in molecular phenotype and differentiation potential. *Stem cell reviews and reports* **9**, 32-43, doi:10.1007/s12015-012-9365-8 (2013).
- 7 Tirosh, I. *et al.* Dissecting the multicellular ecosystem of metastatic melanoma by single-cell RNA-seq. *Science (New York, N.Y.)* **352**, 189-196, doi:10.1126/science.aad0501 (2016).
- 8 Zheng, L. *et al.* Pan-cancer single-cell landscape of tumor-infiltrating T cells. *Science (New York, N.Y.)* **374**, abe6474, doi:10.1126/science.abe6474 (2021).
- 9 Clarke, Z. A. *et al.* Tutorial: guidelines for annotating single-cell transcriptomic maps using automated and manual methods. *Nature protocols* **16**, 2749-2764, doi:10.1038/s41596-021-00534-0 (2021).
- 10 Psaila, B. *et al.* Single-Cell Analyses Reveal Megakaryocyte-Biased Hematopoiesis in Myelofibrosis and Identify Mutant Clone-Specific Targets. *Molecular cell* **78**, 477-492.e478, doi:10.1016/j.molcel.2020.04.008 (2020).
- 11 Wouters, J. *et al.* Robust gene expression programs underlie recurrent cell states and phenotype switching in melanoma. *Nature cell biology* **22**, 986-998, doi:10.1038/s41556-020-0547-3 (2020).

Appendix Fig. 1

Correlation analysis between immunosuppressive function score and 2000 highly variable genes (Cor>0.1, P<0.05)

Correlation analysis between immunosuppressive function score and Perinatal MSCs highly expressed genes (Cor>0.1, P<0.05)

CCL2 ACTG2 KRT18 CD274 TFPI2 DIRAS3 SPON2 CDC20
MEST RBP1 AURKB IL11 GAP43 FABP4 KRT8 ADAM19 PMAIP1
TNFAIP LYPD1 BDKRB1 HGF LRRC17 TMEM88 GAL NLRP1
MDK HHEX QPRT CSRP2 UCP2 PDLIM3 POU2F2 PITX1
MARCKSL1 SLC12A8 SOCS1 HSPA2 SERTAD4-AS1 TMEM158
RP6-91H8.3 CDCA7 E2F1 ARHGDIB NETO2 MLLT11 RP11-89C3.4 CDC45
MGARP CSGALNACT1 ITGA2 MEX3A FAM213A FCHO1 NTF3
BATF3 TMSB15A STXBP2 ISYNA1 SMIM1 MARCH3 SLC7A7
KLHL23 DKK3 RHOA RAC3 GALNT6 GALNT12 RMI2 JUP

Appendix Figure 1. PD-L1 was an important gene mediating the decline of immunosuppressive function during senescence. Venn diagram showing the overlap of genes with high correlation with immunosuppressive function score, only six genes colored in red was genes with immunomodulatory ability.

Appendix Figure 2. Dimensionality reduction and marker genes of BM3. (a), UMAP projection of 2,740 BM3 MSCs. **(b),** Dotplot showing the marker genes of each BM3 MSC cluster.

Appendix Fig. 3

Appendix Figure 3. Quality control of single cell transcriptomic data.

(a), Bar plots showing the cell number, the number of transcripts and the number of genes for all QC-passed cells from each sample (with plot center, box and whiskers corresponding to median, IQR and $1.5 \times \text{IQR}$, respectively). **(b)**, UMAP plot showing the contaminated cluster (C8), which may be generated by experimental manipulation. **(c)**, Heatmap showing specifically expressed genes in C8 with the representative GO enrichment results listed on the right. **(d)**, UMAP showing the clustering result after removing the dissociation-related genes from the datasets. **(e)**, Heatmap displaying the signature genes of each cluster, which were nearly unchanged after removing the dissociation-related genes. **(f)**, Violin plot showing the GSVAscore for the dissociation-related gene set of each subcluster.

Appendix Fig. 4

1. Remove cell cycle related genes from Variable Features, avoid its effect on the downstream PCA analysis

2. Assign each single cell a Cell Cycle Score, and Regress out Cell Cycle Score using ScaleData function

Appendix Figure 4. Effect of removal of cell cycle genes on clustering

(a), UMAP showing seven cluster of MSCs after remove cell cycle related genes from Variable Features. **(b)**, Dot plots showing the scaled expression of representative senescent-related genes for each cluster calculated in appendix Fig. 4a. **(c)**, Correlation matrices showing the Pearson correlation coefficients of the 7 MSC clusters calculated after remove cell cycle related genes and 7 MSC clusters as shown in Fig. 1b. **(d)**, Fractions of subpopulations in six adult versus six perinatal MSC samples (Error bar: Mean \pm s.d.s). P values are generated by two-tailed t test with Welch's correction (** $p < 0.005$; *** $p < 0.0005$). **(e)**, UMAP showing six cluster of MSCs after regress out cell cycle scoring. **(f)**, Dot plots showing the scaled expression of representative senescent-related genes for each cluster calculated in appendix Fig. 4e. **(g)**, Correlation matrices showing the Pearson correlation coefficients of the 6 MSC clusters calculated after regress out cell cycle scoring and 7 MSC clusters as shown in Fig. 1b. **(h)**, Fractions of clusters in six adult versus six perinatal MSC samples (Error bar: Mean \pm s.d.s). P values are generated by two-tailed t test with Welch's correction (** $p < 0.005$; *** $p < 0.0005$).

Appendix Fig. 5

Appendix Figure 5. Trajectory analysis using monocle2

(a-c), Monocle trajectories of MSCs colored by cluster identity (a), predicted cell state (b) and predicted pseudotime value (c). **(d)**, Relative expression level of senescent related genes along the pseudotime sequence.

Appendix Fig. 6

a

b

Appendix Figure 6. RNA velocity analysis of each MSC sample

(a), RNA velocity analysis of 6 MSC sample from adult tissues.

(b), RNA velocity analysis of 6 MSC sample from perinatal tissues.

REVIEWER COMMENTS

Reviewer #1 (Remarks to the Author):

While the authors have made a heroic effort in the revision by substantially increasing sample numbers of different ages and more functional assays to validate the role of GATA2/PD-L1 in MSC senescence, the decision to just merely delete the stemness portion not only ignores the biological identity of the studied cells, but also completely changes the hypothesis & direction of the research. In this “no-stemness “ context, finding any molecular mechanism (i.e. the GATA2/PDL1 axis) loses much its significant because the only functional outcome for non-stem cell senescence, AKA ‘regular’ somatic cell senescence, is proliferation, because there is no issue of differentiation capacity for non-stem cells. And in this non-stem cell context, the finding of PD-L1 involvement in immunomodulation is completely expected, leaving only the modulation by GATA2 being novel. Moreover, if typical non-stem cell/somatic cell senescence is emphasized—meaning reversal and/or changes in differentiation capacity is no longer an issue—then the assessment of only proliferation is not enough, i.e. cell cycle dynamics and other more senescence-specific molecular mechanistic reversals by perturbing GATA2/PD-L1 needs to be evaluated.

The decision to just leave out stemness in the manuscript is actually problematic on several levels: 1) fundamentally, the authors are considering MSCs as stromal cells and NOT stem cells, and therefore has assigned immunomodulatory properties as a functional characteristic of M-stromal-Cs, a function which have yet to reach consensus; 2) the authors still show trilineage differentiation experiments AND use the changes in trilineage differentiation capacity as evidence of senescence (Supp. Fig 5a), so clearly the assayed cells ARE stem cells—how can these functions be just ignored?; and 3) so why do the authors still use mesenchymal STEM cells throughout the article? Perhaps a most troubling concern is that the authors are misunderstanding stem cell biology: in the reply, the statement “if MSCs cannot be proved to be pluripotent stem cells...” shows that the authors are maybe uncertain on stem cell biology in general, and pluripotent stem cells vs. somatic stem cells in particular.

The addition of clinical data perhaps demonstrates the commitment of authors to MSC translational use, but actually does not add to support the hypothesis of the manuscript itself (no involvement of GATA2, PD-L1, or senescence), so it is also unclear why this data was added in the revision.

Minor comment: The authors should make available the .rds file, including meta data, labeled assays, and reduction map information in order to ensure the reproducibility of the scRNA-seq data.

Reviewer #2 (Remarks to the Author):

In their revised manuscript, Gao et al. make substantial improvements to their initial submission and respond to many of the reviewer concerns detailed in the original review. The addition of several experimental datasets further demonstrating a role for GATA2 in PD-L1 regulation, clarification of computational analyses, and important methodological details all strengthen the manuscript. However, despite these improvements, significant concerns remain regarding portions of manuscript content, particularly for single cell RNA-Seq (scRNA-Seq) analyses and interpretations.

MAJOR POINTS

- The inclusion of new mechanistic data in Figure 7 and Supplemental Figure 7 further supports a role for GATA2 in regulating senescence and PD-L1 expression. These additions significantly improve the manuscript and provide support for some of the patterns observed in scRNA-Seq data.
- There remain significant concerns regarding the degree to which cell cycle phase differences potentially contribute to the different MSC clusters defined in scRNA-Seq analysis. In the initial

review, multiple reviewers raised concerns about whether differences in cell cycle phase might be driving the heterogeneity annotated as different MSC “senescent” and “non-aging” clusters. The authors helpfully address some of these points by modifying the language used in describing cell cycle assignments and reframing their interpretation of these data. However, as noted in the initial review, it still remains unclear how much of the gene expression differences in the “proliferative” vs “non-proliferative” clusters are driven simply by differences in cell cycle phase versus differences in senescence. While the authors attempt to address this in their response to initial review, the following issues remain:

- The authors explain that when removing cell cycle genes from the highly variable genes used for PCA and clustering, there are minimal differences in cluster assignments (Appendix 4a-c). However, the “cell cycle genes” used are the Seurat cell cycle gene sets `cc.genes\$s.genes` and `cc.genes\$g2m.genes`. While very effective in calculating single cell cycle scores and regressing out these signatures, these gene sets are not comprehensive; there are many additional genes that may be associated with cell cycle that are not included in these sets. Therefore, even upon removal of these genes, remaining cell cycle-associated genes could still be driving the clustering pattern observed.

- The authors next regress out Seurat cell cycle scores and observe its impact on UMAP visualization and clustering (Appendix 4e). While they note that they still observe different clusters, it is very apparent that the clusters are far less distinct than the initial analysis, suggesting that indeed cell cycle gene expression likely contributes considerably to cluster assignment. Importantly, these critiques are not intended to suggest that the authors’ interpretation of these cells as “senescent” vs. “non-aging” are necessarily incorrect (subsequent experimentation demonstrates very clearly the difference in senescent cell frequencies in these samples!); however, if the presented output of these analyses is a characterization of the transcriptomic heterogeneity of these cells, it is important to acknowledge the contributions of cell cycle to clustering. In sum, while the different clusters may represent and/or overlap with senescent vs. non-aging cells, some portion of this heterogeneity may be driven by cell cycle and/or cell cycle-associated effects.

- In initial submission, multiple reviewers commented on the potentially confounding effects of comparing MSC from different tissue sources. In the more recent submission, the authors have included a helpful comparison of BM-derived MSC from young vs old donors. Importantly, as noted in the text, all samples were collected and processed in a single technical batch, enabling direct comparisons of groups. This addition significantly strengthens the manuscript and adds additional support to the authors’ claims of age-associated senescence programs. However, the manuscript also includes inter-tissue comparisons of four different tissue sources that were processed in different batches. While not necessarily a problem, direct comparisons across these samples without either 1) demonstrating the (unlikely) absence of a batch effect or 2) accounting for potential technical batch effects in differential gene expression testing could lead to spurious results. At the advice of reviewers, the authors did add pseudobulk differential gene expression testing to take advantage of their biological replication. However, in their analysis code, it appears that they did not include a design term for “batch” (which are noted and balanced across groups in Supplemental table 1). While unlikely to dramatically change the results, given the appropriate experimental setup, replication and analysis approach, the DESeq2 model (or alternative approach) should account for “batch” to ensure the most robust results.

In addition, throughout the manuscript, many figures (non-comprehensive examples include figures 1h, 2b, 2f, and others) present expression values (either averaged across cells/samples or for individual cells) for comparison across groups or clusters. These figures appear to be generated with Seurat functions. It is not clear if the values are “integrated” assay values (which are not readily interpretable across group comparisons) or normalized “RNA” assay values. While RNA assay values are appropriate for visualization, they would not account for technical batch differences, which could complicate comparisons. This should be addressed and/or indicated to enable clarity of interpretation for these visualizations.

- In response to the initial review, the authors have helpfully demonstrated consistent RNA velocity patterns in different samples. RNA velocity figures appear multiple times in the manuscript. At multiple points, the authors make interpretations about the relative magnitude of RNA velocity vectors that appear to be based primarily on visual observation of integrated UMAP projections (lines 248-252). These are likely overinterpretations of these data, which while potentially supportive of the authors' model, are not really necessary to prove their points. In particular, inclusion and acknowledgment of differences in relative cell frequency per cluster (lines 252-254, Fig3b, Supplemental Fig 3a) provide much stronger data to support differences in adult vs. perinatal sources; these are controlled for tissue source and batch (Seurat integration) and should not be susceptible to gene expression differences. It is this reviewer's opinion that the same conclusions could be drawn by focusing on these frequency differences rather than the RNA velocity results.

- In response to initial review, per cell gene set scores have now been calculated and compared with AUCell, an appropriate tool for this application at single cell resolution. However, the authors also note application of AUCell to pseudobulk profiles (lines 1055-1056). It is not clear from the available analysis code how AUC scores were compared for differential pathway activity analyses in pseudobulk profiles or if this approach is appropriate.

- The clinical trial data, which while interesting in comparing MSC dosing regimens for clinical response, does not inform the manuscript's primary subject (senescence in MSCs). It seems outside the scope of the present manuscript.

- As noted in the previous review, at multiple points throughout the manuscript main text, the authors use language that suggests firm biological conclusions drawn solely from scRNA-Seq analysis. Some examples include:

- Lines 148-149: "These results collectively indicate that the attenuated MSC proliferation can be ascribed to cellular senescence."

- Lines 202-204: "Thus, our data at single-cell resolution indicate that the loss of proteostasis plays an important role in driving the cellular senescence of C5."

- Lines 257-259: "Additionally, this heterogeneity in the dynamics of transition led to a major increase in the proportion of C7 but caused no significant expansion of C5 and C6 in adult MSCs (Fig. 3b; Supplementary Fig. 3a)."

In each of these cases, the text states firm causal relationships ("ascribed to...", "plays an important role in driving...", "led to a major increase...") inferred from the scRNA-Seq results. While the single cell gene expression data may suggest, infer, be consistent with, and/or inform future investigations, it is not sufficient to support such definitive statements, which would require validating experimentation (which is provided by the authors in several cases). These statements, which appear throughout the text, should be modified accordingly.

Reviewer #3 (Remarks to the Author):

The authors have responded to the points made by the reviewers in detail and added relevant information, experimental data and explanations. The key finding that GATA2 regulates PD-L1 and thus by analogy immunosuppressive function, and that this is impaired on older (senescent) MSC is supported. The general correlation between ageing, stemness and differentiation properties has been more realistically described. Thus, I am of the opinion that the experimental options to strengthen the manuscript and support the finding have been exhausted.

Point by point response to reviewers

We sincerely appreciate the efforts of all the reviewers for their careful evaluation and valuable feedback on our manuscript. Your insightful comments and constructive criticism have significantly help us to improve the quality our work. Following the suggestions by the reviewers, we have made several revisions to the bioinformatics analysis and writing to ensure the accuracy and reliability of the results, and we have also clarified some sections to better convey the research findings. Among the highlights of the new results, we have supplemented the data regarding stemness and further discussed our finding on how cellular senescence affects MSC stemness. In addition, we optimized and complemented the analysis to remove the cell cycle effect and better assess its contribution to cluster formation. The article also includes a detailed discussion of the impact of the cell cycle on cell clustering. Furthermore, we have improved the analysis of RNA velocity by adopting more rigorous strategies and revising the previous over interpreted results, which further enhanced the credibility of our analysis. Finally, we have revised the manuscript to tone down conclusions that are only drawn from scRNA sequencing, thus increasing the rigor of our results. The new data are highlighted in a blue font in the revised manuscript, including the legends and labels of figures and supplementary tables. With the new results, we believe we have addressed all of the reviewers' concerns, substantially improved the accuracy and reliability of our work, and broadened the impacts of our study.

A brief summary of the major new results is listed below, which are divided into four categories.

A. Analysis of cell senescence effects on MSC stemness.

1. The comparison of stem cell signature gene score of seven MSCs clusters derived from adult and perinatal tissue (**Supplementary Fig. 1f**).
2. The stem cell signature gene scores of AD-MSCs, BM-MSCs, PM-MSCs, and UC-MSCs were compared (**Supplementary Fig. 3d**).
3. The real time qPCR analysis of pluripotency marker expression in AD-MSCs, BM-MSCs, PM-MSCs and UC-MSCs (**Supplementary Fig. 3e**).
4. The comparison of stem cell signature gene score of six BM-MSCs clusters derived from young and aged donors (**Supplementary Fig. 5g**).
5. The expanded discussion on the effects of aging on MSC stemness, including areas of uncertainty in stemness function and the limitations of stemness investigation in this study (**line 626**).

B. Further investigation to assess the effect of cell cycle genes on cell clustering.

1. The results of cell clustering after complete removal of cell cycle related genes and avoid the cell cycle genes from affecting downstream clustering analysis (**Appendix Fig. 1**)
2. The results of cell clustering after Seurat cell cycle regression analysis (**Appendix Fig. 2**)

3. The results of cell clustering after cell cycle effect regression by Scanpy “regress_out” function (**Appendix Fig. 3**).
4. A further discussion of the cell cycle effect on MSCs clustering (**line 569**).

C. The revision of RNA velocity analysis.

1. The re-interpretation of RNA velocity analysis on MSCs from adult and perinatal tissues (**line 252**).
2. The RNA velocity analysis of all BM-MSCs from young and aged donors (**Fig. 5g**).

D. Adoption of more rigorous strategies to calculate DEGs and AUCell gene set score.

1. The DEG analysis between different “pseudobulk” replicates was performed using the R package DESeq2 with batch effect correction, and the improved DEG results were used to update GO enrichment analysis (**Supplementary Table 2; Fig. 2d; Supplementary Fig. 2a, b**).
2. The ‘pseudobulk’ strategy were also used to assess differential activities of pathway scored per cell by AUCell with batch effect correction (**Supplementary Table 4**).

The appendix figures were presented as additional results at the end of our response letter (**Appendix Fig. 1-3**). We prefer not to include these data in the manuscript due to space limitations and the scope of our study, but we will be happy to include them if the reviewers or the editors think it is necessary. Unless otherwise noted, all figure callouts below correspond to figures in the revised manuscript.

Reviewers' comments:

Reviewer #1 (Remarks to the Author):

While the authors have made a heroic effort in the revision by substantially increasing sample numbers of different ages and more functional assays to validate the role of GATA2/PD-L1 in MSC senescence, the decision to just merely delete the stemness portion not only ignores the biological identity of the studied cells, but also completely changes the hypothesis & direction of the research. In this “no-stemness” context, finding any molecular mechanism (i.e. the GATA2/PDL1 axis) loses much its significant because the only functional outcome for non-stem cell senescence, AKA ‘regular’ somatic cell senescence, is proliferation, because there is no issue of differentiation capacity for non-stem cells. And in this non-stem cell context, the finding of PD-L1 involvement in immunomodulation is completely expected, leaving only the modulation by GATA2 being novel. Moreover, if typical non-stem cell/somatic cell senescence is emphasized—meaning reversal and/or changes in differentiation capacity is no longer an issue—then the assessment of only proliferation is not enough, i.e. cell cycle dynamics and other more senescence-specific molecular mechanistic reversals by perturbing GATA2/PD-L1 needs to be evaluated.

The decision to just leave out stemness in the manuscript is actually problematic on several levels: 1) fundamentally, the authors are considering MSCs as stromal cells and NOT stem cells, and therefore has assigned immunomodulatory properties as a functional characteristic of M-stromal-Cs, a function which have yet to reach consensus; 2) the authors still show trilineage differentiation experiments AND use the changes in trilineage differentiation capacity as evidence of senescence (Supp. Fig 5a), so clearly the assayed cells ARE stem cells—how can these functions be just ignored?; and 3) so why do the authors still use mesenchymal STEM cells throughout the article? Perhaps a most troubling concern is that the authors are misunderstanding stem cell biology: in the reply, the statement “if MSCs cannot be proved to be pluripotent stem cells...” shows that the authors are maybe uncertain on stem cell biology in general, and pluripotent stem cells vs. somatic stem cells in particular.

We sincerely appreciate reviewer for the constructive comments. We apologize for the possible misunderstanding caused by previous response and we admit that it may not be appropriate to directly remove stemness relevant content. We would like to respond to the reviewers' concerns by the following points:

(1) To clarify, we removed the stem cell aspects of the manuscript not because we believe that MSCs lack stemness and should be considered solely as stromal cells, but rather to emphasize our focus on the immunomodulatory properties of stem cells. Stem cell immunomodulatory property is a crucial research focus in the fields of stem cell biology and cell therapy¹⁻³, and it represents the core content and main innovations of our manuscript. Our study confirms for the first time that cell aging plays a role in the downregulation of PD-L1 expression and elucidates its regulatory mechanism during the in vitro senescence process. Extensive experimental data also showed that the downregulation of PD-L1 is involved in the decline of immunomodulatory capacity of aging MSCs. Given that PD-L1 is a crucial target of cancer immune therapy, its regulatory mechanisms have always been a central focus of research. The GATA2-PDL1 regulatory mechanism that we have discovered holds significant potential for advancing tumor immune study and cell therapy.

(2) Although this study builds upon the previous work of our team⁴⁻¹² and focuses more on the clinical application value of MSCs, we acknowledge the importance of discussing the effect of aging on MSC stemness. According to the valuable suggestions raised by the reviewer, we have re-supplemented the stemness-related results in the article (**Supplementary Fig. 1f, Supplementary Fig. 3d-e, Supplementary Fig. 5g**) and updated the corresponding text description (**line 67, line 146, line 148, line 233, line 271, line 275, line 383, line 402**). We recognize that in the original version of the manuscript, it was inaccurate to infer differences in stemness by comparing the heterogeneity in differentiation capacity between perinatal and adult MSCs, as there is an effect of different tissue origin. Therefore, we have removed this section from the revised manuscript. Instead, in this revision, we have focused on the changes in the

differentiation capacity of MSCs from young and aged donors derived from the same tissue and further discussed it in the Discussion section.

(3) The stemness of MSCs has always been a controversial issue¹³⁻¹⁵. While our study provided evidence that senescence reduces the expression of stem cell signature genes, restricts MSC proliferation, and alters their differentiation potential, research in future is required to fully explore other stem cell characteristics affected by the aging process. Additionally, the senescence-associated transcription factors and underlying regulatory molecular mechanisms we outlined in Fig. 2h require further investigation. Therefore, in this revised version, we have expanded the discussion on the effects of aging on MSC stemness, including what we are not sure about in the stemness function (**line 626**).

The addition of clinical data perhaps demonstrates the commitment of authors to MSC translational use, but actually does not add to support the hypothesis of the manuscript itself (no involvement of GATA2, PD-L1, or senescence), so it is also unclear why this data was added in the revision.

We thank the reviewer for the suggestion and we have removed the clinical trial data from this version of manuscript.

Minor comment: The authors should make available the .rds file, including meta data, labeled assays, and reduction map information in order to ensure the reproducibility of the scRNA-seq data.

We appreciate the reviewer's valuable suggestion and are committed to ensuring the reproducibility of our work. As such, we will make our processed scRNA-seq data publicly available. In compliance with the rules on the administration of human genetic resources of China, we have uploaded our RDS files and meta data (which contain labeled assays and reduction map information) to OMIX/China National Center for Bioinformatics (<https://ngdc.cncb.ac.cn/omix>: accession no. OMIX003273) as described in the **Data Availability** section, and these data will be public accessible once the manuscript is accepted. The following link has been created to allow review of RDS file and meta data while it remains in private status: <https://ngdc.cncb.ac.cn/omix/preview/dneHI96B>. Besides, we have uploaded all sequencing raw data to public databases, and all code related to the data analysis in this manuscript, including the new code used for revision, has been uploaded to Github: https://github.com/GaoYuchenPUMC/MSC_paper. We believe that these data, along with our newly uploaded RDS data, will greatly ensure the reproducibility of this study.

Reviewer #2 (Remarks to the Author):

In their revised manuscript, Gao et al. make substantial improvements to their initial submission and respond to many of the reviewer concerns detailed in the original review. The addition of several experimental datasets further demonstrating a role for GATA2

in PD-L1 regulation, clarification of computational analyses, and important methodological details all strengthen the manuscript. However, despite these improvements, significant concerns remain regarding portions of manuscript content, particularly for single cell RNA-Seq (scRNA-Seq) analyses and interpretations.

We thank the reviewer for the careful evaluation and the kind comments on the revised manuscript.

MAJOR POINTS

- The inclusion of new mechanistic data in Figure 7 and Supplemental Figure 7 further supports a role for GATA2 in regulating senescence and PD-L1 expression. These additions significantly improve the manuscript and provide support for some of the patterns observed in scRNA-Seq data.

We thank the reviewer for the encouraging comments on our revised manuscript.

- There remain significant concerns regarding the degree to which cell cycle phase differences potentially contribute to the different MSC clusters defined in scRNA-Seq analysis. In the initial review, multiple reviewers raised concerns about whether differences in cell cycle phase might be driving the heterogeneity annotated as different MSC “senescent” and “non-aging” clusters. The authors helpfully address some of these points by modifying the language used in describing cell cycle assignments and reframing their interpretation of these data. However, as noted in the initial review, it still remains unclear how much of the gene expression differences in the “proliferative” vs “non-proliferative” clusters are driven simply by differences in cell cycle phase versus differences in senescence. While the authors attempt to address this in their response to initial review, the following issues remain:

- The authors explain that when removing cell cycle genes from the highly variable genes used for PCA and clustering, there are minimal differences in cluster assignments (Appendix 4a-c). However, the “cell cycle genes” used are the Seurat cell cycle gene sets ``cc.genes$s.genes`` and ``cc.genes$g2m.genes``. While very effective in calculating single cell cycle scores and regressing out these signatures, these gene sets are not comprehensive; there are many additional genes that may be associated with cell cycle that are not included in these sets. Therefore, even upon removal of these genes, remaining cell cycle-associated genes could still be driving the clustering pattern observed.

- The authors next regress out Seurat cell cycle scores and observe its impact on UMAP visualization and clustering (Appendix 4e). While they note that they still observe different clusters, it is very apparent that the clusters are far less distinct than the initial analysis, suggesting that indeed cell cycle gene expression likely contributes considerably to cluster assignment.

Importantly, these critiques are not intended to suggest that the authors' interpretation of these cells as "senescent" vs. "non-aging" are necessarily incorrect (subsequent experimentation demonstrates very clearly the difference in senescent cell frequencies in these samples!); however, if the presented output of these analyses is a characterization of the transcriptomic heterogeneity of these cells, it is important to acknowledge the contributions of cell cycle to clustering. In sum, while the different clusters may represent and/or overlap with senescent vs. non-aging cells, some portion of this heterogeneity may be driven by cell cycle and/or cell cycle-associated effects.

We thank the reviewer for raising this question. According to reviewer's concerns we have optimized and complemented the analysis to remove cell cycle effect and further assessing their contribution to cluster formation. The codes for this part of analysis have been uploaded to Github (https://github.com/GaoYuchenPUMC/MSC_paper/tree/main/code_for_Revision%232/1_CellCycleRelatedGenesRegression).

(1) Firstly, we acknowledge that Seurat cell cycle gene set ('cc.genes\$.genes' and 'cc.genes\$g2m.genes', n=97) is not comprehensive enough to remove the cell cycle effect in PCA analysis. Therefore, we selected the cell cycle gene set in the GO database (GOBP_CELL_CYCLE, GO:0007049), which contains 1,847 cell cycle-related genes. This gene set is sufficient to help us remove all cycle genes among highly variable genes and avoid the cell cycle genes from affecting downstream PCA analysis. After removing the cell cycle genes and performing dimensional reduction analysis, we were still able to successfully group the seven clusters that are similar to the version in Figure 1b (**Appendix Fig. 1a**). The differential expression of aging-related genes and Pearson correlation analysis further indicate that MSCs may still be divided into senescent and non-senescent clusters (**Appendix Fig. 1b-c**). Additionally, the removal of cell cycle genes did not change one of our major conclusions that perinatal MSCs contained a lower proportion of senescent cells (**Appendix Fig. 1d**).

(2) Secondly, we understand the reviewer's concern that after cell cycle score regression, cell clusters become less distinct than in the initial analysis, as three non-aging clusters (C1, C2, and C3) merged into two clusters (Non-aging1 and Non-aging2). In this version of the revision, we did not change the strategy of Seurat cell cycle regression, as the Seurat cell cycle gene sets were effective in calculating single-cell cycle scores and regressing out these signatures. However, we found that by adjusting the downstream dimensional reduction analysis parameters (changed to nPC=45, resolution=0.6), the new analysis result again revealed seven populations of MSCs that were similar to the original version (**Appendix Fig. 2a**) and showed differential expression of senescence-related markers (**Appendix Fig. 2b**). We further inferred from Pearson analysis that although the regression of the cell cycle effect may make clustering less obvious on the UMAP visualization, the seven clusters we obtained showed a high similarity in transcriptomic gene expression level compared to the pre-treated version (**Appendix Fig. 2c**), and the frequencies of cells in the new clusters still

supported our conclusion that adult MSCs possess a more senescent state than perinatal MSCs (**Appendix Fig. 2d**). These results collectively suggest that the reduction in the number of cell clusters and less distinct clustering observed in the first round of revision may be due to differences in the selection of parameters, and that there is no significant change in cell clusters after Seurat regression of the cell cycle effect.

(3) Furthermore, to further evaluate the influence of the cell cycle on MSC clustering, we used the Python-based single-cell analysis package Scanpy to assign each cell a cell cycle score and perform cell cycle regression using the Scanpy "regress_out" function. This method is widely recognized for removing cell cycle effects independent of the R language and Seurat package. The results of the Scanpy analysis also showed seven clusters after removing the cell cycle effect on clustering (**Appendix Fig. 3a**). The non-aging clusters (C1r, C2r, and C3r) calculated by Scanpy still highly expressed anti-senescence genes (DNMT1, EZH2, LMNB1), while three aging clusters (C5r, C6r, and C7r) were distinguished by upregulated senescent gene levels: GLB1, TP53, CDKN1A (**Appendix Fig. 3b**). The seven subpopulations calculated by Scanpy were highly correlated at the transcriptomic level compared with the corresponding subpopulation calculated by Seurat (**Appendix Fig. 3c**). Moreover, the cell frequencies of the seven clusters calculated by Scanpy also support our conclusion that perinatal MSCs are less senescent and contain fewer aging cells during in vitro culturing (**Appendix Fig. 3d**). These results collectively suggest that Scanpy and Seurat, two widely recognized tools for removing the cell cycle effect on clustering, can independently reproduce the results that cell clustering is barely affected after cell cycle effect regression.

Based on all the aforementioned analysis results, we suggest that differences in the expression of cell cycle genes among different cells have no significant impact on cell clustering. All analysis results indicate that cell clustering did not change significantly after comprehensive removal of cell cycle genes through widely recognized methods. However, as we compared the proportions of subpopulations before and after removing cell cycle effects, we found that there were subtle changes in subgroup proportions (**Appendix Fig. 1e, Appendix Fig. 2e, Appendix Fig. 3e**), although most of the changes were not significant. As pointed out by the reviewer, our analysis result characterizes all transcriptomic heterogeneity of the cells. Since cellular senescence inevitably leads to expression differences in cell cycle genes, we acknowledge that although these differences are not sufficient to significantly change the clustering, some proportion of the heterogeneity may be driven by cell cycle and/or cell cycle-associated effects. Therefore, we have supplemented the discussion of this issue in the article (**line 569**) and we want to also explain below why we did not remove the cell cycle effect in the clustering analysis in this study:

Firstly, our comprehensive analysis suggested that differences in the expression of cell cycle genes may have an impact on cell clustering, but it was not sufficient to significantly alter the clustering results. After we removed the effect of the cell cycle, there were no significant changes in cell clustering, and the results of the cell cycle

regression did not affect our major conclusions of this study.

Secondly, cellular senescence is defined as a state of permanent cell cycle arrest, the changes in cell cycle during cellular senescence are also an important characteristic.

Thirdly, cell cycle genes may be involved in the proliferation and self-renewal of stem cells. Therefore, the clustering generated by cell cycle genes in stem cells may have biological significance^{16,17}. Additionally, the proliferation capacity is an important factor for measuring MSCs' function, and retaining the influence of the cell cycle on clustering may better reflect the functional state of MSCs. Many studies using single cell technology on stem cell research retain the influence of the cell cycle on clustering¹⁸⁻²⁵, which is in line with common practice and may provide more comprehensive analysis results.

- In initial submission, multiple reviewers commented on the potentially confounding effects of comparing MSC from different tissue sources. In the more recent submission, the authors have included a helpful comparison of BM-derived MSC from young vs old donors. Importantly, as noted in the text, all samples were collected and processed in a single technical batch, enabling direct comparisons of groups. This addition significantly strengthens the manuscript and adds additional support to the authors' claims of age-associated senescence programs. However, the manuscript also includes inter-tissue comparisons of four different tissue sources that were processed in different batches. While not necessarily a problem, direct comparisons across these samples without either 1) demonstrating the (unlikely) absence of a batch effect or 2) accounting for potential technical batch effects in differential gene expression testing could lead to spurious results. At the advice of reviewers, the authors did add pseudobulk differential gene expression testing to take advantage of their biological replication. However, in their analysis code, it appears that they did not include a design term for "batch" (which are noted and balanced across groups in Supplemental table 1). While unlikely to dramatically change the results, given the appropriate experimental setup, replication and analysis approach, the DESeq2 model (or alternative approach) should account for "batch" to ensure the most robust results.

We thank the reviewer for suggesting further improvements to our DEG analysis. According to the reviewer's suggestion, we used DESeq2 to correct for batch effects (design = ~ batch + pseudobulk cluster) when performing DEG analysis on single-cell data from different batches. The codes and the design file for performing batch-corrected "pseudobulk" DEG analysis have been uploaded to GitHub (https://github.com/GaoYuchenPUMC/MSC_paper/tree/main/code_for_Revision%232/2_DEseq2_correct_batch_effect), and we have updated the DEG analysis methodology in the manuscript (**line 1012**). The DEG results obtained after batch effect correction are presented in Supplementary Table 2. Although correcting for batch effects eliminated statistical significance ($p_{adj} > 0.05$) for a small subset of genes, the expression of the aging-related genes we are interested in remains unaffected (**Fig. 1h**;

Fig. 2b, e; Supplementary Fig. 1f; Supplementary Fig. 2e). Additionally, we updated our GO analysis using the new DEG results, which still support our original conclusion that misfolded protein response and DNA damage may contribute to MSC senescence in different clusters (**Fig. 2d; Supplementary Fig. 2a, b**). Overall, the more robust DEG results suggest the existence of differences in the expression levels of aging-related genes between single-cell clusters.

In addition, throughout the manuscript, many figures (non-comprehensive examples include figures 1h, 2b, 2f, and others) present expression values (either averaged across cells/samples or for individual cells) for comparison across groups or clusters. These figures appear to be generated with Seurat functions. It is not clear if the values are “integrated” assay values (which are not readily interpretable across group comparisons) or normalized “RNA” assay values. While RNA assay values are appropriate for visualization, they would not account for technical batch differences, which could complicate comparisons. This should be addressed and/or indicated to enable clarity of interpretation for these visualizations.

We thank the reviewer for raising this question. In this study, we utilized data from the ‘integrated’ assay solely for dimensional reduction (PCA), UMAP visualization, and cell clustering. The other analyses, such as differential expression analysis and gene set enrichment, were conducted using normalized data from the ‘RNA’ assay of Seurat object. All the values presented in the figures were normalized or scaled values obtained from the ‘RNA’ assay. As mentioned by reviewer, we agreed that the RNA assay values did not account for technical batch differences and may complicate comparisons. However, we implemented a rigorous algorithm to eliminate false positives caused by single-cell replicates (pseudobulk strategy) and further corrected the batch effect using DESeq2, which ensured the reliability and appropriateness of our new DEG results for visualization.

Following the reviewer’s suggestion, we have added additional information on data visualization in the "Data Visualization and Statistical Information" section of the Methods (**line 1161**). Furthermore, we have included a statement in the figure legends clarifying that the values used for visualization were from the ‘RNA’ assay (**Fig. 1h; Fig. 2b; Fig. 2e; Fig. 4b; Fig. 5e; Fig. 5j; Supplementary Fig. 1c; Supplementary Fig. 1f; Supplementary Fig. 2e; Supplementary Fig. 3b-c; Supplementary Fig. 5c; Supplementary Fig. 5i; Supplementary Fig. 6d**).

- In response to the initial review, the authors have helpfully demonstrated consistent RNA velocity patterns in different samples. RNA velocity figures appear multiple times in the manuscript. At multiple points, the authors make interpretations about the relative magnitude of RNA velocity vectors that appear to be based primarily on visual observation of integrated UMAP projections (lines 248-252). These are likely overinterpretations of these data, which while potentially supportive of the authors’ model, are not really necessary to prove their points. In particular, inclusion and

acknowledgment of differences in relative cell frequency per cluster (lines 252-254, Fig3b, Supplemental Fig 3a) provide much stronger data to support differences in adult vs. perinatal sources; these are controlled for tissue source and batch (Seurat integration) and should not be susceptible to gene expression differences. It is this reviewer's opinion that the same conclusions could be drawn by focusing on these frequency differences rather than the RNA velocity results.

We thank the reviewer for the valuable suggestion. We totally agree with your suggestion that the variations in the frequency of clusters between adult and perinatal MSCs present more compelling evidence to substantiate their distinct aging statuses., and we have made the following changes to the corresponding content of the manuscript in order to revise our interpretation of RNA velocity results:

(1) We have deleted the original analysis text for Fig. 3a as it may have been an overinterpretation of the data. We have reinterpreted this analysis and now state: "RNA velocity analysis suggests that MSCs from adult and perinatal tissues share a similar cellular transition pattern" (**line 252**). Furthermore, we have modified the text describing the frequencies of cells to highlight the difference in relative cell frequency per cluster between adult and perinatal MSCs (**line 255**).

(2) We have deleted the original RNA velocity analysis and its interpretation in Fig. 5g, which compared the RNA velocity vectors between young and aged BM-MSCs. Instead, we have supplemented the RNA velocity analysis of all aged and young BM-MSCs in Fig. 5g to replace the original context. We now interpret this analysis as: " We also conducted RNA velocity analysis, which suggested that the transition trajectory of BM-MSCs begins from non-aging clusters (BM1, BM2, and BM3) and ends in senescent clusters BM5 and BM6" (**line 409**).

- In response to initial review, per cell gene set scores have now been calculated and compared with AUCCell, an appropriate tool for this application at single cell resolution. However, the authors also note application of AUCCell to pseudobulk profiles (lines 1055-1056). It is not clear from the available analysis code how AUC scores were compared for differential pathway activity analyses in pseudobulk profiles or if this approach is appropriate.

We thank the reviewer for raising this question. We have uploaded the code for differential analysis of AUCCell results to Github

(https://github.com/GaoYuchenPUMC/MSC_paper/tree/main/code_for_Revision%232/3_AUCCell_pseudobulk_analysis). We calculated the activity of gene sets in single cells using AUCCell, which convert the cell-by-gene matrix into a cell-by-gene-set matrix based on normalized data with default parameters. For each single-cell cluster within each sample, we generated a "pseudobulk" cluster by summing the gene set score across all cells within that cluster. The "pseudobulk" clusters of the same type from different tissue groups constitute biological replicates. To correct for batch effects, we

performed differential analysis between different "pseudobulk" replicates using the R package DESeq2 with the parameter "design" set to '~ batch + pseudobulk cluster'. We have supplemented the method for calculating differential AUCell scores in the method section (**line 1049**), the results of the differential analysis after batch effect correction have been updated in Supplementary table 4.

- The clinical trial data, which while interesting in comparing MSC dosing regimens for clinical response, does not inform the manuscript's primary subject (senescence in MSCs). It seems outside the scope of the present manuscript.

We thank the reviewer for the insightful comment and we have removed the clinical trial data from this version of manuscript.

- As noted in the previous review, at multiple points throughout the manuscript main text, the authors use language that suggests firm biological conclusions drawn solely from scRNA-Seq analysis. Some examples include:

- Lines 148-149: "These results collectively indicate that the attenuated MSC proliferation can be ascribed to cellular senescence."
- Lines 202-204: "Thus, our data at single-cell resolution indicate that the loss of proteostasis plays an important role in driving the cellular senescence of C5."
- Lines 257-259: "Additionally, this heterogeneity in the dynamics of transition led to a major increase in the proportion of C7 but caused no significant expansion of C5 and C6 in adult MSCs (Fig. 3b; Supplementary Fig. 3a)."

In each of these cases, the text states firm causal relationships ("ascribed to...", "plays an important role in driving...", "led to a major increase...") inferred from the scRNA-Seq results. While the single cell gene expression data may suggest, infer, be consistent with, and/or inform future investigations, it is not sufficient to support such definitive statements, which would require validating experimentation (which is provided by the authors in several cases). These statements, which appear throughout the text, should be modified accordingly.

We thank the reviewer for the valuable suggestion. According to reviewer's concerns, we have tone down conclusions that are only drawn from scRNA sequencing. Specific revisions for this part are shown below:

(1) **line 123**: The interpretation of AUCell heatmap has been modified to "The top expressed gene sets in each cluster are displayed on the heatmap, which suggested that MSCs from C1, C2 and C3 highly expressed genes involved in cell cycle progression and proliferation".

(2) **line 148**: The interpretation of gene set score analysis in Fig. 1 has been modified to "Based on the available evidence, it seems possible that the downregulation of MSC proliferative and stem cell signature genes may be linked to cellular senescence".

(3) **line 164:** The conclusion has been revised to “Collectively, our comprehensive bioinformatic analysis suggests that clinically used MSC products from various tissue origins have an underestimated level of heterogeneity, and our findings have also helped reconstruct a progressive senescence process in MSCs, which is characterized by a decline in proliferation capacity and stem cell features”.

(4) **line 173:** The interpretation of DEG analysis in Fig. 2b has been modified to “The abundant expression of genes such as TP53, CDKN1A and CDKN2A suggested that senescence of C6 and C7 is linked to genomic stress”.

(5) **line 182:** The interpretation of gene set score analysis in Fig. 2c has been modified to “Gene set scoring analysis also suggested that, compared to MSCs from C5 and non-senescent clusters, the unresolved DNA damage might be a contributing factor to the cellular senescence observed in MSCs from C6 and C7, accompanied by downregulated PI3K-AKT pathway activity in these cells”.

(6) **line 186:** The conclusion has been revised to “Based on these results, we inferred that genomic stress generated during in vitro expansion might be an inducer of cell senescence in C6 and C7”.

(7) **line 195:** The interpretation of GO analysis has been modified to “These results indicate that cellular stress in C5 is likely to be induced by the abnormal accumulation of unfolded or misfolded proteins in the ER.”

(8) **line 204:** The conclusion has been revised to “Together, these results suggest that the loss of proteostasis might contribute to the cellular senescence of C5”.

(9) **line 245:** The conclusion has been revised to “Collectively, our data offer valuable insights into the underlying biochemical mechanisms and molecular regulatory networks that may be involved in MSC senescence.”

(10) **line 262:** The conclusion has been modified to “These results suggest that compared to perinatal MSCs, MSCs derived from adult tissue exhibit higher levels of cellular senescence”.

(11) **line 276:** We have revised the conclusion to “Collectively, these analyses suggest a possible molecular regulatory mechanism that may contribute to the observed differences in aging levels between MSCs derived from adult and perinatal sources”.

(12) **line 407:** The interpretation of Pearson correlation analysis in Fig. 5f has been modified to “Our Pearson correlation analysis also suggested that cells from BM1-3 have high correlation value with C1-3, while BM5-6 show more similarity to aging cluster C5-7”.

(13) **line 412:** The conclusion has been revised to “Overall, based on another single-cell dataset, we inferred that the cellular senescence process during in vitro expansion may result in the emergence of MSC clusters with different aging statuses.”

Reviewer #3 (Remarks to the Author):

The authors have responded to the points made by the reviewers in detail and added relevant information, experimental data and explanations. The key finding that GATA2 regulates PD-L1 and thus by analogy immunosuppressive function, and that this is impaired on older (senescent) MSC is supported. The general correlation between ageing, stemness and differentiation properties has been more realistically described. Thus, I am of the opinion that the experimental options to strengthen the manuscript and support the finding have been exhausted.

We appreciate the time and effort the reviewer devoted to the review process, and we are grateful for reviewer’s valuable comments on this work.

Reference

1. Wang, Y., Fang, J., Liu, B., Shao, C. & Shi, Y. Reciprocal regulation of mesenchymal stem cells and immune responses. *Cell Stem Cell* **29**, 1515-1530 (2022).
2. Rice, C.M., Kemp, K., Wilkins, A. & Scolding, N.J. Cell therapy for multiple sclerosis: an evolving concept with implications for other neurodegenerative diseases. *Lancet (London, England)* **382**, 1204-1213 (2013).
3. Fiorina, P., Voltarelli, J. & Zavazava, N. Immunological applications of stem cells in type 1 diabetes. *Endocrine reviews* **32**, 725-754 (2011).
4. Lu, L.L., *et al.* Isolation and characterization of human umbilical cord mesenchymal stem cells with hematopoiesis-supportive function and other potentials. *Haematologica* **91**, 1017-1026 (2006).
5. Zhang, Y., *et al.* Effects of human umbilical cord-derived mesenchymal stem cells on anterior chamber-associated immune deviation. *International immunopharmacology* **15**, 114-120 (2013).
6. Du, W., *et al.* VCAM-1+ placenta chorionic villi-derived mesenchymal stem cells display potent pro-angiogenic activity. *Stem cell research & therapy* **7**, 49 (2016).
7. Lu, S., *et al.* CD106 is a novel mediator of bone marrow mesenchymal stem cells via NF- κ B in the bone marrow failure of acquired aplastic anemia. *Stem cell research & therapy* **8**, 178 (2017).
8. Wei, Y., *et al.* JNKi- and DAC-programmed mesenchymal stem/stromal cells from hESCs facilitate hematopoiesis and alleviate hind limb ischemia. *Stem cell research & therapy* **10**, 186 (2019).
9. Zhao, Q., *et al.* Systematic comparison of hUC-MSCs at various passages reveals the variations of signatures and therapeutic effect on acute graft-versus-host disease. *Stem cell research & therapy* **10**, 354 (2019).
10. Wei, Y., *et al.* High-efficient generation of VCAM-1(+) mesenchymal stem cells with multidimensional superiorities in signatures and efficacy on aplastic anaemia mice. *Cell Prolif* **53**, e12862 (2020).
11. Sun, T., *et al.* Multilevel defects in the hematopoietic niche in essential thrombocythemia. *Haematologica* **105**, 661-673 (2020).
12. Zhang, L., *et al.* Bone marrow-derived mesenchymal stem/stromal cells in patients with acute myeloid leukemia reveal transcriptome alterations and deficiency in cellular vitality. *Stem cell research & therapy* **12**, 365 (2021).
13. Soliman, H., *et al.* Multipotent stromal cells: One name, multiple identities. *Cell Stem Cell* **28**, 1690-1707 (2021).
14. Caplan, A.I. Mesenchymal Stem Cells: Time to Change the Name! *Stem Cells Transl Med* **6**, 1445-1451 (2017).
15. Keating, A. Mesenchymal stromal cells: new directions. *Cell Stem Cell* **10**, 709-716 (2012).
16. Kowalczyk, M.S., *et al.* Single-cell RNA-seq reveals changes in cell cycle and differentiation programs upon aging of hematopoietic stem cells. *Genome research* **25**, 1860-1872 (2015).
17. Nguyen, Q.H., *et al.* Single-cell RNA-seq of human induced pluripotent stem cells

- reveals cellular heterogeneity and cell state transitions between subpopulations. *Genome research* **28**, 1053-1066 (2018).
18. Ranzoni, A.M., *et al.* Integrative Single-Cell RNA-Seq and ATAC-Seq Analysis of Human Developmental Hematopoiesis. *Cell Stem Cell* **28**, 472-487.e477 (2021).
 19. Zhu, Y., *et al.* Characterization and generation of human definitive multipotent hematopoietic stem/progenitor cells. *Cell discovery* **6**, 89 (2020).
 20. Ibarra-Soria, X., *et al.* Defining murine organogenesis at single-cell resolution reveals a role for the leukotriene pathway in regulating blood progenitor formation. *Nat Cell Biol* **20**, 127-134 (2018).
 21. Zeng, Y., *et al.* Tracing the first hematopoietic stem cell generation in human embryo by single-cell RNA sequencing. *Cell Res* **29**, 881-894 (2019).
 22. Athanasiadis, E.I., *et al.* Single-cell RNA-sequencing uncovers transcriptional states and fate decisions in haematopoiesis. *Nat Commun* **8**, 2045 (2017).
 23. Fidanza, A., *et al.* Single-cell analyses and machine learning define hematopoietic progenitor and HSC-like cells derived from human PSCs. *Blood* **136**, 2893-2904 (2020).
 24. Xia, J., *et al.* A single-cell resolution developmental atlas of hematopoietic stem and progenitor cell expansion in zebrafish. *Proc Natl Acad Sci U S A* **118**(2021).
 25. Wu, C.L., *et al.* Single cell transcriptomic analysis of human pluripotent stem cell chondrogenesis. *Nat Commun* **12**, 362 (2021).

Appendix Fig. 1: Remove cell cycle related genes (Geneset: GOBP_CELL_CYCLE_GO.0007049, n=1847) from Variable Features, avoid its effect on the downstream PCA analysis

Appendix Figure 1. Effect of removal of cell cycle genes on clustering

(a), UMAP showing seven cluster of MSCs after remove cell cycle related genes from Variable Features. (b), Dot plots showing the scaled expression of representative senescent-related genes in RNA assay for each cluster calculated in appendix Fig. 1a. (c), Correlation matrices showing the Pearson correlation coefficients of the 7 MSC clusters calculated after remove cell cycle related genes and 7 MSC clusters as shown in Fig. 1b. (d), Fractions of subpopulations in six adult versus six perinatal MSC samples (Error bar: Mean \pm s.d.s). (e), Scatter plot comparing the changes in cell frequencies for each cluster before and after removal of cycle genes. P values are generated by two-tailed t test with Welch's correction (* $p < 0.05$; ** $p < 0.005$; *** $p < 0.0005$).

Appendix Fig. 2: Assign each single cell a Cell Cycle Score and Regress out Cell Cycle Score using Seurat ScaleData function

Appendix Figure 2. Effect of regressing out cell cycle score on clustering

(a), UMAP showing seven clusters of MSCs after regress out cell cycle scoring by Seurat (b), Dot plots showing the scaled expression of representative senescent-related genes in RNA assay for each cluster calculated in appendix Fig. 2a. (c), Correlation matrices showing the Pearson correlation coefficients of the 7 MSC clusters calculated after regress out cell cycle scoring and 7 MSC clusters as shown in Fig. 1b (d), Fractions of subpopulations in six adult versus six perinatal MSC samples (Error bar: Mean \pm s.d.s). (e), Scatter plot comparing the changes in cell frequencies for each cluster before and after regress out cell cycle score. P values are generated by two-tailed t test with Welch's correction (* $p < 0.05$; ** $p < 0.005$; *** $p < 0.0005$).

Appendix Fig. 3: Assign each single cell a Cell Cycle Score and Regress out Cell Cycle Score using Scanpy regress_out function

Appendix Figure 3. Effect of regressing out cell cycle score on clustering

(a), UMAP showing seven clusters of MSCs after regress out cell cycle scoring by scanpy (b), Dot plots showing the scaled expression of representative senescent-related genes in RNA assay for each cluster calculated in appendix Fig. 2a. (c), Correlation matrices showing the Pearson correlation coefficients of the 7 MSC clusters calculated after regress out cell cycle scoring and 7 MSC clusters as shown in Fig. 1b (d), Fractions of subpopulations in six adult versus six perinatal MSC samples (Error bar: Mean \pm s.d.s). (e), Scatter plot comparing the changes in cell frequencies for each cluster before and after regress out cell cycle score. P values are generated by two-tailed t test with Welch's correction (* $p < 0.05$; ** $p < 0.005$; *** $p < 0.0005$).

REVIEWER COMMENTS

Reviewer #1 (Remarks to the Author):

This revised version is much improved especially in uncovering the heterogeneity between different sources of human MSCs, as well as providing strong evidence for less senescence in chronologically youthful perinatal MSCs compared to adult MSCs and its inverse relationship to immunosuppressive properties. However, the 'add-back' of MSC stemness still results in similar problems of the initial version of the manuscript.

1) Directly equating the 3 pluripotency factors to stand for MSC stemness factors is, at best inappropriate and at worst wrong/unproven. If the authors wish to use these 3 factors, then these genes must be labeled appropriately as "pluripotency stem cell signature genes" and not generically as "stem cell signature genes", and then further referenced as to why these factors perhaps may act to be select for MSCs/non-pluripotent stem cells (i.e. Pochampally et al, Blood 2004 for hBMMSCs; Fukuchi et al, Stem Cells 2004 for fetal placental MSCs).

Specific comments:

2) The use of the term 'MSC stemness' is quite controversial as the authors acknowledge, which is also reflected in the literature by the paucity of publications using this term, therefore use of another and more descriptive term(s) is advised, i.e. "MSC multi-lineage differentiation capacity".

3) Since the authors still include tri-lineage differentiation data, and the definition for any stem cell critically includes multi-lineage potential, it would be more logical to assess the tri-lineage differentiation transcriptomic programs (i.e. Runx2/Osterix & downstream osteogenic genes; CEBPb/PPARg & downstream adipogenic genes; Sox9 & downstream chondrogenic genes) to stand for MSC stemness. Moreover, the authors continue to use the term "stem cells" rather than "stromal cells" throughout the manuscript, therefore incorporation of bioinformatics analyses of differentiation transcriptional programs would not only be appropriate but important. In fact, since the authors already found that in functional differentiation assays senescent MSCs are more prone to adipogenesis than osteogenesis, it is definitely worthwhile to add bioinformatics data relevant to these functional differentiation assays.

Minor comments:

a) Line 170: perhaps adding or replacing "biochemical" with "molecular" is more accurate (this was done in the summary sentence of the paragraph/section)

b) Lines 473-4: perhaps adding "at baseline" after "senescence" in "adult tissue-derived MSCs have a higher degree of senescence AT BASELINE than perinatal MSCs" would be useful to readers.

Reviewer #2 (Remarks to the Author):

In the present submission, the authors have addressed nearly all of this reviewer's concerns regarding single cell RNA-Seq (scRNA-seq) analysis. Specific comments and remaining issues are detailed below:

Cell cycle effects on clustering

While the extensive effort the authors have undertaken to assess the impact of cell cycle associated gene expression is impressive and welcome, as they now note in their updated text, based on the biology under investigation, it is difficult if not impossible to fully separate cell cycle-associated gene expression patterns from senescence. This confounding issue is not necessarily a problem as it is managed with supporting experimental data; the key issue is to address it in interpretation of analysis results. The newly updated language addresses this issue. It would be further improved by also including some of the language from the rebuttal: "After we removed the effect of the cell cycle, there were no significant changes in cell clustering, and the results of the cell cycle regression did not affect

our major conclusions of this study.” It should not be necessary to include the rebuttal appendix data in the final manuscript.

Visualization of expression values

The updated figure legend information informing (generally appropriate) use of Seurat “RNA assay” data for gene expression values addresses the main critique of the previous review. However, for those figures that include gene expression data generated from multiple technical batches, it is not fully appropriate to directly compare these values. This is not a major issue; the authors are correct in stating that they “implemented a rigorous algorithm to eliminate false positives caused by single-cell replicates (pseudobulk strategy) and further corrected the batch effect using DESeq2, which ensured the reliability and appropriateness of our new DEG results for visualization.” This is sufficient to ensure that the genes selected for plotting are indeed significant (i.e. not a consequence of batch effects), but has no impact on the expression values themselves which are visualized. At line 1166, the authors state that “Additionally, batch effects were corrected to ensure accurate comparisons.” Assuming “batch effects were corrected” refers to actually transforming the expression values for plotting, there is no evidence in the analysis code that any such correction was applied. None of this is expected to change results or interpretations, but for those figures that include visualization of expression values from multiple batches, it would be most accurate to note “(non-batch corrected)” (or similar) in the figure legends.

Pseudobulk gene set scores

As noted in the previous review, the authors appropriately applied AUCell for per cell gene set scores at single cell resolution. Also in the previous review, a question was raised about using AUCell scores for pseudobulk profiles. While the authors have now provided an explanation and updated analysis code, the question remains: is simply summing AUCell scores into pseudobulk profiles an appropriate / accurate method for gene set testing/activity? This reviewer is not aware of endorsement of this approach by the AUCell developers or any relevant benchmarking study. An alternative approach might be to use well-established “bulk” RNA-Seq gene set tools (e.g. GSEA, QuSage, others) on pseudobulk profiles calculated from single cell gene expression (i.e. rather than calculating gene set scores PER CELL and THEN summing for pseudobulk, sum gene expression to pseudobulk and then run appropriate gene set tools).

Point by point response to reviewers

We sincerely appreciate the efforts of all the reviewers for their careful evaluation and valuable feedback on our manuscript. Following the suggestions by the reviewers, we have made several revisions to the bioinformatics analysis and writing, which further enhanced the accuracy and reliability of our results. Among the highlights of the new results, we have included additional bioinformatic analysis to elucidate the impact of senescence on MSC trilineage differentiation capacity. Additionally, we have optimized our gene set analysis by utilizing the GSVA algorithm on pseudobulk samples calculated from single-cell gene expression data, which has increased the rigor of our analysis. Finally, we have also modified the text description of MSC stemness and revised the interpretation of pluripotency factors to enhance the credibility of our results. The new data are highlighted in a blue font in the revised manuscript, including the legends and labels of figures and supplementary tables. All figure callouts below correspond to figures in the revised manuscript. With the new results, we believe we have addressed all of the reviewers' concerns, substantially improved the accuracy and reliability of our work, and broadened the impacts of our study.

A brief summary of the major new results is listed below, which are divided into two categories.

A. Analysis of the effects of cell senescence on MSC trilineage differentiation capacity.

1. The comparison of adipogenesis score, osteogenesis score and chondrogenesis score of six BM-MSCs clusters derived from young and aged donor (**Supplementary Fig. 5i, Supplementary table 3**).
2. The tri-lineage differentiation scores of young and aged BM-MSC samples were compared (**Supplementary Fig. 5j**).
3. The comparison of the expression level of CEBPB, PPARG, RUNX2 and SOX9 in six BM-MSCs clusters derived from young and aged donors (**Supplementary Fig. 5k**).
4. The expression level of CEBPB, PPARG, RUNX2 and SOX9 in young and aged BM-MSC samples were compared (**Supplementary Fig. 5l**).

B. Adoption of more rigorous strategies for pseudobulk gene set analysis.

1. The 'pseudobulk' strategy were used to assess differential activities of pathway scored by GSVA with batch effect correction by limma package (**Fig. 1c, Supplementary Table 4**).

Reviewers' comments:

Reviewer #1 (Remarks to the Author):

This revised version is much improved especially in uncovering the heterogeneity

between different sources of human MSCs, as well as providing strong evidence for less senescence in chronologically youthful perinatal MSCs compared to adult MSCs and its inverse relationship to immunosuppressive properties. However, the ‘add-back’ of MSC stemness still results in similar problems of the initial version of the manuscript.

We thank the reviewer for the encouraging comments on our revised manuscript.

1) Directly equating the 3 pluripotency factors to stand for MSC stemness factors is, at best inappropriate and at worst wrong/unproven. If the authors wish to use these 3 factors, then these genes must be labeled appropriately as “pluripotency stem cell signature genes” and not generically as “stem cell signature genes”, and then further referenced as to why these factors perhaps may act to be select for MSCs/non-pluripotent stem cells (i.e. Pochampally et al, Blood 2004 for hBMMSCs; Fukuchi et al, Stem Cells 2004 for fetal placental MSCs).

We thank the reviewer for raising this question. We acknowledge that a more rigorous description and interpretation of experimental data related to pluripotency genes is necessary. According to reviewer’s suggestions, we have made the following revisions to the manuscript:

(1) We have re-interpreted the qPCR data of pluripotency stem cell genes mentioned in **line 277**, and have incorporated references (reference 62 and 63) recommended by the reviewer to explain why pluripotency factors may be used to represent the pluripotency of MSCs.

(2) In the discussion section related to pluripotency gene data (**line 658**), we have revised the term "stem signature genes" to "pluripotency stem cell signature genes".

(3) As the stem cell signature gene set used for the gene set score analysis contained pluripotency genes, we have updated the figures and textual description related to stem cell signature gene set analysis (**Supplementary Fig. 1f, Supplementary Fig. 3d, Supplementary Fig. 5g; line 146, line 149, line 272, line 408**) to use the term "pluripotency stem cell signature" instead of "stem cell signature".

Specific comments:

2) The use of the term ‘MSC stemness’ is quite controversial as the authors acknowledge, which is also reflected in the literature by the paucity of publications using this term, therefore use of another and more descriptive term(s) is advised, i.e. “MSC multi-lineage differentiation capacity”.

We thank the reviewer for the constructive comments. According to reviewer’s suggestions, we have replaced the term 'stemness' with more specific and descriptive terms. Specific revisions for this part are shown below:

(1) **line 661:** In this study, we have also investigated the effect of cell aging on MSC **pluripotency gene expression and multi-lineage differentiation capacity**.

(2) **line 662:** Research has shown that aging can have a negative impact on the **pluripotency** and self-renewal capacity of stem cells.

(3) **line 666:** which suggest that the heterogeneity in **pluripotency gene expression** among MSC populations could be related to differences in the senescence states of individual cells.

(4) **line 668:** It is worth noting that although **pluripotency stem cell signature genes** are widely used to demonstrate the self-renewal ability and **multipotent differentiation capacity** of MSCs, whether these genes can fully stand for MSC **pluripotency** in the aging process is still controversial and needs to be proved in future work.

(5) **line 674:** Although the general consensus is that a stronger differentiation potential represents better **pluripotency** of stem cells.

3) Since the authors still include tri-lineage differentiation data, and the definition for any stem cell critically includes multi-lineage potential, it would be more logical to assess the tri-lineage differentiation transcriptomic programs (i.e. Runx2/Osterix & downstream osteogenic genes; CEBPb/PPARG & downstream adipogenic genes; Sox9 & downstream chondrogenic genes) to stand for MSC stemness. Moreover, the authors continue to use the term “stem cells” rather than “stromal cells” throughout the manuscript, therefore incorporation of bioinformatics analyses of differentiation transcriptional programs would not only be appropriate but important. In fact, since the authors already found that in functional differentiation assays senescent MSCs are more prone to adipogenesis than osteogenesis, it is definitely worthwhile to add bioinformatics data relevant to these functional differentiation assays.

We thank the reviewer for raising this question. We have incorporated the following analysis to the manuscript based on the suggestions provided by the reviewers:

(1) We analyzed the expression levels of key transcriptional regulators of MSC adipogenesis (CEBPB and PPARG), osteogenesis (RUNX2; the SP7, also known as Osterix, was not detected in scRNA sequencing data), and chondrogenesis (SOX9) in young and aged BM-MSC scRNA data. We found that the expression of CEBPB and PPARG were significantly increased in the aging BM-MSCs clusters (BM5, BM6), while RUNX2 and SOX9 did not show significant expression differences among clusters (**Supplementary Fig. 5k**). Besides, we also investigated the expression of these transcription factors at the pseudobulk bulk level. While not statistically significant, we observed that CEBPB was upregulated in aged samples at the pseudobulk level, whereas the expression of RUNX2 was decreased in aged BM-MSCs

(**Supplementary Fig. 5l**). These results indicate that senescent MSCs may upregulate the expression of adipogenic transcription factors while downregulating the expression of osteogenic regulators.

(2) We collected gene sets related to MSC adipogenesis (which includes CEBPB/PPARG and its downstream adipogenic genes), osteogenesis (which includes RUNX2 and its downstream osteogenic genes) and chondrogenesis (which includes SOX9 and its downstream chondrogenic genes), and these gene sets were used to calculate gene set scores for assessing the trilineage differentiation capacity of MSCs (**Supplementary table 3**). Firstly, we calculated the gene set scores for each BM-MSC cluster (**Supplementary Fig. 5i**). The results showed that the adipogenesis score increased in the aging clusters (BM5, BM6), while these clusters exhibited decreased expression of osteogenesis-related genes. Furthermore, we found that BM-MSCs derived from aged donors exhibited upregulation of adipogenic genes but decreased expression of osteogenic genes (**Supplementary Fig. 5j**). Although no significant difference in chondrogenic ability was observed between young and aged BM-MSCs during the in vitro differentiation experiment, our transcriptome-level analysis suggested that the expression of chondrogenic-related genes was downregulated in aged BM-MSCs. (**Supplementary Fig. 5i-j**). These findings suggest that senescence may disturb the balance of MSC multi-lineage differentiation ability, resulting in weakened osteogenesis and increased adipogenic tendency, which is consistent with the results of the in vitro differentiation experiments (**Supplementary Fig. 5a**).

(3) We have revised the title of the fifth section to: “Altered multi-lineage differentiation ability and impaired immunosuppressive capacity were observed in BM-MSCs from aged donors” (**line 381**). The textual description of the above bioinformatics analysis is provided in **line 424**.

Minor comments:

a) Line 170: perhaps adding or replacing “biochemical” with “molecular” is more accurate (this was done in the summary sentence of the paragraph/section)

We thank the reviewer for the valuable suggestion. We have revised the title of the second section to “Molecular basis of human MSC senescence.” (**line 171**)

b) Lines 473-4: perhaps adding “at baseline” after “senescence” in “adult tissue-derived MSCs have a higher degree of senescence AT BASELINE than perinatal MSCs” would be useful to readers.

We thank the reviewer for the valuable suggestion. We have revised the sentence to read as follows: “These results suggest a significant correlation between our transcriptome and proteome data for MSCs and provide evidence from a multiomics perspective to demonstrate that adult tissue-derived MSCs have a higher degree of senescence at baseline than perinatal MSCs.” (**line 503**)

Reviewer #2 (Remarks to the Author):

In the present submission, the authors have addressed nearly all of this reviewer's concerns regarding single cell RNA-Seq (scRNA-seq) analysis. Specific comments and remaining issues are detailed below:

We are grateful for reviewer's kind comments on our revised manuscript.

Cell cycle effects on clustering

While the extensive effort the authors have undertaken to assess the impact of cell cycle associated gene expression is impressive and welcome, as they now note in their updated text, based on the biology under investigation, it is difficult if not impossible to fully separate cell cycle-associated gene expression patterns from senescence. This confounding issue is not necessarily a problem as it is managed with supporting experimental data; the key issue is to address it in interpretation of analysis results. The newly updated language addresses this issue. It would be further improved by also including some of the language from the rebuttal: "After we removed the effect of the cell cycle, there were no significant changes in cell clustering, and the results of the cell cycle regression did not affect our major conclusions of this study." It should not be necessary to include the rebuttal appendix data in the final manuscript.

We appreciate the reviewer's valuable suggestion. To further clarify the impact of cell cycle genes on MSC clustering, we have added the following statement in **line 609** "After we removed the effect of the cell cycle, there were no significant changes in cell clustering, and the results of the cell cycle regression did not affect our major conclusions of this study."

Visualization of expression values

The updated figure legend information informing (generally appropriate) use of Seurat "RNA assay" data for gene expression values addresses the main critique of the previous review. However, for those figures that include gene expression data generated from multiple technical batches, it is not fully appropriate to directly compare these values. This is not a major issue; the authors are correct in stating that they "implemented a rigorous algorithm to eliminate false positives caused by single-cell replicates (pseudobulk strategy) and further corrected the batch effect using DESeq2, which ensured the reliability and appropriateness of our new DEG results for visualization." This is sufficient to ensure that the genes selected for plotting are indeed significant (i.e. not a consequence of batch effects), but has no impact on the expression values themselves which are visualized. At line 1166, the authors state that "Additionally, batch effects were corrected to ensure accurate comparisons." Assuming "batch effects were corrected" refers to actually transforming the expression values for plotting, there is no evidence in the analysis code that any such correction was applied.

We thank the reviewer for the constructive comments. We apologize for the previous language error, and we would like to clarify that our original intention was to convey that the differential genes visualized in the figure were obtained from DESeq2 differential analysis with batch correction. To rectify this error, we have revised the sentence in **line 1205** as follows: “Furthermore, to ensure that only significant genes were selected for plotting and that they were not the result of batch effects, we used DESeq2 to correct for batch effects in the differential gene analysis (design = ~ batch + pseudobulk cluster).”

None of this is expected to change results or interpretations, but for those figures that include visualization of expression values from multiple batches, it would be most accurate to note “(non-batch corrected)” (or similar) in the figure legends.

We thank the reviewer for the valuable suggestion. To address the issue of figures that display visualization of expression values from multiple batches, we have included the following description in the corresponding figure legend: “The displayed values were non-batch corrected.”

Pseudobulk gene set scores

As noted in the previous review, the authors appropriately applied AUCell for per cell gene set scores at single cell resolution. Also in the previous review, a question was raised about using AUCell scores for pseudobulk profiles. While the authors have now provided an explanation and updated analysis code, the question remains: is simply summing AUCell scores into pseudobulk profiles an appropriate / accurate method for gene set testing/activity? This reviewer is not aware of endorsement of this approach by the AUCell developers or any relevant benchmarking study. An alternative approach might be to use well-established “bulk” RNA-Seq gene set tools (e.g. GSVA, QuSage, others) on pseudobulk profiles calculated from single cell gene expression (i.e. rather than calculating gene set scores PER CELL and THEN summing for pseudobulk, sum gene expression to pseudobulk and then run appropriate gene set tools).

We thank the reviewer for raising this question. We recognize that in our case, which involves batch differences and biological replicates, it is more appropriate to use bulk RNA-Seq gene set algorithms on “pseudobulk” samples calculated from single-cell gene expression data. To achieve this, we summed the gene expression values in each cluster to create pseudobulk profiles, and subsequently computed the GSVA score for each pseudobulk sample. However, since the GSVA calculation produced negative values that are not integers, it was not suitable to use the DESeq2 algorithm for differential expression analysis and batch correction, as this method requires positive integer values. Instead, we utilized the limma package to remove batch effects and perform differential expression analysis. The "removeBatchEffect" function from the limma package was employed to eliminate potential impact of batch differences on the calculation of GSVA scores. Subsequently, we used limma to calculate the differential activities of pathways, and the mean GSVA scores for all pseudobulk samples per

cluster were displayed in the heatmap (**Fig. 1c; Supplementary table 4**). We have supplemented the methods of this part in **line1052**, and the codes used for this analysis have been uploaded to Github

(https://github.com/GaoYuchenPUMC/MS_C_paper/blob/main/code_for_Revision%233/Pseudobulk_GSVA.R).

REVIEWERS' COMMENTS

Reviewer #1 (Remarks to the Author):

The last revisionary efforts are very useful, but just a few corrections are still needed since the authors appear to still have misunderstood basic concepts of MSC differentiation capacity vs. pluripotency (which MSCs and other adult/somatic stem cells do not possess).

Line 235: please add "factors" after "pluripotency"

Lines 278-79: please delete "represent the pluripotent signature of" and replace with revise "be better selection markers for"

Line 281: please replace "pluripotency" with "multipotent differentiation capacity"

Line 671: please replace "pluripotency" with "stem cell capacity"

Line 675: "pluripotency" with "differentiation capacity"

Reviewer #2 (Remarks to the Author):

The authors have addressed nearly all of the previously articulated concerns regarding single cell RNA-Seq (scRNA-Seq) and related analyses. One remaining minor issue related to gene set analysis of pseudobulk scRNA-Seq data is detailed below.

The application of GSVA for geneset enrichment testing on pseudobulk scRNA-Seq values addresses previous concerns regarding the appropriateness of summing/aggregating single cell gene set values for grouped analyses. However, a minor improvement to the workflow may make for more robust results. As implemented, the authors take the pseudobulk count matrix, "correct" for batch effects using the `removeBatchEffect` function (limma), calculate GSVA scores, and then test for differential gene set enrichment in limma. Rather than "correcting" counts, which can have unintended effects, the workflow could be modified to take the pseudobulk count matrix, calculate GSVA scores, and test for differential gene set enrichment in limma while accounting for batch in the corresponding design matrix (i.e. \sim batch + pseudobulk cluster), as has been implemented elsewhere in the manuscript for differential gene expression testing. For visualization (e.g. Fig 1C), median GSVA values can be used.

Line 121: Resulting GSVA scores should not be referred to as "single-cell enrichment scores"; "pseudobulk" or similar would be more appropriate.

Point by point response to reviewers

We sincerely appreciate the encouraging and constructive comments provided by all the reviewers. Following the valuable suggestions from the reviewers, we have made further enhancements to the methodology of the bioinformatics analysis and revised the textual content of the manuscript. We have made revisions to the structure of the manuscript and enhanced the descriptions of the statistical methods, as well as other sections. The new data are highlighted in a blue font in the revised manuscript, including the legends and labels of figures.

Reviewers' comments:

Reviewer #1 (Remarks to the Author):

The last revisionary efforts are very useful, but just a few corrections are still needed since the authors appear to still have misunderstood basic concepts of MSC differentiation capacity vs. pluripotency (which MSCs and other adult/somatic stem cells do not possess).

We thank the reviewer for the encouraging comments on our revised manuscript.

Line 235: please add “factors” after “pluripotency”

Lines 278-79: please delete "represent the pluripotent signature of" and replace with revise "be better selection markers for”

Line 281: please replace “pluripotency” with “multipotent differentiation capacity”

Line 671: please replace “pluripotency” with “stem cell capacity”

Line 675: “pluripotency” with “differentiation capacity”

We thank the reviewer for raising this question. We have revised all corresponding words in the manuscript based on the suggestions from the reviewer.

Reviewer #2 (Remarks to the Author):

The authors have addressed nearly all of the previously articulated concerns regarding single cell RNA-Seq (scRNA-Seq) and related analyses. One remaining minor issue related to gene set analysis of pseudobulk scRNA-Seq data is detailed below.

We thank the reviewer for the encouraging comments.

The application of GSVA for geneset enrichment testing on pseudobulk scRNA-Seq values addresses previous concerns regarding the appropriateness of summing/aggregating single cell gene set values for grouped analyses. However, a minor improvement to the workflow may make for more robust results. As

implemented, the authors take the pseudobulk count matrix, “correct” for batch effects using the `removeBatchEffect` function (limma), calculate GSVA scores, and then test for differential gene set enrichment in limma. Rather than “correcting” counts, which can have unintended effects, the workflow could be modified to take the pseudobulk count matrix, calculate GSVA scores, and test for differential gene set enrichment in limma while accounting for batch in the corresponding design matrix (i.e. \sim batch + pseudobulk cluster), as has been implemented elsewhere in the manuscript for differential gene expression testing. For visualization (e.g. Fig 1C), median GSVA values can be used.

We thank the reviewer for the valuable suggestion. In response to this suggestion, we summed the gene expression values within each cluster to generate pseudobulk profiles. Subsequently, we calculated the GSVA score for each pseudobulk cluster. We then utilized the limma package to perform differential gene set scoring analysis. The parameter of the `model.matrix` function was set to ' \sim 0+pseudobulk_cluster+batch' to further remove the influence of batch effects on limma differential analysis. The updated results, including the heatmap and supplementary tables (**Fig. 1c; Supplementary table 4**), have been updated. We have supplemented the methods of this part in **line 1054**, and the codes used for this analysis have been uploaded to Github (https://github.com/GaoYuchenPUMC/MS_C_paper/blob/main/code_for_Revision%234/GSVA_analysis.R)

Line 121: Resulting GSVA scores should not be referred to as “single-cell enrichment scores”; “pseudobulk” or similar would be more appropriate.

We thank the reviewer for raising this question. We have revised “single-cell enrichment scores” in **line 112** to “enrichment scores for each pseudobulk cluster”.